# $T\overline{T}$ in JT gravity and BF gauge theory

**Stephen Ebert[1], Christian Ferko[2], Hao-Yu Sun[3] and Zhengdi Sun[4]**

**1** Mani L. Bhaumik Institute for Theoretical Physics, University of California,
Los Angeles, CA, 90095, USA
**2** Center for Quantum Mathematics and Physics (QMAP),
Department of Physics & Astronomy, University of California, Davis, CA 95616, USA
**3** Theory Group, Department of Physics, University of Texas, Austin, TX 78712, USA
**4** Department of Physics, University of California, San Diego, CA 92093, USA

## Abstract

JT gravity has a first-order formulation as a two-dimensional BF theory, which can be viewed as the dimensional reduction of the Chern-Simons description of $3d$ gravity. We consider $T\overline{T}$-type deformations of the $(0 + 1)$-dimensional dual to this $2d$ BF theory and interpret the deformation as a modification of the BF theory boundary conditions. The fundamental observables in this deformed BF theory, and in its $3d$ Chern-Simons lift, are Wilson lines and loops. In the $3d$ Chern-Simons setting, we study modifications to correlators involving boundary-anchored Wilson lines which are induced by a $T\overline{T}$ deformation on the $2d$ boundary; results are presented at both the classical level (using modified boundary conditions) and the quantum-mechanical level (using conformal perturbation theory). Finally, we calculate the analogous deformed Wilson line correlators in $2d$ BF theory below the Hagedorn temperature where the principal series dominates over the discrete series.

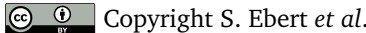

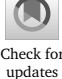

# 1 Introduction

Pure 3$d$ gravity (in either asymptotically AdS$_3$, dS$_3$ or Minkowski backgrounds) is defined by the Einstein-Hilbert action and the usual Gibbons-Hawking-York boundary term, which in Euclidean signature is

$$I[g_{\mu\nu}] = -\frac{1}{16\pi G}\int_{M_3} d^3x\sqrt{g}\,(R - 2\Lambda) - \frac{1}{8\pi G}\int_{\partial M_3} d^2x\sqrt{h}\,(K - 1)\,. \tag{1}$$

Here $G$ is the 3$d$ Newton constant, $R$ is the Ricci scalar of the 3$d$ metric $g_{\mu\nu}$, $\Lambda$ is the cosmological constant, and $K$ is the trace of the extrinsic curvature.

Because it is simpler than gravitational theories living in spacetime dimensions greater than three, 3$d$ gravity has been – and continues to be – a very successful theoretical laboratory probing several important features of quantum gravity in low dimensions. In 3$d$ gravity, there are no propagating bulk degrees of freedom (i.e., there are no gravitational waves) due to the Weyl tensor vanishing identically, but the theory still admits black hole solutions of finite mass and Bekenstein-Hawking entropy [1]. Before the full statement of AdS/CFT duality [2–4] was proposed, an important early discovery was found by Brown and Henneaux [5] where they showed that AdS$_3$ contains two copies of an asymptotic Virasoro symmetry with non-vanishing central charge $c = \frac{3}{2G}$ up to loop corrections in $G$.[1] The large central charge limit corresponds to semi-classical behavior, whereas finite central charge leads to fully quantized AdS$_3$ gravity. The quantum theory is described by boundary gravitons under Virasoro symmetry [5,6] equipped with a well-defined Hilbert space and a spectrum of local operators.

There are two salient features of 3$d$ gravity motivating this work. The first feature is that the 3$d$ Einstein-Hilbert action (1) in the first order formulation may be written semi-classically

---

[1]We choose units such that the AdS length scale is unity.

as a Chern-Simons gauge theory, as observed by [7,8] (analogous theories of gravity coupled to higher-spin fields can be obtained by enlarging the Chern-Simons gauge group). One can then calculate fundamental observables in the gravitational Chern-Simons theory, such as Wilson lines and loops obtained from path-ordered exponential integrals of the one-form connection $A_\mu(x)$ along an open interval and a closed contour respectively.

In what follows, we will refer to Wilson lines for the Chern-Simons gauge field $A_\mu$ as "gravitational Wilson lines." Given two endpoints $Z_1 = (r_1, z_1)$ and $Z_2 = (r_2, z_2)$ on the AdS$_3$ boundary, the gravitational Wilson line anchored between $Z_1$ and $Z_2$ is written

$$W[Z_2, Z_1] = P \exp\left( \int_{Z_1}^{Z_2} A_\mu(x)\, dx^\mu \right). \tag{2}$$

In the classical (i.e., large-$c$) limit, the object $W[Z_2, Z_1]$ has the peculiar property that it transforms as a bi-local primary operator at its endpoints. Evidence was presented in [9, 10] that, at least perturbatively in $\frac{1}{c}$, it appears that the *quantum* Wilson line transforms as a bi-local primary operator at its endpoints as well. As argued in [9–16], another useful feature of the Wilson line is that it serves as a convenient repackaging of the Virasoro vacuum OPE block

$$\langle T_{zz}(w_1) \cdots T_{zz}(w_n) W[z_2, z_1] \rangle_0 = \langle T_{zz}(w_1) \cdots T_{zz}(w_n) O(z_2) O(z_1) \rangle_0, \tag{3}$$

where $\langle W[z_2, z_1] \rangle_0$ will be later defined by (202) in terms of a path-ordered exponential integral involving only the stress tensor operator's holomorphic component $T_{zz}$.

The Virasoro vacuum OPE block – whose characteristics are similar to those of $\langle W[z_2, z_1] \rangle$ – precisely captures all operators built out of the stress tensor operator appearing in the OPE of two primary operators $O(z_1)$ and $O(z_2)$. Schematically (suppressing numerical factors, coordinate dependence, and derivatives in the OPE coefficient $C_{OOO_i}$, as well as omitting the $T_{\bar{z}\bar{z}}$ piece), the first term in the following OPE of two primary operators

$$O(z_2) O(z_1) = (1 + T_{zz} + T_{zz} T_{zz} + \cdots) + \sum_i C_{OOO_i} (O_i + O_i T_{zz} + O_i T_{zz} T_{zz} + \cdots), \tag{4}$$

corresponds to the Virasoro vacuum OPE block.

From the bulk perspective, the (open) gravitational Wilson line computes the exponential of the worldline action for a massive point particle including the effects of gravitational self-interaction which renormalize its mass [9, 10, 17, 18]. The closed gravitational Wilson line or, in other words, the Wilson loop measures the holonomy of the gauge connection. In the context of a BTZ black hole, when the Wilson loop wraps around the horizon, its value yields the Bekenstein-Hawking entropy [19, 20].

The second feature is that, upon dimensionally reducing 3$d$ gravity (1) on a circle with radius equal to the dilaton $\Phi$, one obtains the JT gravity action[2]

$$I[g_{\mu\nu}^{(2)}, \Phi] = -\frac{1}{16\pi G} \int_{M_2} d^2 x \sqrt{g^{(2)}} \Phi (R - 2\Lambda) - \frac{1}{8\pi G} \int_{\partial M_2} d\tau \sqrt{\gamma} \Phi (K - 1), \tag{5}$$

where $g_{\mu\nu}^{(2)}$ is the 2$d$ bulk metric and $\gamma_{\tau\tau}$ is the 1$d$ boundary metric. The holographic dual description of JT gravity is the 1$d$ Schwarzian theory [24–28], which is the universal low-energy limit of the SYK model [29, 30].

Just as one can reduce 3$d$ gravity on a circle in the metric formalism, one may also perform the dimensional reduction for the Chern-Simons description of 3$d$ gravity and obtain the so-called BF gauge theory which enjoys similar features to 2$d$ Yang-Mills gauge theory with

---

[2]The fact that dimensional reduction of 3$d$ gravity on a circle produces JT gravity was first noticed by [21]. For more modern observations involving the interplay between 3$d$ gravity and JT gravity, see [22, 23].

a (non-)compact group. See the incomplete list of references [31–37] for discussions on 2$d$ Yang-Mills gauge theory with (non-)compact groups. BF theory with gauge group $G$ is holographically dual to the worldline theory of a particle with a free kinetic Lagrangian and which moves on target space $G$; we call this the "particle-on-a-group" theory. Given these complementary perspectives and technical advantages of lower dimensional gravity, it is desirable to understand these corners of the following diagram below in different settings.

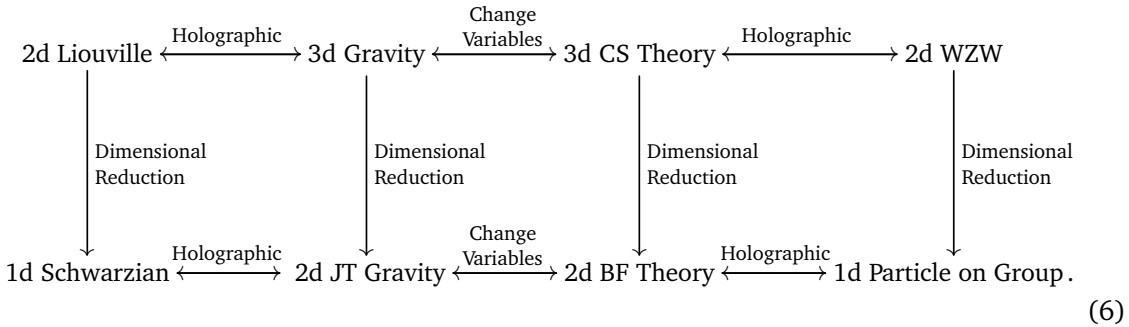

$$\tag{6}$$

One such setting of interest is to study this diagram under Zamolodchikov's irrelevant double-trace $T\overline{T}$ operator [38].[3] The 2$d$ $T\overline{T}$ operator is a bilinear operator constructed of the stress tensor $T_{\mu\nu}$ which can be expressed as

$$\det\left(T_{\mu\nu}\right) = \frac{1}{2}\left(\left(T^{\mu}_{\ \mu}\right)^2 - T^{\mu\nu}T_{\mu\nu}\right), \tag{7}$$

and is unambiguously defined by point-splitting up to total derivatives of local operators that can be neglected [38, 43]. At the classical level, given a seed theory's Lagrangian $\mathcal{L}^{(0)}$, the $T\overline{T}$ flow is captured by the differential equation

$$\frac{\partial \mathcal{L}^{(\lambda)}}{\partial \lambda} = -2\det\left(T^{(\lambda)}_{\mu\nu}\right), \tag{8}$$

where the notation $T^{(\lambda)}_{\mu\nu}$ emphasizes that at each step along the flow, the stress tensor is re-computed from the deformed Lagrangian $\mathcal{L}^{(\lambda)}$.

Irrelevant deformations are notoriously difficult to understand, compared to marginal and relevant deformations. Turning on an irrelevant operator will generically turn on infinitely many additional operators at high energies, which modifies the theory in the UV and lead to a loss of predictive power. However, the $T\overline{T}$ deformation is one irrelevant operator which circumvents these technical difficulties: a $T\overline{T}$-deformed quantum field theory remains under some analytic control and is "solvable" in a sense which we will make precise shortly. Because this operator is irrelevant, a $T\overline{T}$ flow may seem to be the opposite of a conventional renormalization group flow, which is triggered by the addition of a relevant operator. However, a conventional RG flow connects a family of local QFTs which are controlled by an RG fixed point in the UV. It is known that a $T\overline{T}$-deformed field theory is not a local QFT, and thus not controlled by a CFT in the UV, so this flow is *not* like a conventional RG flow even in reverse.

A remarkable property which follows from the behavior of $T\overline{T}$ one-point functions is that the finite-volume spectrum of a $T\overline{T}$-deformed theory satisfies a differential equation which relates energies in the deformed theory to those in the seed theory [43, 44]. This is an example of what we mean by saying that the deformation is "solvable." More precisely, the $T\overline{T}$

---

[3]There is a single-trace $T\overline{T}$ operator, often referred to as $D(x)$ following the notation of [39], which has connections to little string theory [40–42]. The results in this paper will not address the single-trace version, but applying this single-trace deformation to a boundary CFT dual to an AdS spacetime dramatically changes the spacetime's large-$r$ behavior from asymptotically AdS to asymptotically linear dilaton.

deformation is said to be solvable because certain quantities in the deformed theory – which also include the torus partition function [45–47], flat space S-matrix [48, 49] and correlation functions [50–61] – can be computed in terms of the corresponding data in the undeformed theory. Furthermore, the $T\overline{T}$ operator in $2d$ preserves several symmetries of a given seed theory. For example, some preserved symmetries include integrability [43, 49] and supersymmetry [62–66]; for additional results on manifestly supersymmetric $T\overline{T}$-like flows, see [67–72]. However, conformal symmetry is not preserved because the flow parameter $\lambda$ introduces a dimensionful scale in the theory.

More than half a decade has passed since the founding developments by [43, 44], and a plethora of diverse applications of the $T\overline{T}$ deformation have been thoroughly investigated. We will not survey these applications, but instead refer the reader to [73] for a comprehensive review. However, we will mention two proposals for interpreting the $T\overline{T}$ deformation in holography which are relevant to the results presented in this paper.

The first proposal is that adding the double-trace $T\overline{T}$ deformation to a seed theory's action, when $\lambda > 0$,[4] corresponds to cutting off the bulk spacetime at a finite radial distance $r_c$. The authors of [74] first noticed this in AdS$_3$/CFT$_2$, and the $T\overline{T}$-deformed boundary theory is interpreted to be located at a finite radius $r_c = \frac{\pi\lambda}{4G}$. Although the boundary theory at the finite cutoff is non-conformal, there is evidence that the holographic conjecture still holds under $T\overline{T}$ from perturbatively matching the bulk and deformed boundary planar correlators up to two-loops [50, 60]. This prescription reproduces some of the properties of the $T\overline{T}$ deformation; however, it suffers from conceptual challenges because infinitely many energy levels in the dual field theory are complex-valued for the positive sign of the deformation parameter. Additionally, it is also not clear whether one can consistently impose Dirichlet boundary conditions at a finite radius in a theory of gravity.

Furthermore, despite the successful perturbative agreement between deformed bulk and boundary position space correlators, they may not be trustworthy in the sense that one does not expect these correlation functions to exist non-perturbatively. There are indications from [41, 42] that the position space correlators under the single-trace version of $T\overline{T}$ are not well-defined due to a vanishing radius of convergence in conformal perturbation theory.[5]

The second holographic proposal is for the "good sign," $\lambda < 0$. This choice of sign modifies the asymptotic boundary conditions for the metric as $r \to \infty$, but does *not* cut off the spacetime at finite $r$ [76]. Although this modification imposes the Dirichlet condition at finite radial cutoff for the "bad sign" of the deformation parameter $\lambda$, the prescription can also be applied in the case of the "good sign," for which the bulk theory still extends to $r \to \infty$. In this context, the entire spectrum of the boundary field theory remains real, at least for sufficiently small values of $\lambda$.

These two holographic interpretations of the $T\overline{T}$ deformation in the metric formalism for $3d$ gravity have been studied in the Chern-Simons formalism by [60, 77–81]. In particular, the Chern-Simons analysis of [77] studies $3d$ gravity in Lorentzian signature for the good sign of $\lambda$ and shows that a $T\overline{T}$ deformation of the dual CFT corresponds to a modification of the boundary conditions for the gauge field $A_\mu$. These deformed boundary conditions can be interpreted as defining a new variational principle where a certain linear combination of the undeformed source and expectation value is held fixed, and this combination is thus treated as a new deformed source. On the other hand, as motivated earlier in the introduction, the Wilson line is a fundamental observable in the gravitational Chern-Simons formalism and therefore it is natural to study this observable in the deformed theory. We will see in our classical AdS$_3$ Wilson line analysis that this interpretation of the boundary $T\overline{T}$ deformation as a linear mixing of sources and expectation values can be used to understand the effect of such a deformation

---

[4]We refer to $\lambda > 0$ as the "bad sign" of the deformation parameter, whereas the "good sign" is $\lambda < 0$.

[5]The momentum space correlators under the single-trace deformation were studied in [75].

on Wilson line observables, and in particular the result is consistent with the previous analyses of $T\overline{T}$-deformed scalar correlators [50, 51, 53].

To complete the corners of low-dimensional gravity in the diagram (6), we wish to understand the bottom half under the $T\overline{T}$ deformation. The dimensional reduction from $3d$ to JT gravity, its dual Schwarzian description, and partition functions in the metric formalism were originally studied by [82, 83] for the good sign of $\lambda$.[6] However, unlike the extensive literature on $T\overline{T}$ deformations in the metric and Chern-Simons descriptions of $3d$ gravity, such $T\overline{T}$-like deformations of the BF theory description of JT gravity (and its dual "particle-on-a-group" theory) have received less attention, which is one of the motives for this paper. Another motivation is that, just as Wilson lines are fundamental observables in the Chern-Simons description of $3d$ gravity, it is important to understand gravitational Wilson lines and their correlators in the BF theory, including their $T\overline{T}$-deformed versions.

To be more concrete, in this work we will consider deformations of two-dimensional gauge or gravity theories which are constructed in the following way. Begin with a three-dimensional bulk gravity theory which is dual to a $2d$ CFT. Deform the boundary CFT by the $T\overline{T}$ operator and interpret this deformation as a modification of the bulk gravity theory. Then dimensionally reduce this scenario on the circle to obtain a correspondence between a deformed $2d$ gravity theory and a dual one-dimensional theory. One can also rewrite the gravity theory in gauge theory variables and study the deformation of the $2d$ gauge theory. In the diagram (6), this corresponds to deforming the $2d$ WZW model in the top-right corner, and then studying the image of this deformation under the sequence of maps relating this theory to $2d$ JT gravity and $2d$ BF theory.

We emphasize that the image of this deformation is *not* the same as directly applying the $T\overline{T}$ deformation in the JT gravity or BF theory itself. Indeed, in the JT case it is not clear how to define a local stress tensor in a theory of gravity, and in the BF case the theory is topological so the stress tensor vanishes. We also note that, although we consider $T\overline{T}$-like deformations of two-dimensional AdS gravity theories, our procedure is quite different from defining the $T\overline{T}$ deformation for a $2d$ field theory on a fixed $AdS_2$ geometry. The latter problem has been considered in [85, 86]. Likewise, although the deformation of BF gauge theory treated in this manuscript is not the same as performing a $T\overline{T}$ deformation of a $2d$ gauge theory directly, such direct deformations of gauge theories have been considered for $2d$ Yang-Mills both with and without matter [87–90]. Instead, in the present work we study a deformation which is holographically dual to a $T\overline{T}$-like deformation of the boundary $(0 + 1)$-dimensional theory, rather than a $T\overline{T}$-deformation of the $2d$ gauge theory itself.

Furthermore, in this paper we focus on Wilson lines in $2d$ theories deformed by the image of the *double-trace* version of the $T\overline{T}$ operator (where image is meant in the sense described above). A conceptually related analysis involving deformed Wilson loops in the *single-trace* setting was presented in [91]. However, in that work, the Wilson line was computed for the DBI gauge field living on a D1-brane, rather than the gauge field arising from a gauge theory presentation of a gravity theory. In this paper, we content ourselves with the double-trace version of the $T\overline{T}$ deformation and Wilson lines for gauge fields associated with gravitational theories. One reason for doing this is that the bulk gravity dual to a single-trace $T\overline{T}$-deformed CFT also involves the dilaton and the 3-form flux $H_3$, and writing the kinetic terms for these fields in Chern-Simons variables is somewhat unwieldy. A second reason is that, although the gravity solution relevant for single-trace $T\overline{T}$ is locally $AdS_3$ in the deep interior, in the asymptotic region the solution approaches a linear dilaton spacetime. The linear dilaton region is qualitatively different from $AdS_3$ (in fact its causal structure is, in some sense, more similar

---

[6]The bad sign of $\lambda$ was studied by [84]. In the case of the bad sign of $\lambda$, one immediately encounters a complex-valued energy spectrum and partition function when $E > \frac{1}{8\lambda}$. To cure this problem and obtain a real-valued partition function, [84] included a non-perturbative contribution.

to that of Minkowski space), and therefore the Chern-Simons formulation of $3d$ gravity is not straightforwardly applicable in this regime.

The layout of this paper is as follows. Sections 2 and 3 are reviews of standard results about $3d$ gravitational Chern-Simons theory and $2d$ JT gravity, respectively, including the interpretation of a boundary $T\overline{T}$ deformation in both theories. In Section 4, we find the change in BF theory boundary conditions which corresponds to a $T\overline{T}$-like deformation of the dual $1d$ theory for two different choices of boundary conditions in the seed theory. In Section 5, we semi-classically compute corrections to the gravitational Wilson line due to a $T\overline{T}$ deformation of the boundary theory, including terms up to $O(\lambda^2 c^0)$. As a consistency check from the classical Wilson line correlators, we use conformal perturbation theory and match the results of [50, 51, 53] for $T\overline{T}$-deformed scalar correlators. In Section 6, we first study the deformed BF theory's boundary spectrum to find that the contribution from the principal series dominates the discrete series only below the Hagedorn temperature. The $T\overline{T}$-deformed Schwarzian theory description of the boundary spectrum is only valid below the Hagedorn temperature. We conclude the section by computing deformed Wilson lines and their correlators in the BF theory below the Hagedorn temperature. In Section 7, we conclude with a brief summary and discussions on possible extensions of the results presented in this paper. We list our conventions in Appendix A for $3d$ and JT gravity. Appendix B derives the non-perturbative deformed action for the $1d$ Schwarzian theory dual to $2d$ JT gravity for several background geometries – including the Euclidean AdS disk, Euclidean AdS double trumpet, and Lorentzian dS disk – under the $T\overline{T}$ deformation.

## 2   $T\overline{T}$ Deformations in $3d$ Chern-Simons Theory

In this section, we will review various facts about the presentation of three-dimensional AdS$_3$ gravity as a Chern-Simons theory. In particular, we recall that the bulk interpretation of a $T\overline{T}$ deformation in the boundary CFT is a change in the boundary conditions for the Chern-Simons gauge field, which was first pointed out in [77]. None of the content of this section is new; we include these results only to make the present work self-contained and to fix our notation.

### 2.1   $3d$ $SL(2,\mathbb{R})$ Gravitational Chern-Simons

The $(2+1)$-dimensional Einstein-Hilbert action in the first order formulation – thought of as either a classical theory or as a quantum mechanical theory described via perturbation theory in Newton's constant $G$ – is expressible as the difference between two $(2+1)$-dimensional Chern-Simons actions [7, 8]

$$S_{\text{EH}} = S_{\text{CS}}[A] - S_{\text{CS}}[\bar{A}], \tag{9}$$

where the Chern-Simons action for the connection $A$ is

$$S_{\text{CS}}[A] = \frac{1}{16\pi G} \int_{M_3} \text{Tr}\left(A \wedge dA + \frac{2}{3} A \wedge A \wedge A\right). \tag{10}$$

For the context of AdS$_3$ in Lorentzian signature, the connections $A$ and $\bar{A}$ are $\mathfrak{sl}(2,\mathbb{R})$-valued 1-forms. They encode geometric information, such as the vielbein $E^m = E^m_\mu dx^\mu$ and spin connection $\Omega^m = \frac{1}{2}\varepsilon^{mnl}\Omega_{\mu nl}dx^\mu$, respectively defined as[7]

$$A = (\Omega^m + E^m)L_m, \quad \bar{A} = (\Omega^m - E^m)\bar{L}_m. \tag{11}$$

---

[7]When writing indices for the three-dimensional bulk theory, we use Greek letters $\mu, \nu$ for curved (spacetime) indices and Latin letters $m, n$ for flat (tangent space) indices. For indices referring to the $2d$ boundary theory, we use middle Latin letters $i, j, k$ for curved indices and early Latin letters $a, b, c$ for flat indices.

The generators of the Lie algebra $\mathfrak{sl}(2,\mathbb{R})$ are denoted by $L_{0,\pm 1}$ and we work in the fundamental representation of $\mathfrak{sl}(2,\mathbb{R})$.[8]

The equations of motion for the Chern-Simons connections from (9) imply flatness

$$F = dA + A \wedge A = 0, \quad \bar{F} = d\bar{A} + \bar{A} \wedge \bar{A} = 0, \tag{12}$$

which are equivalent to the Einstein equations and the zero torsion condition, and are solved locally by $A = g^{-1} dg$ and $\bar{A} = \bar{g}^{-1} d\bar{g}$ for $g, \bar{g} \in \mathfrak{sl}(2,\mathbb{R})$.

By gauge fixing the radial component of the connections, one can find a class of solutions to (12) which we refer to as "Bañados-type solutions" [92]. These solutions parametrize the space of asymptotically AdS$_3$ with a trivial boundary metric:

$$
\begin{aligned}
A &= b(r)(d + a(z)) b(r), \quad b(r) = r^{L_0}, \\
\bar{A} &= \bar{b}(r)(d + \bar{a}(\bar{z})) \bar{b}(r)^{-1}, \quad \bar{b}(r) = r^{\bar{L}_0},
\end{aligned}
\tag{13}
$$

where $r$ is the holographic radial direction and the boundary connections are

$$
\begin{aligned}
a(z) &= \left( L_1 + \frac{6}{c} T_{zz}(z) L_{-1} \right) dz = \begin{pmatrix} 0 & \frac{6}{c} T_{zz}(z) \\ -1 & 0 \end{pmatrix} dz, \\
\bar{a}(\bar{z}) &= \left( \bar{L}_1 + \frac{6}{c} T_{\bar{z}\bar{z}}(\bar{z}) \bar{L}_{-1} \right) d\bar{z} = \begin{pmatrix} 0 & 1 \\ -\frac{6}{c} T_{\bar{z}\bar{z}}(\bar{z}) & 0 \end{pmatrix} d\bar{z}.
\end{aligned}
\tag{14}
$$

However, the Bañados-type boundary conditions (14) are not the most general solutions consistent with AdS$_3$ asymptotics. The most general asymptotically AdS$_3$ metric is described by a Fefferman-Graham expansion of the metric. It was shown in [77] that, in Chern-Simons variables, such an expansion corresponds to a solution where the connections $a$ and $\bar{a}$ take the more general form

$$
\begin{aligned}
a_i &= 2e_i^+ L_+ - f_i^- L_- + \omega_i L_0 = \frac{1}{2} \begin{pmatrix} \omega_i & -2f_i^- \\ -4e_i^+ & -\omega_i \end{pmatrix}, \\
\bar{a}_i &= f_i^+ L_+ - 2e_i^- L_- + \omega_i L_0 = \frac{1}{2} \begin{pmatrix} \omega_i & -4e_i^- \\ -2f_i^+ & -\omega_i \end{pmatrix}.
\end{aligned}
\tag{15}
$$

The connections (13) are solutions to the equations of motion when

$$
\begin{aligned}
da + a \wedge a &= 0, \\
d\bar{a} + \bar{a} \wedge \bar{a} &= 0.
\end{aligned}
\tag{16}
$$

Substituting (15) into (16), we find

$$
\begin{aligned}
d\omega - 2\varepsilon_{ab} e^a \wedge f^b &= 0, \\
de^a - \varepsilon^a{}_b e^b \wedge \omega &= 0, \\
df^a - \varepsilon^a{}_b f^b \wedge \omega &= 0, \\
e^a \wedge f_a &= 0,
\end{aligned}
\tag{17}
$$

which are the zero torsion conditions for the frame $e^a$ with spin connection $\omega$. Since by our conventions early Latin indices $a, b$ are flat while middle Latin indices $i, j$ are curved, $\varepsilon^{ab}$ is the Levi-Civita symbol with constant entries $\varepsilon_{+-} = -\varepsilon_{-+} = 1$, while $\varepsilon^{ij}$ is the Levi-Civita tensor with curved indices $\varepsilon^{x^+ x^-} = -\varepsilon^{x^- x^+} = \frac{1}{2e}$.

---

[8]The generators $L_0$, $L_1$ and $L_{-1}$ are defined in (A.2).

In the presence of a boundary, additional boundary terms are needed in the action in order to have a well-defined variational principle. Varying the Einstein-Hilbert action written in terms of the Chern-Simons connections (9) gives

$$
\begin{aligned}
\delta S_{\text{EH}} &= \delta S_{\text{CS}}[A] - \delta S_{\text{CS}}[\bar{A}] \\
&= \frac{1}{8\pi G} \int_{M_3} \text{Tr}\left(F \wedge \delta A - \bar{f} \wedge \delta \bar{A}\right) - \frac{1}{16\pi G} \int_{\partial M_3} \text{Tr}\left(A \wedge \delta A - \bar{A} \wedge \delta \bar{A}\right).
\end{aligned}
\tag{18}
$$

We desire a variational principle with Dirichlet boundary conditions for the metric, which corresponds to holding $e^a$ fixed at the boundary but letting $f^a$ vary. However, going on-shell by using the connections in (15), we find that (18) reduces to

$$
\delta S_{\text{CS}}[A] - \delta S_{\text{CS}}[\bar{A}] = -\frac{1}{8\pi G} \int_{\partial M_3} \varepsilon_{ab} \left(e^a \wedge \delta f^b - f^a \wedge \delta e^b\right),
\tag{19}
$$

which does not vanish and is inconsistent with the specified boundary conditions. We must therefore add the following boundary term to (18):

$$
S_{\text{bdry}} = -\frac{1}{8\pi G} \int_{\partial M_3} \varepsilon_{ab} \left(A^a \wedge A^b + \bar{A}^a \wedge \bar{A}^b\right).
\tag{20}
$$

The result now is consistent with Dirichlet boundary conditions, since

$$
\delta S_{\text{EH}} + \delta S_{\text{bdry}} = \frac{1}{4\pi G} \int_{\partial M_3} \varepsilon_{ab} f^a \wedge \delta e^b.
\tag{21}
$$

From the GKPW dictionary [3,4], it is understood that $e^a$ is the source and $f^a$ is the expectation value of the dual operator. In particular, the operator dual to the boundary vielbein is the stress tensor. By identifying

$$
\delta S = 4 \int_{\partial M_3} d^2 x \, (\det e) \, T^i_{\;a} \, \delta e^a_i,
\tag{22}
$$

we find that

$$
T^i_{\;a} = \frac{1}{4\pi G} \varepsilon_{ab} \varepsilon^{ij} f^b_j,
\tag{23}
$$

with $\nabla_{[i} f^a_{j]} = 0$.

When we turn to the two-dimensional BF theory in Section 4, it will be convenient to refer to the dimensional reduction of the 3d Chern-Simons action (10) on a circle.[9] The resulting dimensionally reduced theory is equivalent to BF theory with a particular choice of boundary term. To perform this reduction, we first write

$$
\begin{aligned}
8\pi G S_{\text{CS}} &= \frac{1}{2} \int_{M_3} \text{Tr}\left(A \wedge dA + \frac{2}{3} A \wedge A \wedge A\right) \\
&= \frac{1}{2} \int_{M_3} d^3 x \, \varepsilon^{\mu\nu\rho} \text{Tr}\left(A_\mu \partial_\nu A_\rho + \frac{2}{3} A_\mu A_\nu A_\rho\right) \\
&= \int_{M_3} \text{Tr}\left(A_\varphi F_{tr} + A_r \partial_\varphi A_t\right) + \frac{1}{2} \oint_{\partial M_3} \text{Tr} A_t^2,
\end{aligned}
\tag{24}
$$

___

[9]The 3d Chern-Simons theory can itself be viewed as the dimensional reduction of a 4d version of Chern-Simons. An interesting connection between 4d Chern-Simons and $T\bar{T}$ was discussed in [93].

where the base manifold $M_3$ is equipped with coordinates $x^\mu = \{r, t, \varphi\}$ and $r$ is the holographic coordinate for which $r \to 0$ is the location of the conformal boundary.

Next we impose the boundary condition $A_t = A_\varphi|_{\partial M_3}$ so that $\phi \equiv A_\varphi$ and $\partial_\varphi = 0$ (see [94]). Doing this yields

$$S_{\mathrm{BF}} = \int_{M_2} \mathrm{Tr}(\phi F) + \frac{1}{2} \oint_{\partial M_2} \mathrm{Tr}(\phi^2). \tag{25}$$

The first term is the usual action for $2d$ BF theory, which in this case has gauge group $G = SL(2, \mathbb{R})$. The degrees of freedom in this theory are a gauge field $A_\mu$ with field strength $F_{\mu\nu}$ along with an $SL(2, \mathbb{R})$-valued scalar field $\phi$. We will again consider this action in Section 3.1 where we will recall that the theory is equivalent to JT gravity. The second term of (25) controls the dynamics of a boundary degree of freedom which can be described either via the Schwarzian theory or the particle-on-a-group theory. We refer to this as a "Schwarzian-type" boundary term, which will be revisited in Section 4.2.

## 2.2 Interpretation of $T\overline{T}$ Deformation

The $3d$ gravitational Chern-Simons theory which we have just reviewed is dual to a conformal field theory on the $2d$ boundary of the spacetime via the usual AdS/CFT correspondence. On the other hand, in any two-dimensional field theory enjoying translation invariance, once can define a deformation by the double-trace $T\overline{T}$ operator. Our goal in the present section is to apply this deformation to the boundary CFT and interpret the resulting flow in terms of bulk Chern-Simons variables. We follow the discussion of [77] where this analysis first appeared.

We must first express the $T\overline{T}$ deformation in terms of the asymptotic expansion coefficients for the Chern-Simons gauge fields. We have already seen, for instance in (21) and (23), that the functions $e_i^a$ correspond to the boundary vielbein (or equivalently the metric) and that the $f_i^a$ are the dual expectation values which encode the stress tensor as

$$T^i{}_a = \frac{1}{4\pi G} \varepsilon_{ab} \varepsilon^{ij} f_j^b. \tag{26}$$

On the other hand, using the definition of the determinant in terms of the Levi-Civita symbol, the $T\overline{T}$ operator can be written as

$$T\overline{T} = -2\varepsilon^{ab} \varepsilon_{ij} T^i{}_a T^j{}_b. \tag{27}$$

In terms of the one-forms $f^-$ and $f^+$, one therefore has

$$T\overline{T} = \frac{1}{(4\pi G)^2} f^- \wedge f^+. \tag{28}$$

The flow equation for the boundary action can thus be written as

$$\frac{\partial S}{\partial \lambda} = \frac{1}{(4\pi G)^2} \int_{\partial M_3} f^- \wedge f^+. \tag{29}$$

We note that this is a flow equation for the *combined* boundary action, which in the undeformed case is a sum of three terms:

$$S(\lambda = 0) = S_{\mathrm{CS}}[A] - S_{\mathrm{CS}}[\overline{A}] + S_{\mathrm{bdry}}. \tag{30}$$

In Section 2.1, we saw that variation of the first two terms $S_{\mathrm{CS}}[A] - S_{\mathrm{CS}}[\overline{A}]$ generated a boundary variation of the form $\varepsilon_{ab}(e^a \wedge \delta f^b - f^a \wedge \delta e^b)$. The first term involving $\delta f^b$ was unsuitable for our desired variational principle, so we added $S_{\mathrm{bdry}}$ to cancel this variation.

We will make the ansatz that the finite-$\lambda$ deformed boundary action has the same structure as a sum of three terms involving sources $e_i^a(\lambda)$ and dual expectation values $f_i^a$. In this ansatz we allow the sources to acquire $\lambda$ dependence under the flow, but not the expectation values. As a result, the total boundary variation (21) of our $\lambda$-dependent ansatz takes the form

$$\delta S = \frac{1}{4\pi G} \int_{\partial M_3} \varepsilon_{ab} f^a \wedge \delta e^b(\lambda). \tag{31}$$

We now substitute this ansatz into the flow equation (29). More precisely, if the boundary action $S$ satisfies (29), then its variation satisfies

$$\frac{\partial(\delta S)}{\partial \lambda} = \frac{1}{(4\pi G)^2} \int_{\partial M_3} \delta(f^- \wedge f^+). \tag{32}$$

This then implies

$$\int_{\partial M_3} \varepsilon_{ab} f^a \wedge \delta\left(\frac{\partial e^b(\lambda)}{\partial \lambda}\right) = \frac{1}{4\pi G} \int_{\partial M_3} \varepsilon_{ab} f^a \wedge \delta f^b. \tag{33}$$

We see that (33) will be satisfied if

$$\frac{\partial e^b(\lambda)}{\partial \lambda} = \frac{1}{4\pi G} f^b. \tag{34}$$

Since $f^b$ is independent of $\lambda$ by assumption, this equation can be trivially integrated to find

$$e_i^a(\lambda) = e_i^a(0) + \frac{\lambda}{4\pi G} f_i^a, \tag{35}$$

and $f_i^a(\lambda) = f_i^a(0)$. One can show that, if the spin connection $\omega$ vanishes in the seed theory (as we will typically assume), then $\omega(\lambda) = 0$ along the flow as well. We have therefore characterized the full solution to the flow equation.

*Remarks on Deformed Boundary Conditions*

We now pause to make several comments on this interpretation. We see that the effect of a boundary $T\overline{T}$ deformation is to rotate our undeformed source $e_i^a$ into a new source $e_i^a(\lambda)$ which depends linearly on the corresponding undeformed expectation value. Since $e_i^a$ determines the boundary metric, this means that the deformed theory sees an effective stress-tensor-dependent metric. This is reminiscent of the result of $T\overline{T}$-deforming a two-dimensional field theory defined on a cylinder of radius $R$. As we will review around equation (163), in the zero-momentum sector, this deformation has the interpretation of placing the theory on a cylinder with an effective energy-dependent radius $\widetilde{R}(R, E_n)$.

Next, we note that although the sources $e_i^a$ have been modified, the variational principle defining our theory has not changed when expressed in terms of the new sources. The deformed boundary variation solving our flow is written as (31), which vanishes if the sources $e_i^a(\lambda)$ are held fixed. Therefore the theory described by these $T\overline{T}$-deformed boundary conditions still corresponds to a variational principle where the metric is held fixed at the boundary but the dual expectation value is free to fluctuate. All that has changed is the expression for this fixed metric in terms of the undeformed metric and stress tensor.

A third remark concerns the trace flow equation for the $T\overline{T}$ deformation. Because there is no dimensionful scale in a CFT, if one solves a $T\overline{T}$ flow beginning from a CFT seed then the resulting theory has a single scale set by $\lambda$. By noting that the derivative of the action with

respect to this single scale $\lambda$ is controlled by the trace of the stress tensor, while on the other hand the derivative of the action is related to the $T\overline{T}$ operator by the definition of the flow, one can derive the relation

$$T^{\mu}_{\ \mu}(\lambda) = -2\lambda T\overline{T}(\lambda),\tag{36}$$

along the flow. Because the modified boundary conditions (35) correspond to a $T\overline{T}$-deformation of a CFT, it is an instructive sanity check to verify explicitly that the trace flow equation (36) holds. And indeed, the trace of the deformed stress tensor with respect to the deformed metric is

$$
\begin{aligned}
T^{i}_{\ i} &= \eta^{ij} e^{a}_{i} T_{ja} \\
&= \frac{1}{4\pi G}\left(e^{a}_{i}(0) + \frac{\lambda}{4\pi G} f^{a}_{i}\right)\left(\varepsilon_{ab}\varepsilon^{ij} f^{b}_{j}\right) \\
&= \frac{\lambda}{(4\pi G)^2}\varepsilon_{ab}\varepsilon^{jk} f^{a}_{i} f^{b}_{j},
\end{aligned}\tag{37}
$$

where in the last step we have used that the undeformed stress tensor is traceless by assumption. On the other hand, at finite $\lambda$ the combination $T\overline{T}$ is given by (27):

$$
\begin{aligned}
T\overline{T} &= -2\varepsilon^{ab}\varepsilon_{ij} T^{i}_{\ a} T^{j}_{\ b} \\
&= \frac{-2}{(4\pi G)^2}\varepsilon^{ab}\varepsilon_{ij}\left(\varepsilon_{ac}\varepsilon^{ik} f^{c}_{k}\right)\left(\varepsilon_{bd}\varepsilon^{jn} f^{d}_{n}\right) \\
&= \frac{-2}{(4\pi G)^2}\varepsilon_{ab}\varepsilon^{jk} f^{a}_{i} f^{b}_{j},
\end{aligned}\tag{38}
$$

where we have repeatedly used the 2$d$ contracted epsilon identity $\varepsilon^{in}\varepsilon_{ij} = \delta^{n}_{\ j}$. Comparing (37) to (38), we see that the trace flow equation $T^{\mu}_{\ \mu}(\lambda) = -2\lambda T\overline{T}(\lambda)$ holds as expected.

We make a fourth and final comment, which is a trivial observation in this case but could conceivably be relevant for generalizations of the procedure described here. We emphasized around equation (30) that the undeformed action $S(\lambda = 0) = S_{\text{EH}} + S_{\text{bdry}}$ includes a boundary term which was added by hand in order to give a particular variational principle. Since the process of deforming the action by $T\overline{T}$ and the process of adding the boundary term $S_{\text{bdry}}$ are two distinct steps, there are naïvely two ways to proceed:

(I) First add the boundary term $S_{\text{bdry}}$ to get the total boundary action $S$. Then solve the flow equation (29) for this combined action.

(II) First solve the flow equation $\left.\frac{\partial S_{\text{EH}}}{\partial \lambda}\right|_{\text{bdry}} = \frac{1}{(4\pi G)^2}\int_{\partial M_3} f^{-} \wedge f^{+}$ which only deforms the first contribution to the action. Solve this by identifying new sources $e^{a}_{i}(\lambda)$. After doing this, add a new boundary term $S_{\text{bdry}}(\lambda)$ by hand to restore the desired variational principle.

In the discussion above we performed the deformation described by (I). However, it is straightforward to see that procedure (II) gives the same result precisely because the dual expectation values $f^{a}_{i}$ do not flow according to our ansatz. To show this we recall from (19) that

$$\left.\delta S_{\text{EH}}\right|_{\text{on-shell}} = -\frac{1}{8\pi G}\int_{\partial M_3}\varepsilon_{ab}\left(e^{a}\wedge\delta f^{b} - f^{a}\wedge\delta e^{b}\right).\tag{39}$$

Suppose that we had allowed both $e_i^a$ and $f_i^a$ to acquire $\lambda$ dependence along our flow. Then the derivative of this boundary variation would be

$$\frac{\partial(\delta S_{\text{EH}})}{\partial \lambda} = -\frac{1}{8\pi G} \int_{\partial M_3} \varepsilon_{ab} \left( \frac{\partial e^a(\lambda)}{\partial \lambda} \wedge \delta f^b(\lambda) + e^a(\lambda) \wedge \frac{\partial(\delta f^b(\lambda))}{\partial \lambda} - \frac{\partial f^a(\lambda)}{\partial \lambda} \wedge \delta e^b(\lambda) \right.$$
$$\left. - f^a(\lambda) \wedge \frac{\partial(\delta e^b(\lambda))}{\partial \lambda} \right). \tag{40}$$

In order to satisfy the flow equation $\frac{\partial(\delta S_{\text{EH}})}{\partial \lambda}\Big|_{\text{on-shell}} = \frac{1}{(4\pi G)^2} \int_{\partial M_3} \delta(f^- \wedge f^+)$, whose right side is again

$$\frac{1}{(4\pi G)^2} \int_{\partial M_3} \delta(f^- \wedge f^+) = \frac{1}{(4\pi G)^2} \int_{\partial M_3} \varepsilon_{ab} f^a \wedge \delta f^b, \tag{41}$$

we must have

$$\frac{\partial e^a(\lambda)}{\partial \lambda} \wedge \delta f^b(\lambda) + e^a(\lambda) \wedge \frac{\partial(\delta f^b(\lambda))}{\partial \lambda} - \frac{\partial f^a(\lambda)}{\partial \lambda} \wedge \delta e^b(\lambda) - f^a(\lambda) \wedge \frac{\partial(\delta e^b(\lambda))}{\partial \lambda} =$$
$$-\frac{1}{2\pi G} f^a \wedge \delta f^b. \tag{42}$$

The left side involves both $\delta e^a$ and $\delta f^a$ whereas the right side only involves $\delta f^a$. If these two variations are both independent, non-zero, and $\lambda$-dependent, it seems that we cannot have a solution. However, if we assume that $f^a$ and therefore $\delta f^a$ are independent of $\lambda$ as we did before, in addition to imposing that $\delta e^a(\lambda) = 0$ according to our choice of deformed variational principle, the equation reduces to

$$\frac{\partial e^a(\lambda)}{\partial \lambda} \wedge \delta f^b(\lambda) = -\frac{1}{2\pi G} f^a \wedge \delta f^b. \tag{43}$$

The solution to this simple flow is

$$e_i^a(\lambda) = e_i^a(0) - \frac{\lambda}{2\pi G} f_i^a. \tag{44}$$

Up to an overall rescaling of $\lambda$ by a factor of $-\frac{1}{2}$, this is the same solution as (35). This completes the first step of the alternate deformation procedure described in (II), but we must still add a new boundary term so that the combined boundary action is consistent with the variational principle $\delta e^a = 0$ that we have assumed. Our $\lambda$-dependent deformed boundary variation, before adding this boundary term, is simply

$$\delta S_{\text{EH}}(\lambda)\Big|_{\text{on-shell}} = -\frac{1}{8\pi G} \int_{\partial M_3} \varepsilon_{ab} \left( e^a(\lambda) \wedge \delta f^b - f^a \wedge \delta e^b(\lambda) \right). \tag{45}$$

But because this has exactly the same form as the variation (19) which we saw in the undeformed case, we may repeat the same procedure and add the term $S_{\text{bdry}}$ as defined in (20), except replacing $e_i^a$ with $e_i^a(\lambda)$ everywhere that it appears in the expansions of $A^a$ and $A^b$. The result is again

$$\delta S_{\text{EH}}(\lambda)\Big|_{\text{on-shell}} + \delta S_{\text{bdry}}(\lambda) = \frac{1}{4\pi G} \int_{\partial M_3} \varepsilon_{ab} f^a \wedge \delta e^b(\lambda), \tag{46}$$

exactly as we found before.

The upshot of this simple calculation is that the two processes described above – first adding a boundary term and then deforming, or first deforming and then adding a boundary term – commute in the calculation we consider here. However, in another setting where both the sources and expectation values become $\lambda$-dependent, performing the second deformation procedure (II) would produce a flow equation analogous to (42) which is not obviously equivalent to the flow of procedure (I). In such cases, one must choose a prescription in order to define the deformation.

## 3   $T\overline{T}$ Deformations in $2d$ JT Gravity

We now review features of JT gravity, and its BF gauge theory description, which are relevant to later sections when we study the $T\overline{T}$ deformation in BF theory. As in Section 2, none of the material in this discussion is new. For instance, the interpretation of a $T\overline{T}$-like deformation in the boundary dual to $2d$ JT gravity was considered in [82, 83] and we follow their discussion closely in Section 3.2. We include a reminder of these results here in order to facilitate comparison with the new results of Section 4, where we present an analogous interpretation of the $T\overline{T}$ deformation in BF variables.

### 3.1   JT gravity as a BF Gauge Theory

In the introduction of Section 1, we mentioned that one salient feature of $3d$ gravity motivating the present work is that it can be dimensionally reduced on a circle to yield JT gravity as described by the action (5). The goal of this subsection is to recall the standard statement that this $2d$ dilaton gravity theory can be equivalently written in gauge theory variables as a BF theory. Our treatment will follow [18].

One way of motivating this reformulation is to note that $3d$ gravity is equivalent to a Chern-Simons theory, as we reviewed in Section 2, and that the dimensional reduction of this $3d$ Chern-Simons theory is a BF gauge theory. Indeed, we saw this reduction explicitly around equation (25). These observations are summarized by the sub-diagram formed by the second and third columns of (6):

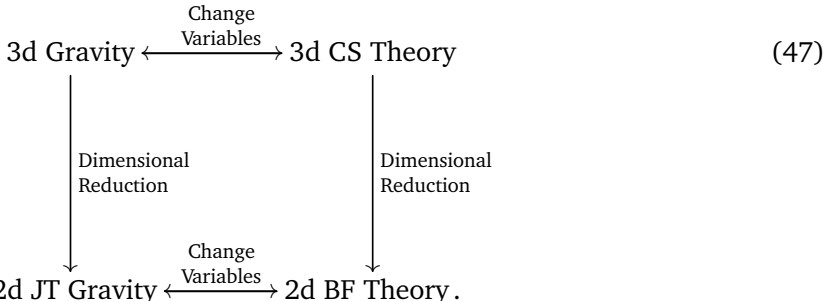

$$\tag{47}$$

We have now reviewed all of the arrows in (47) except for the change of variables linking the two theories in the bottom row. Although it is clear that such a change of variables must exist by consistency of the diagram, it is instructive to spell out the map explicitly.

Recall that the BF theory in Euclidean signature is described by the action

$$I_{\text{BF}} = -i \int_{M_2} \text{Tr}(\phi F) \,, \tag{48}$$

where $\phi$ is a scalar field and $F$ is the field strength of the gauge field $A_\mu$. At the moment we will only be concerned with the bulk equations of motion and therefore will not include

any additional boundary term like the one which appeared in (25). The equations of motion arising from (48) are

$$\phi: \quad F = 0,$$
$$A_\mu: \quad D_\mu \phi = \partial_\mu \phi - [A_\mu, \phi] = 0. \tag{49}$$

On the other hand, beginning from the action (5) of JT gravity and setting $\Lambda = -1$, one finds the equations of motion

$$\Phi: \quad R = -2,$$
$$g_{\mu\nu}: \quad \nabla_\mu \nabla_\nu \Phi = g_{\mu\nu} \Phi. \tag{50}$$

Next we will argue that the JT equations of motion in (50) are equivalent to the BF equations of motion in (49). To do this, we first expand the BF fields in terms of generators defined in Appendix A.2:

$$A(x) = e^2(x)P_2 + e^1(x)P_1 + \omega(x)P_0, \quad \phi(x) = \phi^1(x)P_1 + \phi^2(x)P_2 + \phi^0(x)P_0, \tag{51}$$

where

$$P_0 = \begin{pmatrix} 0 & \frac{1}{2} \\ -\frac{1}{2} & 0 \end{pmatrix}, \quad P_1 = \begin{pmatrix} 0 & \frac{1}{2} \\ \frac{1}{2} & 0 \end{pmatrix}, \quad P_2 = \begin{pmatrix} \frac{1}{2} & 0 \\ 0 & -\frac{1}{2} \end{pmatrix}. \tag{52}$$

Written in differential form notation, the equation of motion for $A_\mu$ in equation (49) becomes $d\phi - A \wedge \phi = 0$. The exterior derivative of the scalar $\phi$ is

$$d\phi = d\phi^1(x)P_1 + d\phi^2(x)P_2 + d\phi^0(x)P_0. \tag{53}$$

Meanwhile, a short calculation gives

$$A \wedge \phi = \left(e^2 \wedge \phi^1 - e^1 \wedge \phi^2\right)P_0 + \left(e^2 \wedge \phi^0 - \omega \wedge \phi^2\right)P_1 + \left(\omega \wedge \phi^1 - e^1 \wedge \phi_0\right)P_2. \tag{54}$$

Putting everything together, we find

$$d\phi^0(x) = e^2(x) \wedge \phi^1(x) - e^1(x) \wedge \phi^2(x),$$
$$d\phi^1(x) = e^2(x) \wedge \phi^0(x) - \omega(x) \wedge \phi^2(x),$$
$$d\phi^2(x) = \omega(x) \wedge \phi^1(x) - e^1(x) \wedge \phi^0(x). \tag{55}$$

We now act with the covariant derivative $\nabla_\mu$ on the equation for $d\phi^0$ in (55). At the risk of being pedantic, we pause to clarify one point of possible confusion. When acting on a generalized tensor with both curved (spacetime) indices and flat (tangent space) indices, the action of the covariant derivative $\nabla_\mu$ involves Christoffel symbol terms associated with the curved indices and spin connection terms associated with the flat indices. For instance, on the vielbein $e_\nu^a$ with one curved and one flat index, one has

$$\nabla_\mu e_\nu^a = \partial_\mu e_\nu^a + \omega_{\mu}{}^a{}_b e_\nu^b - \Gamma^\sigma{}_{\nu\mu} e_\sigma^a. \tag{56}$$

Since the covariant derivative annihilates the vielbein by the zero-torsion constraint $\tau^a = de^a + \omega_b^a \wedge e^b = 0$, the combination (56) vanishes.

However, the equations (55) are covariant with respect to their single curved index but not with respect to the implicit flat index on the vielbeins. It is easiest to see this by writing the equations in components. For instance, the $\phi^0$ equation is

$$\partial_\mu \phi^0 = \phi^1 e_\mu^2 - \phi^2 e_\mu^1. \tag{57}$$

Although this equation has a free $\mu$ index, there is no free $a$ index in the $e_\mu^a$ factors. Indeed, this equation could never have been covariant with respect to such a tangent space index since

the quantity $\partial_\mu \phi^0$ on the left has no flat indices. Therefore, when we act with the covariant derivative, there will be no spin connection terms introduced in the derivatives of vielbein factors. One has

$$\nabla_\mu e_\nu^2 = \partial_\mu e_\nu^2 - \Gamma^\sigma{}_{\nu\mu} e_\sigma^2, \tag{58}$$

and likewise for $\nabla_\nu e_\mu^1$. But since $\nabla_\mu e_\nu^a = 0$, equation (56) implies

$$\partial_\mu e_\nu^2 - \Gamma^\sigma{}_{\nu\mu} e_\sigma^2 = -\omega_\mu{}^2{}_b e_\nu^b, \tag{59}$$

and again a similar equation for $\nabla_\nu e_\mu^1$. Using $\omega^1{}_2 = -\omega^2{}_1 = \omega$, we find

$$\begin{aligned}
\nabla_\mu e_\nu^1 &= -\omega_\mu{}^1{}_b e_\nu^b = -\omega_\mu e_\nu^2, \\
\nabla_\mu e_\nu^2 &= -\omega_\mu{}^2{}_b e_\nu^b = \omega_\mu e_\nu^1.
\end{aligned} \tag{60}$$

Now we are prepared to act with the covariant derivative on the $\phi^0$ equation of motion. On the left, the result is a two-tensor with components $\nabla_\mu \nabla_\nu \phi^0$. One finds

$$\begin{aligned}
\nabla_\mu \nabla_\nu \phi^0 &= \left(\partial_\mu \phi^1\right) e_\nu^2 - \left(\partial_\mu \phi^2\right) e_\nu^1 + \phi^1 \left(\nabla_\mu e_\nu^2\right) - \phi^2 \left(\nabla_\mu e_\nu^1\right) \\
&= \left(\partial_\mu \phi^1\right) e_\nu^2 - \left(\partial_\mu \phi^2\right) e_\nu^1 + \phi^1 \omega_\mu e_\nu^1 + \phi^2 \omega_\mu e_\nu^2.
\end{aligned} \tag{61}$$

On the other hand, writing the second and third equations of (55) in components gives $\partial_\mu \phi^1 = \phi^0 e_\mu^2 - \phi^2 \omega_\mu$ and $\partial_\mu \phi^2 = \phi^1 \omega_\mu - \phi^0 e_\mu^1$. Substituting these into (61) gives

$$\begin{aligned}
\nabla_\mu \nabla_\nu \phi^0 &= \left(\phi^0 e_\mu^2 - \phi^2 \omega_\mu\right) e_\nu^2 - \left(\phi^1 \omega_\mu - \phi^0 e_\mu^1\right) e_\nu^1 + \phi^1 \omega_\mu e_\nu^1 + \phi^2 \omega_\mu e_\nu^2 \\
&= \left(e_\mu^1 e_\nu^1 + e_\mu^2 e_\nu^2\right) \phi^0.
\end{aligned} \tag{62}$$

If we identify the metric as $g_{\mu\nu} = e_\mu^1 e_\nu^1 + e_\mu^2 e_\nu^2$ and assume that the JT dilaton $\Phi$ is proportional to the BF field $\phi^0$, then this is exactly the metric equation of motion in (50):

$$\nabla_\mu \nabla_\nu \Phi = g_{\mu\nu} \Phi. \tag{63}$$

We therefore have demonstrated that the JT gravity equations of motion (50) are recovered from the BF equations of motion in (49) after making the change of variables

$$\phi^0 = \frac{i}{4\pi G} \Phi, \qquad g_{\mu\nu} = e_\mu^1 e_\nu^1 + e_\mu^2 e_\nu^2. \tag{64}$$

Here the choice of the proportionality factor $\frac{i}{4\pi G}$ between $\phi^0$ and $\Phi$ is required by our normalizations for the BF and JT actions in (48) and (5), respectively. Under this identification, we see that the expansion coefficients $e^a$ appearing in the BF gauge field $A_\mu$ are interpreted as the frame fields in the JT gravity theory, whereas the field $\omega$ defines the spin connection, which satisfies $d\omega = \frac{R}{2} e^1 \wedge e^2$ for a $2d$ manifold. In this correspondence, the $\phi^1, \phi^2$ equations of motion are mapped onto the torsionless conditions $\tau^a = de^a + \omega^a_b \wedge e^b = 0$. This completes our review of the final arrow on the bottom row of (47) linking JT gravity with BF gauge theory.

Next, we explain the boundary conditions and the choice of boundary term for the BF gauge theory which recovers the Schwarzian action. Variation of the BF action on-shell yields the boundary action

$$\delta I_{\text{BF}} = -i \int_{\partial M_2} d\tau \, \text{Tr}\left(\phi \, \delta A_\tau\right), \tag{65}$$

with $\tau$ parametrizing the one-dimensional boundary $\partial M_2$.

Thus the variation (65) of the BF action vanishes if $A_\tau$ is held fixed on $\partial M_2$. In fact, from JT gravity's first-order formulation, the spin connection and frame are already fixed so no boundary term is required to have a well-defined variational principle. Unfortunately, this means that the BF theory cannot be holographcally dual to the Schwarzian because the theory is topologically trivial. In particular, the observables of the theory would depend on the holonomy around the boundary rather than depending on the local value of $A_\tau$. To recover the Schwarzian dynamics, one includes a string defect $I_{\text{string}}$ around a loop $L \subset M_2$, which yields the modified action

$$I = -i \int_{M_2} \text{Tr}(\phi F) - \oint_0^\beta du\, V(\phi), \quad V(\phi) = \frac{\nu}{4} \text{Tr}\,\phi^2. \tag{66}$$

The second term in (66) is the string defect with coupling $\nu$, and $u$ is the proper length parametrization of the loop with circumference $\beta$. This form of $V(\phi)$ is consistent with the boundary term in (25) which we expect from the dimensional reduction of Chern-Simons and, as we will see, correctly recovers the Schwarzian action.

The overall action (66) preserves the defect diffeomorphisms, and the degrees of freedom from the string defect are realized by the Schwarzian theory as [18] showed by evaluating the action (66) using the solution to the equation of motion (49) for $\phi(u)$ along $L$. To see the derivation more explicitly, we parametrize the boundary fields $\phi$ and $A_\tau$ by[10]

$$A_\tau = \omega \ell_0 + e_+ \ell_+ + e_- \ell_-, \quad \phi = \phi_+ \ell_+ + \phi_- \ell_- + \phi_0 \ell_0, \tag{67}$$

where

$$\ell_0 = iP_0, \quad \ell_+ = -iP_1 - P_2, \quad \ell_- = -iP_1 + P_2,$$
$$\omega = -i\omega_\tau\Big|_{\partial M_2}, \quad e_+ = \frac{ie_\tau^1 - e_\tau^2}{2}\Big|_{\partial M_2}, \quad e_- = \frac{ie_\tau^1 + e_\tau^2}{2}\Big|_{\partial M_2}. \tag{68}$$

We compute the commutator

$$[A_\tau, \phi] = (e_+ \phi_0 - \omega \phi_+)\ell_+ + (\omega \phi_- - e_- \phi_0)\ell_- + 2(e_+ \phi_- - e_- \phi_+)\ell_0, \tag{69}$$

to write the complete set of equations of motion $D_\tau \phi = 0$ at the loop

$$\begin{aligned}
\ell_0: \quad &\partial_\tau \phi_0 = 2(e_+ \phi_- - e_- \phi_+), \\
\ell_-: \quad &\partial_\tau \phi_- = \omega \phi_- - e_- \phi_0, \\
\ell_+: \quad &\partial_\tau \phi_+ = e_+ \phi_0 - \omega \phi_+.
\end{aligned} \tag{70}$$

To solve the equations at the loop (70), we perform the same change of variables as [18]

$$\phi_-(\tau) = \frac{2e_-}{\partial_\tau u(\tau)} \implies \phi_-(u) = 2e_- \tau', \tag{71}$$

where

$$\partial_\tau \phi_i = (\partial_\tau u)(\partial_u \phi_i) = \frac{\partial_u \phi_i}{\tau'}. \tag{72}$$

Substituting the above into the equation of motion for the $\ell_-$ component, we find that

$$\phi_0 = \frac{-\partial_\tau \phi_- + \omega \phi_-}{e_-} = -\frac{2\tau''}{\tau'} + 2\omega\tau'. \tag{73}$$

---

[10]Note that a different representation of the generators is used when solving the equations of motion for the field $\phi$ at the loop. See (A.14) for the definitions of $\ell_0$, $\ell_+$ and $\ell_-$.

Then, solving for the $\ell_0$ component, one uses $\phi_-$ and $\phi_0$ to find

$$
\begin{aligned}
\phi_+ &= \frac{1}{e_-}\left(-\frac{1}{2}\partial_\tau \phi_0 + e_+ \phi_-\right) \\
&= 2\left(e_+ \tau' + \frac{\tau'''}{2e_-\tau'^2} - \frac{\omega\tau''}{2e_-\tau'} - \frac{\tau''^2}{2e_-\tau'^3}\right).
\end{aligned}
\tag{74}
$$

We found all the components for the field

$$
\nu\phi(u) = 2e_-\tau'\ell_- + 2\left(\omega\tau' - \frac{\tau''}{\tau'}\right)\ell_0 + 2\left(e_+\tau' + \frac{\tau'''}{2e_-\tau'^2} - \frac{\omega\tau''}{2e_-\tau'} - \frac{\tau''^2}{2e_-\tau'^3}\right)\ell_+.
\tag{75}
$$

Here $\tau(u)$ is further constrained by the $\ell_+$ component of the equation $D_u\phi = 0$, which gives

$$
4(\det A_\tau)\,\tau'^4\tau'' + 3\tau''^3 - 4\tau'\tau''\tau''' + \tau'^2\tau'''' = 0,
\tag{76}
$$

where $\tau(u)$ is monotonic so $\tau'(u) \neq 0$ and $\det A_\tau = e_-e_+ - \frac{\omega^2}{4}$.

Now we are ready to evaluate the string defect action by computing $\mathrm{Tr}\,\phi^2$. This computation is straightforward as

$$
\phi^2 = \frac{1}{\nu^2}\begin{pmatrix} -4e_-e_+\tau'^2 + \omega^2\tau'^2 + \frac{3\tau''^2}{\tau'^2} - \frac{2\tau'''}{\tau'} & 0 \\ 0 & -4e_-e_+\tau'^2 + \omega^2\tau'^2 + \frac{3\tau''^2}{\tau'^2} - \frac{2\tau'''}{\tau'} \end{pmatrix}
\tag{77}
$$

and

$$
\begin{aligned}
V(\phi) &= \frac{\nu}{4}\mathrm{Tr}\,\phi^2 \\
&= \frac{1}{\nu}\left(\{\tau(u),u\} + 2\tau'(u)^2\det A_\tau\right) \\
&= \frac{1}{\nu}\left\{\tan\left(\sqrt{(\det A_\tau)\,\tau(u)}\right),u\right\}.
\end{aligned}
\tag{78}
$$

As expected, we have recovered the Schwarzian action[11] from including the string defect in the BF action (66), which gives

$$
I = -\frac{1}{\nu}\int_0^\beta du\left\{\tan\left(\sqrt{(\det A_\tau)\,\tau(u)}\right),u\right\}.
\tag{79}
$$

## 3.2 Interpretation of $T\overline{T}$ Deformation

Before we begin with the $T\overline{T}$ deformation in JT gravity, we first recall how a general class of related deformations are defined and their meaning in AdS/CFT. Following [83], we deform a seed action $I(0)$ via a generic operator $M_\lambda$ as

$$
I(\lambda) = I(0) + \int d\tau\,\sqrt{\gamma}\,M_\lambda(T_{\tau\tau},\gamma^{\tau\tau}),
\tag{80}
$$

where the variational principle in the undeformed theory (where $M_0 = 0$) is defined by

$$
\delta I(0) = \frac{1}{2}\int d\tau\,\sqrt{\gamma}\,T_{\tau\tau}\delta\gamma^{\tau\tau}.
\tag{81}
$$

---

[11]One can show that this derivation of the Schwarzian theory holds for any $\Lambda$ by using the more general parameterization $A_\tau = \omega\ell_0 + \sqrt{\Lambda}e_+\ell_+ + \sqrt{\Lambda}e_-\ell_-$. Equivalently, this corresponds to replacing the determinant in (79) by $\det A_\tau = \Lambda e_-e_+ - \frac{\omega^2}{4}$.

With the deformation (80), one finds the following variation:

$$\delta I(\lambda) = \delta I(0) + \int d\tau \left[ \delta\left(\sqrt{\gamma}\right) M_\lambda + \sqrt{\gamma}\delta M_\lambda \right].\tag{82}$$

Using the facts that

$$\delta M_\lambda = \frac{\partial M_\lambda}{\partial T_{\tau\tau}}\delta T_{\tau\tau} + \frac{\partial M_\lambda}{\partial \gamma^{\tau\tau}}\delta\gamma^{\tau\tau}\tag{83}$$

and

$$\delta\left(\sqrt{\gamma}\right) = -\frac{\sqrt{\gamma}}{2\gamma^{\tau\tau}}\delta\gamma^{\tau\tau},\tag{84}$$

we find (82) is written as

$$\delta I(\lambda) = \frac{1}{2}\int d\tau \sqrt{\gamma}\left[\left(T_{\tau\tau} - \frac{M_\lambda}{\gamma^{\tau\tau}} + 2\frac{\partial M_\lambda}{\partial\gamma^{\tau\tau}}\right)\delta\gamma^{\tau\tau} + 2\frac{\partial M_\lambda}{\partial T_{\tau\tau}}\delta T_{\tau\tau}\right].\tag{85}$$

We wish to identify sources and expectation values by demanding that we can rewrite (85) in terms of $\lambda$-dependent quantities $\widetilde{T}_{\tau\tau}$, $\widetilde{\gamma}_{\tau\tau}$ as

$$\begin{aligned}\delta I(\lambda) &= \frac{1}{2}\int d\tau \frac{\widetilde{T}_{\tau\tau}}{\sqrt{\widetilde{\gamma}^{\tau\tau}}}\delta\widetilde{\gamma}^{\tau\tau}\\ &= \frac{1}{2}\int d\tau \frac{\widetilde{T}_{\tau\tau}}{\sqrt{\widetilde{\gamma}^{\tau\tau}}}\left(\frac{\partial\widetilde{\gamma}^{\tau\tau}}{\partial T_{\tau\tau}}\delta T_{\tau\tau} + \frac{\partial\widetilde{\gamma}^{\tau\tau}}{\partial\gamma^{\tau\tau}}\delta\gamma^{\tau\tau}\right).\end{aligned}\tag{86}$$

Here the operator $\widetilde{T}_{\tau\tau}$ is sourced by $\widetilde{\gamma}^{\tau\tau}$. In other words, the deformation changes the variational principle from one where $\gamma^{\tau\tau}$ is held fixed to one where $\widetilde{\gamma}^{\tau\tau}$ is fixed.

Comparing (85) and (86), we find the following coupled PDEs for the deformed boundary stress tensor and metric:

$$\begin{aligned}\frac{\widetilde{T}_{\tau\tau}}{\sqrt{\widetilde{\gamma}^{\tau\tau}}}\frac{\partial\widetilde{\gamma}^{\tau\tau}}{\partial T_{\tau\tau}} &= 2\sqrt{\gamma}\frac{\partial M_\lambda}{\partial T_{\tau\tau}},\\ \frac{\widetilde{T}_{\tau\tau}}{\sqrt{\widetilde{\gamma}^{\tau\tau}}}\frac{\partial\widetilde{\gamma}^{\tau\tau}}{\partial\gamma^{\tau\tau}} &= \sqrt{\gamma}\left(T_{\tau\tau} - \frac{M_\lambda}{\gamma^{\tau\tau}} + 2\frac{\partial M_\lambda}{\partial\gamma^{\tau\tau}}\right),\end{aligned}\tag{87}$$

with the initial conditions $\widetilde{T}_{\tau\tau}(\lambda = 0) = T_{\tau\tau}$ and $\widetilde{\gamma}^{\tau\tau}(\lambda = 0) = \gamma^{\tau\tau}$.

To further illustrate, we focus on a specific class of deformations that only depend on the trace of the stress tensor $T_{\tau\tau}\gamma^{\tau\tau}$. It is convenient to express our ansatz in terms of the dimensionless combination

$$X = \lambda T_{\tau\tau}\gamma^{\tau\tau}.\tag{88}$$

We assume

$$\widetilde{T}_{\tau\tau} = T_{\tau\tau}\xi(X), \quad \widetilde{\gamma}^{\tau\tau} = \gamma^{\tau\tau}\chi(X),\tag{89}$$

where $\xi(0) = \chi(0) = 1$ so that we recover the undeformed stress tensor and metric as $\lambda \to 0$. On the other hand, by dimensional analysis we can write the function $M_\lambda(\lambda, T_{\tau\tau}, \gamma^{\tau\tau})$ in the form

$$M_\lambda = \frac{1}{\lambda}m_\lambda(X).\tag{90}$$

By substituting (89)-(90) into the system of coupled PDEs (87), we find the pair of equations

$$\chi'(X) = \frac{2\sqrt{\chi(X)}m_\lambda'(X)}{X\xi(X)},$$

$$\sqrt{\chi(X)}\big(X - m_\lambda'(X) + 2Xm_\lambda'(X)\big) = X\xi(X)\big(\chi(X) + X\chi'(X)\big). \tag{91}$$

The usual double-trace $T\overline{T}$ deformation is quadratic in stress tensors, so one might be interested in studying a deformation which is proportional to the combination $X^2 = (\lambda T_{\tau\tau}\gamma^{\tau\tau})^2$, since this is the only dimensionless and reparameterization-invariant stress tensor bilinear in $(0+1)$-dimensions. This corresponds to a deformation of the form

$$M_\lambda = \frac{1}{\lambda}X^2$$
$$= \lambda(T_{\tau\tau}\gamma^{\tau\tau})^2. \tag{92}$$

Using the form (92) of the deformation, the equations (91) become

$$\xi(X) = \frac{(3X+1)\sqrt{\chi(X)}}{\chi(X) + X\chi'(X)}, \qquad \chi'(X) = \frac{4\sqrt{\chi(X)}}{\xi(X)}, \tag{93}$$

which have the solutions

$$\chi(X) = \frac{1}{(1-X)^4}, \qquad \xi(X) = (1-X)^3. \tag{94}$$

We have therefore found that, for the form of the deformation $M_\lambda = \lambda(T_{\tau\tau}\gamma^{\tau\tau})^2$ motivated by the usual $T\overline{T}$ deformation,[12] the solution is

$$\widetilde{T}_{\tau\tau}(\lambda) = T_{\tau\tau}(1 - \lambda T_{\tau\tau}\gamma^{\tau\tau})^3, \quad \widetilde{\gamma}^{\tau\tau}(\lambda) = \frac{\gamma^{\tau\tau}}{(1 - \lambda T_{\tau\tau}\gamma^{\tau\tau})^4}. \tag{96}$$

However, as mentioned in Section 1, and derived in Appendix A of [82], despite (92) being proportional to a double-trace operator $T^2$, it is not suitable as a $T\overline{T}$ deformation for JT gravity with a Dirichlet cutoff. The following choice of operator is suitable for the $T\overline{T}$ deformation [82] as the correct deformed energy spectrum is recovered:

$$M_\lambda = -2\lambda O T_{\tau\tau}\gamma^{\tau\tau}, \tag{97}$$

where the operator $O$ is sourced by the boundary dilaton $\Phi_b$ as

$$O = \frac{1}{\sqrt{\gamma}}\frac{\delta I}{\delta\Phi_b}. \tag{98}$$

The seed theory action is now deformed as

$$I(\lambda) = I(0) + \int d\tau\sqrt{\gamma}M_\lambda\big(T_{\tau\tau}, \gamma^{\tau\tau}, O, \Phi_b\big), \tag{99}$$

where the variation of the undeformed theory is

$$\delta I(0) = \int d\tau\sqrt{\gamma}\left(\frac{1}{2}T_{\tau\tau}\delta\gamma^{\tau\tau} + O\delta\Phi_b\right). \tag{100}$$

---

[12]For a multi-trace deformation $M_\lambda^{(n)} = \lambda_n(T_{\tau\tau}\gamma^{\tau\tau})^{2n}$ with coupling $\lambda_n$, one finds via solving (91)

$$\widetilde{T}_{\tau\tau}(\lambda_n) = T_{\tau\tau}\big(1 - \lambda_n(T_{\tau\tau}\gamma^{\tau\tau})^{2n-1}\big)^{\frac{4n-1}{2n-1}}, \quad \widetilde{\gamma}^{\tau\tau}(\lambda_n) = \gamma^{\tau\tau}\big(1 - \lambda_n(T_{\tau\tau}\gamma^{\tau\tau})^{2n-1}\big)^{\frac{4n}{1-2n}}. \tag{95}$$

In order to identify the variational principle of the deformed theory, we demand that $\delta I(\lambda)$ can be written in terms of $\lambda$-dependent sources and expectation values as

$$\delta I(\lambda) = \int d\tau \sqrt{\gamma} \left( \frac{1}{2} T_{\tau\tau}(\lambda) \delta \gamma^{\tau\tau}(\lambda) + O(\lambda) \delta \Phi_b(\lambda) \right). \tag{101}$$

Following the same procedure as in the previous example with $M_\lambda(T_{\tau\tau}, \gamma^{\tau\tau})$, we find the sources and expectation values transform as

$$\gamma_{\tau\tau}(\lambda) = \gamma_{\tau\tau}(0)(1 + 2\lambda O(0))^2, \quad T_{\tau\tau}(\lambda) = T_{\tau\tau}(0)(1 + 2\lambda O(0))^2,$$
$$\Phi_b(\lambda) = \Phi_b(0) - 2\lambda T(0), \quad O(\lambda) = \frac{O(0)}{1 + 2\lambda O(0)}, \tag{102}$$

which satisfy

$$\delta I(\lambda) = \delta I(0) - 2\lambda \delta \left( \int d\tau \sqrt{\gamma} O(0) T_{\tau\tau}(0) \gamma^{\tau\tau}(0) \right). \tag{103}$$

The $\lambda$-dependent sources and expectation values (102) describe the full solution for the bulk JT gravity fields which corresponds to performing a $T\overline{T}$-like deformation of the $1d$ boundary theory. As in the analogous deformation of $3d$ gravitational Chern-Simons reviewed in Section 2.2, we note that the result can be intereteted as a linear mixing of sources and expectation values, although in this case each source becomes a function of the dual expectation value for a *different* operator – for instance, the metric becomes dependent on the field $O$ which is dual to the dilaton $\Phi_b$.

# 4 $T\overline{T}$ Deformed Boundary Conditions in BF Theory

In the previous section, we have seen that the interpretation of a boundary $T\overline{T}$ deformation in JT gravity is a particular $\lambda$-dependent mixing (102) of the metric $\gamma_{\tau\tau}$, dilaton $\Phi_b$, and their dual operators. Since JT gravity can also be written in BF variables, there must be an analogous interpretation of the boundary $T\overline{T}$ deformation in this language. The goal of the current section is to make this BF interpretation explicit.

Because BF gauge theory is topological, all of the dynamics of the theory occur at the boundary. As a result, the choice of boundary term – and therefore, the variational principle – is an important input for defining the theory. We consider $T\overline{T}$-type deformations for two choices of boundary terms: one which gives a variational principle analogous to that of the JT gravity theory, and one whose boundary theory is the Schwarzian.

## 4.1 Deformation with JT-type Boundary Term

First we will determine the choice of boundary term in BF theory which gives a variational principle most analogous to that of the JT gravity theory. We saw in (100) that the on-shell variation of the JT gravity action takes the form

$$\delta I \Big|_{\text{on-shell}} = \int d\tau \sqrt{\gamma} \left( \frac{1}{2} T_{\tau\tau} \delta \gamma^{\tau\tau} + O \delta \Phi_b \right). \tag{104}$$

This boundary term vanishes if we fix the value of the (inverse) metric $\gamma^{\tau\tau}$ and the dilaton $\Phi_b$ at the boundary. The operators dual to the metric and dilaton are then the boundary stress tensor $T_{\tau\tau}$ and the operator $O$, respectively.

On the other hand, the variation of the BF action $I_{\text{BF}} = -i \int_{M_2} \text{Tr}(\phi F)$ was given in (65) as

$$\delta I_{\text{BF}}\Big|_{\text{on-shell}} = -i \int_{\partial M_2} d\tau \, \text{Tr}(\phi \, \delta A_\tau) \,. \tag{105}$$

We parameterize the BF theory fields in terms of $SL(2,\mathbb{R})$ generators as

$$A_\mu(x) = e_\mu^+(x) L_+ + e_\mu^-(x) L_- + \omega_\mu(x) L_0 \,, \qquad \phi(x) = \phi^+(x) L_+ + \phi^-(x) L_- + \phi^0(x) L_0 \,, \tag{106}$$

following the conventions in (A.2). Note that we use the notation $L_+$ for $L_1$ and $L_-$ for $L_{-1}$. In terms of the functions appearing in this expansion, the boundary term (105) is

$$\delta I_{\text{BF}}\Big|_{\text{on-shell}} = -i \int d\tau \left( \frac{1}{2} \phi^0 \delta \omega_\tau - \phi^+ \delta e_\tau^- - \phi^- \delta e_\tau^+ \right) . \tag{107}$$

The asymptotic values of the expansion coefficients $e_\tau^\pm$ in the BF fields are interpreted as the einbein for the one-dimensional boundary theory. These fields are therefore the BF analogue of the boundary metric $\gamma_{\tau\tau}$. Likewise, the boundary value of the BF variable $\phi^0$ is proportional to the boundary dilaton $\Phi_b$ in JT variables.

Thus we see that the naïve BF action, without any added boundary term, corresponds to a different variational principle than that of JT gravity. In order for the variation (107) to vanish, we must fix the boundary values of $e_\tau^\pm$ (which corresponds to fixing the boundary metric) but *not* the boundary value of $\phi^0$; rather the asymptotic value of $\omega_\tau$ is held fixed. In JT gravity language, this corresponds to a variational principle where the value of the dual operator $O$ is held fixed but the boundary dilaton $\Phi_b$ is free to vary.

We can, of course, modify the BF variational principle by adding an appropriate boundary term. Suppose that we choose the BF action to be

$$I = I_{\text{BF}} + I_{\text{bdry}} \,, \qquad I_{\text{bdry}} = \frac{i}{2} \int d^2x \, \sqrt{g} \, n_\mu \partial^\mu \left( \phi^0 \omega_\tau \right) , \tag{108}$$

where $n_\mu$ is a unit normal vector in the radial direction. The corresponding contribution to the boundary variation is

$$\delta I_{\text{bdry}} = \frac{i}{2} \int d\tau \left( \omega_\tau \delta \phi^0 + \phi^0 \delta \omega_\tau \right) . \tag{109}$$

This cancels the $\phi^0 \omega_\tau$ term appearing in (105). The total boundary variation is now

$$\delta I\Big|_{\text{on-shell}} = i \int d\tau \left( \frac{1}{2} \omega_\tau \delta \phi^0 + \phi^+ \delta e_\tau^- + \phi^- \delta e_\tau^+ \right) . \tag{110}$$

Demanding that this boundary term vanish leads us to a variational principle where $e_\tau^\pm$ and $\phi^0$ are held fixed at the boundary. This is the direct BF theory analogue of the variational principle in JT gravity, where the boundary metric and dilaton are held fixed, so we will refer to this choice as "JT-type boundary conditions."

We now wish to identify the modification of these JT-type boundary conditions which corresponds to a $T\bar{T}$-like deformation of the dual $(0+1)$-dimensional theory. There are two ways that one might identify the appropriate form of the deforming operator. One way is to dimensionally reduce the $T\bar{T}$ operator written in 3$d$ Chern-Simons variables, which takes the form $f^- \wedge f^+$ as reviewed in Section 2.2. Recall that, in Chern-Simons language, the operators $f^a$ are dual to the boundary vielbeins $e^a$ and therefore the $f^a$ contain the boundary stress tensor. Therefore, upon such a reduction, one component of $f$ reduces to the one-dimensional

stress tensor $T_{\tau\tau}$ which is dual to the boundary einbein $e_\tau$. Since the component of the metric in the direction along which we reduce is identified with the field $\phi^0$, the other component of $f$ reduces to the operator dual to $\phi^0$, which is $\omega_\tau$. Therefore, the dimensional reduction instructs us to deform the boundary action by an operator constructed from the combination $T_{\tau\tau}\omega_\tau$ (contracted with the appropriate einbein factors to yield a quantity which is a scalar under diffeomorphisms).

The other way that one might identify the deforming operator is by using the combination $O T$ which defined the $T\overline{T}$ deformation in JT variables and converting all expressions to BF variables. We now carry out this procedure and demonstrate that it produces an operator of the schematic form $T_{\tau\tau}\omega_\tau$ which is suggested by dimensional reduction. The (Hilbert) definition of the boundary stress tensor is

$$
\begin{aligned}
T_{\tau\tau} &= -\frac{2}{\sqrt{\gamma_{\tau\tau}}}\frac{\delta I}{\delta\gamma^{\tau\tau}} \\
&= -\frac{2}{\sqrt{\gamma_{\tau\tau}}}\left(\frac{\delta I}{\delta e_\tau^+}\frac{\delta e_\tau^+}{\delta\gamma^{\tau\tau}}\bigg|_{e_\tau^-} + \frac{\delta I}{\delta e_\tau^-}\frac{\delta e_\tau^-}{\delta\gamma^{\tau\tau}}\bigg|_{e_\tau^+}\right).
\end{aligned}
\tag{111}
$$

The map from the metric $\gamma_{\tau\tau}$ to the boundary BF fields $e_\tau^\pm$ is simply

$$
\gamma_{\tau\tau} = -4 e_\tau^+ e_\tau^-, \qquad \gamma^{\tau\tau} = -\frac{1}{4 e_\tau^+ e_\tau^-}.
\tag{112}
$$

Note that, according to our conventions (68), the relative minus sign in the definition (112) of $\gamma_{\tau\tau}$ is required to have a positive-definite worldline metric since

$$
e_\tau^+ e_\tau^- = \frac{1}{4}\left(i e_\tau^1 - e_\tau^2\right)\left(i e_\tau^1 + e_\tau^2\right) = -\frac{1}{4}\left(\left(e_\tau^1\right)^2 + \left(e_\tau^2\right)^2\right).
\tag{113}
$$

Thus the derivatives appearing in the stress tensor can be written

$$
\begin{aligned}
\frac{\delta e_\tau^+}{\delta\gamma^{\tau\tau}}\bigg|_{e_\tau^-} &= \frac{1}{(\gamma^{\tau\tau})^2}\cdot\frac{1}{4 e_\tau^-} = -\frac{e_\tau^+}{\gamma^{\tau\tau}}, \\
\frac{\delta e_\tau^-}{\delta\gamma^{\tau\tau}}\bigg|_{e_\tau^+} &= \frac{1}{(\gamma^{\tau\tau})^2}\cdot\frac{1}{4 e_\tau^+} = -\frac{e_\tau^-}{\gamma^{\tau\tau}}.
\end{aligned}
\tag{114}
$$

Meanwhile, from (110) we see that $\frac{\delta I}{\delta e_\tau^+} = i\phi^-$ and $\frac{\delta I}{\delta e_\tau^-} = i\phi^+$. So the stress tensor is

$$
T_{\tau\tau} = \frac{2i}{\sqrt{\gamma^{\tau\tau}}}\left(\phi^-\cdot\frac{e_\tau^+}{\gamma^{\tau\tau}} + \phi^+\cdot\frac{e_\tau^-}{\gamma^{\tau\tau}}\right),
\tag{115}
$$

and its trace is

$$
T = T_{\tau\tau}\gamma^{\tau\tau} = \frac{i}{\sqrt{-e_\tau^+ e_\tau^-}}\left(e_\tau^+ \phi^- + \phi^+ e_\tau^-\right).
\tag{116}
$$

Next we express the operator $O$ dual to the dilaton in BF variables. Using the map

$$
\Phi = -\frac{i}{4}\phi^0,
\tag{117}
$$

one has from (98) that

$$
\begin{aligned}
O &= \frac{1}{\sqrt{\gamma}}\frac{\delta I}{\delta\Phi_b} \\
&= \frac{2i}{\sqrt{-e_\tau^+ e_\tau^-}}\frac{\delta I}{\delta\phi^0} \\
&= -\frac{1}{\sqrt{-e_\tau^+ e_\tau^-}}\omega_\tau.
\end{aligned}
\tag{118}
$$

We conclude that, in BF variables, the combination which corresponds to a boundary $T\overline{T}$ deformation is

$$OT = \frac{i}{e_\tau^+ e_\tau^-}\left(e_\tau^+\phi^- + \phi^+ e_\tau^-\right)\omega_\tau.$$

(119)

As claimed, this matches the expectation described above from dimensionally reducing the boundary deformation of $3d$ Chern-Simons.

Now that we have identified the appropriate $T\overline{T}$ operator for a boundary deformation of our BF theory, we will apply the deformation and study the resulting modified boundary conditions. To do this, we promote the sources $\phi^0$ and $e_\tau^\pm$ to $\lambda$-dependent quantities and attempt to solve for the $\lambda$ dependence. The boundary variation now takes the form

$$\delta I(\lambda)\Big|_{\text{on-shell}} = i\int d\tau \left(\frac{1}{2}\omega_\tau\,\delta\phi^0(\lambda) + \phi^+\delta e_\tau^-(\lambda) + \phi^-\delta e_\tau^+(\lambda)\right).$$

(120)

As we discussed around equation (39), there are naïvely two ways to deform the boundary action: we can either deform the combined action (120) which already includes a boundary term, or we can deform the action without the boundary term which includes variations of both the sources and expectation values. Here we will take the latter strategy. Without the boundary term, the full variation of the action is

$$\delta I_{BF}(\lambda) = i\int d\tau \left(\frac{1}{2}(\delta\omega_\tau)\phi^0(\lambda) + \frac{1}{2}\omega_\tau(\delta\phi^0(\lambda)) + (\delta\phi^+)e_\tau^-(\lambda) + \phi^+(\delta e_\tau^-(\lambda))\right.$$

(121)

$$\left. + (\delta\phi^-)e_\tau^+(\lambda) + \phi^-(\delta e_\tau^+(\lambda))\right).$$

(122)

We will re-scale our flow equation by a factor of $-\frac{1}{4}$ for convenience, writing

$$\frac{\partial I_{BF}}{\partial\lambda} = -\frac{1}{4}\int \sqrt{\gamma_{\tau\tau}}\,d\tau\, OT = i\int \sqrt{\gamma_{\tau\tau}}\,d\tau\,\frac{1}{(-4e_\tau^+ e_\tau^-)}\left(e_\tau^+\phi^- + \phi^+ e_\tau^-\right)\omega_\tau.$$

(123)

This implies that the variation of the action also satisfies the flow

$$\frac{\partial(\delta I_{BF}(\lambda))}{\partial\lambda} = -\frac{1}{4}\int d\tau\,\delta(OT\,\sqrt{\gamma_{\tau\tau}}) = i\int d\tau\,\delta\left[\sqrt{\gamma_{\tau\tau}}\,\frac{1}{(-4e_\tau^+ e_\tau^-)}\left(e_\tau^+\phi^- + \phi^+ e_\tau^-\right)\omega_\tau\right].$$

(124)

We impose the variational principle that, at any point along the $T\overline{T}$ flow, the $\lambda$-dependent expressions for the sources $e_\tau^\pm(\lambda)$ and $\phi^0(\lambda)$ are held fixed. In particular, this means that $\delta e_\tau^\pm(\lambda) = 0$ and thus no terms are generated on the right side of (124) from $\delta$ acting on $e_\tau^\pm$ or on $\sqrt{\gamma_{\tau\tau}}$. Likewise, the terms involving the variations of these sources in (121) also vanish. We then take the $\lambda$ derivative of the surviving terms in (121) and set the result equal to the right side of (124) to obtain

$$i\int (e(\lambda)d\tau)\frac{1}{e(\lambda)}\left(\frac{1}{2}\delta\omega_\tau\frac{\partial(\phi^0(\lambda))}{\partial\lambda} + \delta\phi^+\frac{\partial(e_\tau^-(\lambda))}{\partial\lambda} + \delta\phi^-\frac{\partial(e_\tau^+(\lambda))}{\partial\lambda}\right)$$

$$= i\int (e(\lambda)d\tau)\frac{1}{e(\lambda)^2}\left(\left(e_\tau^+(\lambda)\delta\phi^- + e_\tau^-(\lambda)\delta\phi^+\right)\omega_\tau + \left(e_\tau^+(\lambda)\phi^- + e_\tau^-(\lambda)\phi^+\right)\delta\omega_\tau\right).$$

(125)

To ease notation we have defined $e(\lambda) = \sqrt{-4e_\tau^+(\lambda)e_\tau^-(\lambda)}$, so that $\gamma_{\tau\tau}(\lambda) = e(\lambda)^2$, and we have multiplied and divided by $e(\lambda)$ on the left side. We can now read off the differential equations for the $\lambda$-dependent sources from (125), finding

$$\frac{\partial(\phi^0(\lambda))}{\partial\lambda} = \frac{2}{e(\lambda)}\left(e_\tau^+(\lambda)\phi^- + e_\tau^-(\lambda)\phi^+\right),$$

$$\frac{\partial(e_\tau^-(\lambda))}{\partial\lambda} = \frac{1}{e(\lambda)}e_\tau^-(\lambda)\omega_\tau,$$

$$\frac{\partial(e_\tau^+(\lambda))}{\partial\lambda} = \frac{1}{e(\lambda)}e_\tau^+(\lambda)\omega_\tau.$$
$$(126)$$

We note that these differential equations imply that the ratios

$$\hat{e} = \sqrt{-\frac{e_\tau^+}{e_\tau^-}}, \qquad \hat{e}^{-1} = \sqrt{-\frac{e_\tau^-}{e_\tau^+}} \qquad (127)$$

do not flow with $\lambda$:

$$\frac{\partial(\hat{e}^2)}{\partial\lambda} = -\frac{(\partial_\lambda e_\tau^+)e_\tau^- - (\partial_\lambda e_\tau^-)e_\tau^+}{\left(e_\tau^-\right)^2}$$

$$= -\frac{e_\tau^+ e_\tau^- \omega_\tau - e_\tau^- e_\tau^+ \omega_\tau}{e\left(e_\tau^-\right)^2}$$

$$= 0. \qquad (128)$$

Since $\frac{e_\tau^+}{e} = \frac{1}{2}\hat{e}$ and $\frac{e_\tau^-}{e} = \frac{1}{2}\hat{e}^{-1}$, we can write the flow equations as

$$\frac{\partial(\phi^0(\lambda))}{\partial\lambda} = \left(\hat{e}\phi^- + \hat{e}^{-1}\phi^+\right),$$

$$\frac{\partial(e_\tau^-(\lambda))}{\partial\lambda} = \frac{1}{2}\hat{e}^{-1}\omega_\tau,$$

$$\frac{\partial(e_\tau^+(\lambda))}{\partial\lambda} = \frac{1}{2}\hat{e}\omega_\tau. \qquad (129)$$

Because the right sides of the three differential equations in (129) are independent of $\lambda$, they can be trivially solved to find

$$\phi^0(\lambda) = \phi^0(0) + \lambda\left(\hat{e}\phi^- + \hat{e}^{-1}\phi^+\right),$$

$$e_\tau^+(\lambda) = e_\tau^\pm(0) + \frac{\lambda}{2}\hat{e}^{-1}\omega_\tau,$$

$$e_\tau^-(\lambda) = e_\tau^\pm(0) + \frac{\lambda}{2}\hat{e}\omega_\tau. \qquad (130)$$

Replacing hatted quantities with the original variables, we conclude

$$\phi^0(\lambda) = \phi^0(0) + \frac{2\lambda}{e(0)}\left(e_\tau^+(0)\phi^- + e_\tau^-(0)\phi^+\right),$$

$$e_\tau^\pm(\lambda) = e_\tau^\pm(0) + \frac{\lambda}{e(0)}e_\tau^\pm(0)\omega_\tau. \qquad (131)$$

The rotated sources (131) define the full solution to the $T\overline{T}$ flow at finite $\lambda$. Similarly to the case of 3$d$ Chern-Simons variables, the $T\overline{T}$ deformation of 2$d$ BF theory corresponds to a linear

mixing of the expectation values $\phi^{\pm}$ and $\omega_{\tau}$ into the sources $e_{\tau}^{\pm}$ and $\phi^0$. We reiterate that the variational principle remains unchanged along the $T\overline{T}$ flow; the new $\lambda$-dependent source fields $e_{\tau}^{\pm}(\lambda)$ and $\phi^0(\lambda)$ remain fixed at the boundary at any $\lambda$, although the expressions for these sources in terms of the sources in the undeformed theory are modified according to (131). Note that we have chosen to solve the flow equation for the action $I_{BF}$, without the boundary term. To complete the solution and ensure the correct variational principle, we must go back and add a $\lambda$-dependent boundary term $I_{\text{bdry}}(\lambda)$ which takes the same form as that given in (108) except with $\phi^0$ replaced by $\phi^0(\lambda)$ as in (130).

Although this result is morally analogous to the mixing of sources and expectation values in the 3$d$ Chern-Simons context, we briefly comment on two superficial differences. The first is that, in the Chern-Simons context reviewed in Section 2.2, both the variation of the action $\delta S = \frac{1}{4\pi G} \int_{\partial M_3} \varepsilon_{ab} f^a \wedge \delta e^b$ and the $T\overline{T}$ deformation $S_{f^+ f^-} = \frac{1}{32\pi^2 G^2} \int_{\partial M_3} f^- \wedge f^+$ are written as integrals of differential 2-forms, and therefore the integrals do not contain any measure factors. However in our deformation we have chosen to write the deforming operator $OT$ as a scalar rather than as a one-form. Since the variation of the on-shell action (110) is written as the integral of a boundary one-form rather than a scalar, our solution (131) to the flow equation must introduce extra factors of $e$ to compensate for the difference in measure between the two integrals. No such measure factors appeared in the solution to the Chern-Simons flow. In principle, the presence of such $\lambda$-dependent factors could have spoiled the conclusion that the $T\overline{T}$ flow generates only a linear mixing of sources and expectation values, rather than some more complicated behavior. However, as we saw in equation (129), only certain $\lambda$-independent combinations enter the flow equation, so the simple linear form of the solution is preserved.

The second difference is that, in the Chern-Simons context, each source $e_i^a$ became a function of its dual expectation value $f_i^a$. In the BF theory solution, however, each source field became a function of the dual expectation value for a *different* field. From the perspective of dimensional reduction, this difference simply arises because we have split the two-dimensional metric into a one-dimensional metric and a scalar field $\phi^0$. This splitting causes the dual operators $T$ and $O$ to appear asymmetrically in the action, even though both expectation values descend from the two-dimensional stress tensor on the boundary of the Chern-Simons theory. Therefore the apparent difference that each source rotates into an expectation value dual to a different operator is merely an artifact of the splitting performed as part of the reduction.

To summarize this section, we have found that the addition of the appropriate $OT$-like operator in BF variables can be interpreted as a linear mixing of sources and expectation values, in the same way as the $f^- \wedge f^+$ deformation implemented such a linear mixing in 3$d$ Chern-Simons variables. It is worth pointing out that this feature is fairly generic: the addition of a double-trace operator constructed out of the expectation values dual to certain sources should generally correspond to precisely this type of change in boundary conditions, in which the sources become dependent upon the expectation values. Indeed, part of the standard lore from AdS/CFT is that the addition of a double-trace operator in the field theory corresponds to a modification of the boundary conditions for the bulk fields [95, 96], although this behavior has more often been studied in the case of relevant or marginal operators rather than irrelevant operators.

By way of analogy, we mention that there is another example where the addition of a double-trace operator leads to such a linear mixing. Let us return to the context of JT gravity and consider the operator $O$ dual to the dilaton as defined in (98). One could consider adding a boundary term to the JT gravity Lagrangian of the form

$$I_{O^2} = \mu \int_{\partial M_2} d\tau \, \sqrt{\gamma_{\tau\tau}} \, O^2 \,. \tag{132}$$

As discussed in [97], the combined on-shell variation of the JT gravity action plus $I_{O^2}$ is

$$\delta(I_{\text{BF}} + I_{O^2})\Big|_{\text{on-shell}} = \int d\tau \sqrt{\gamma_{\tau\tau}} \left( \frac{1}{2} \left( T_{\tau\tau} - \mu g_{\tau\tau} O^2 \right) \delta g^{\tau\tau} + O \delta (\Phi + 2\mu O) \right). \qquad (133)$$

This corresponds to a variational principle where both the metric $g_{\tau\tau}$ and the combination

$$\Phi(\mu) = \Phi(0) + 2\mu O \qquad (134)$$

are held fixed on the boundary. Of course, this has an identical interpretation as a linear mixing of the operator $\Phi$ into its dual expectation value $O$. We therefore see that the $f^- \wedge f^+$ deformation of $3d$ Chern-Simons, the $OT$ operator written in $2d$ BF variables, and the $O^2$ deformation of JT gravity are all of this qualitatively similar form.

## 4.2 Deformation with Schwarzian-type Boundary Term

Although the JT-like boundary conditions considered in the preceding subsection led to an especially simple modification under a $T\overline{T}$-like deformation, these are in some sense not the most natural boundary conditions to consider in BF theory. For instance, we saw in equation (25) that the dimensional reduction of $3d$ Chern-Simons theory produces a BF theory action of the form

$$I = -i \int_{M_2} \text{Tr}(\phi F) - \frac{i}{2} \oint_{\partial M_2} d\tau \, \text{Tr}(\phi^2)$$
$$\equiv I_{\text{BF}} + I_{\text{bdry}}, \qquad (135)$$

which contains an additional boundary term compared to the bare BF action (48) and where we have introduced a factor of $\frac{1}{2}$ by convention. In addition to its emergence from dimensional reduction, this boundary term is also of interest since it gives rise to a dual Schwarzian theory on the $1d$ boundary, as we reviewed in Section 3.1. We would therefore like to study the $T\overline{T}$ deformation of the theory with this choice of boundary term as well.

First it is instructive to see what variational principle this additional boundary term yields in the undeformed case. We have already seen that the on-shell variation of the first term $I_{BF}$ gives

$$\delta I_{\text{BF}}\Big|_{\text{on-shell}} = i \int_{\partial M_2} d\tau \, \text{Tr}(\phi \delta A_\tau), \qquad (136)$$

which means that the combined boundary variation including contributions from both the bulk and boundary terms is

$$\delta I\Big|_{\text{on-shell}} = \frac{i}{2} \int_{\partial M_2} d\tau \, \text{Tr}(\phi \, \delta A_\tau - \phi \, \delta \phi). \qquad (137)$$

This boundary variation will vanish if we demand that the value of the one-form $A_\mu$ on the boundary is equal to the combination $\phi \, d\tau$ where $d\tau$ is the one-form appearing in the boundary length element. We write this condition as

$$A_\tau\Big|_{\text{bdry}} = \phi\Big|_{\text{bdry}}. \qquad (138)$$

After imposing this condition, the boundary term can be written as

$$I_{\text{bdry}} = -\frac{i}{2} \int_{\partial M_2} d\tau \, \text{Tr}(A_\tau^2). \qquad (139)$$

On the other hand, the equation of motion $F = 0$ requires that $A_\mu$ is pure gauge, and can therefore be written as

$$A_\mu = g^{-1} \partial_\mu g \,, \tag{140}$$

for some group element $g \in SL(2, \mathbb{R})$. This means that the boundary term (139) becomes

$$I_{\mathrm{bdry}} = -\frac{i}{2} \int_{\partial M_2} d\tau \, \mathrm{Tr}\left( (g^{-1} \partial_\mu g)(g^{-1} \partial^\mu g) \right)$$

$$= -\frac{i}{2} \int_{\partial M_2} d\tau \, g_{ij}(x) \dot{x}^i \dot{x}^j \,, \tag{141}$$

where in the last step we have introduced coordinates $x^i(t)$ on the three-dimensional group manifold $M = SL(2, \mathbb{R})$ and the canonical metric $g_{ij}$ on $M$. The equivalence of the two lines of (141) is a standard result concerning the Cartan metric tensor on a Lie group $G$ which is induced by the Killing form on the Lie algebra $\mathfrak{g}$ of $G$.

We therefore see that, with the choice of boundary term appearing in (135), the BF theory has a boundary degree of freedom whose dynamics are described by the theory of a free particle moving on the $SL(2, \mathbb{R})$ group manifold. This theory is equivalent to the Schwarzian theory in its usual presentation [22, 98], so this derivation provides a complementary way to see that BF gauge theory is dual to the Schwarzian, although in somewhat different language than that used in Section 3.1.

We now turn to the issue of $T\overline{T}$ deforming these boundary conditions and, by extension, the dual $1d$ theory. We first note that the procedure we followed in the case of JT-type boundary conditions will not work here because we have modified the variational principle of the theory. In Section 4.1, since the value of the boundary field $\phi^0$ was held fixed at the boundary, we were free to define the operator $O$ dual to $\phi^0$ as in equation (118) and then construct the deformation $OT$. However, with our Schwarzian-type boundary conditions, the value of $\phi^0$ is not held fixed to some constant value at the boundary but is rather related to the value of $A_\tau$ via (138), and $A_\tau$ is viewed as a boundary degree of freedom. We therefore cannot define the operator $O$ in the same way and must identify a suitable replacement for $O$ in some other way.

We propose that the correct scalar which replaces $O$ for this choice of boundary conditions is the Lagrangian itself. To motivate this choice we must take a brief detour and explain a rewriting of the $T\overline{T}$ deformation for $(0+1)$-dimensional theories which appeared in [72]. First recall from [82] that, by using the $T\overline{T}$ trace flow equation and dimensionally reducing from $(1+1)$ to $(0+1)$ dimensions, one finds that the appropriate version of the $T\overline{T}$ deformation in quantum mechanics is

$$\frac{\partial I}{\partial \lambda} = \int d\tau \, \frac{H^2}{\frac{1}{2} - 2\lambda H} \,, \tag{142}$$

where $H$ is the Hamiltonian[13] of the theory, namely the object appearing in the Euclidean action as

$$I = \int d\tau \, H \,. \tag{143}$$

The solution to this flow equation is simply

$$H(\lambda) = \frac{1}{4\lambda} \left( 1 - \sqrt{1 - 8\lambda H_0} \right) . \tag{144}$$

---

[13]Because $I = \int d\tau \, H$, in our conventions the Hamiltonian is equivalent to the Euclidean Lagrangian, although the Hamiltonian is written in terms of canonical momenta $p_\mu$ rather than $\dot{X}^\mu$. This agrees with the conventions of [82], but differs from the common convention $H(X^\mu, p^\mu) = -L_E(X^\mu, \dot{X}^\mu)$.

We note in passing that, for free kinetic seed theories of the form we are interested in here, both the deformed Hamiltonian and the deformed Lagrangian have a similar square root form (144). Beginning from a generic non-linear sigma model with the Lagrangian

$$L = \frac{1}{2} G_{\mu\nu}(X) \dot{X}^\mu \dot{X}^\nu, \tag{145}$$

the Hamiltonian is given by

$$H = \frac{1}{2} G^{\mu\nu}(X) p_\mu p_\nu. \tag{146}$$

Under the $T\overline{T}$-deformation, the Hamiltonian becomes

$$H_\lambda = \frac{1 - \sqrt{1 - 4\lambda G^{\mu\nu}(X) p_\mu p_\nu}}{4\lambda}. \tag{147}$$

The deformed (Lorentzian) Lagrangian is then recovered by Legendre transformation from the deformed Hamiltonian:

$$L_\lambda = p_\mu \dot{X}^\mu - H_\lambda = \frac{\sqrt{1 + 4\lambda G_{\mu\nu}(X) \dot{X}^\mu \dot{X}^\nu} - 1}{4\lambda}. \tag{148}$$

Thus both $H_\lambda$ and $L_\lambda$ are determined by a square-root-type function of their undeformed values, up to sending $\lambda \to -\lambda$, which changes the operator driving the flow by a sign.

Next, one can define the (Euclidean) Hilbert stress tensor associated with $I$ via

$$T^{(\text{Hilb})} = -\frac{2}{\sqrt{g^{\tau\tau}}} \frac{\delta S_E}{\delta g^{\tau\tau}} = H - 2 \frac{\partial H(g^{\tau\tau})}{\partial g^{\tau\tau}} \bigg|_{g^{\tau\tau}=1}. \tag{149}$$

We note that this is a variation with respect to the worldline metric $g_{\tau\tau}$, not to be confused with the target-space metric $g_{ij}(x)$ appearing in (141). The expression (149) for $T^{(\text{Hilb})}$ simplifies in the case of $T\overline{T}$ deformations of seed theories which are "purely kinetic" in the following sense. Suppose that we begin with an undeformed Hamiltonian $H_0$ with the property that, when $H_0$ is coupled to worldline gravity, it takes the form

$$H_0(g^{\tau\tau}) = g^{\tau\tau} \mathcal{H}, \tag{150}$$

where $\mathcal{H} \equiv H_0(g^{\tau\tau} = 1)$ is some expression which is independent of the worldline metric. For instance, $\mathcal{H} = g_{ij}(x) \dot{x}^i \dot{x}^j$ in (141). Under the $T\overline{T}$ flow (142), the deformed theory $H(\lambda)$ will also only depend on the worldline metric through the combination $g^{\tau\tau} \mathcal{H}$. For such theories, the Hilbert stress tensor is

$$T^{(\text{Hilb})} = H(\lambda) - 2 \frac{\partial H}{\partial H_0} \frac{\partial H_0(g^{\tau\tau})}{\partial g^{\tau\tau}} \bigg|_{g^{\tau\tau}=1}$$

$$= H(\lambda) - 2\mathcal{H} \frac{\partial H}{\partial \mathcal{H}}. \tag{151}$$

In particular, substituting the explicit solution (144) for a $T\overline{T}$-like flow into (151) and multiplying by $H$ itself, one can compute the combination

$$HT^{(\text{Hilb})} = H \left( H - 2H_0 \frac{\partial H}{\partial H_0} \right)$$

$$= -\frac{\left( \sqrt{1 - 8\lambda H_0} - 1 \right)^2}{16\lambda^2 \sqrt{1 - 8\lambda H_0}}. \tag{152}$$

On the other hand, substituting the same solution (144) into the definition of the deforming operator in (142) gives

$$\frac{H^2}{\frac{1}{2} - 2\lambda H} = \frac{\left(\sqrt{1 - 8\lambda H_0} - 1\right)^2}{8\lambda^2 \sqrt{1 - 8\lambda H_0}}. \tag{153}$$

Thus we find that for purely kinetic seed theories one has the equivalence

$$HT^{(\text{Hilb})} = -\frac{1}{2}\left(\frac{H^2}{\frac{1}{2} - 2\lambda H}\right). \tag{154}$$

Up to a constant rescaling, we can therefore view the $T\overline{T}$ flow as being driven by the combination $HT^{(\text{Hilb})}$. We will refer to this as the $HT$ deformation for simplicity. This therefore suggests that the replacement for the operator $O$ in the $OT$ deformation is the Hamiltonian (or Euclidean Lagrangian) itself, since the object $T \equiv T_{\tau\tau} \gamma^{\tau\tau}$ defined in (111) corresponds to the Hilbert stress tensor $T^{(\text{Hilb})}$.

We now demonstrate explicitly that the proposed deformation yields the expected square-root form of the solution for the boundary action. After coupling the boundary action (139) to worldline gravity, one has

$$I_{\text{bdry}} = -\frac{i}{2}\int \sqrt{g_{\tau\tau}}\, d\tau\, g^{\tau\tau}\, \text{Tr}\left(A_\tau^2\right). \tag{155}$$

This is a seed theory of purely kinetic form since $H_0 = g^{\tau\tau}\, \text{Tr}(A_\tau^2) = g^{\tau\tau}\mathcal{H}$ for $\mathcal{H} = \text{Tr}(A_\tau^2)$. We make an ansatz for the finite-$\lambda$ deformed (Euclidean) Lagrangian as

$$I_{\text{bdry}}(\lambda) = -\frac{i}{2}\int d\tau\, \frac{1}{\lambda} f(\lambda\mathcal{H}). \tag{156}$$

The Hilbert stress tensor at finite $\lambda$ is therefore given by

$$T^{(\text{Hilb})} = \frac{1}{\lambda}\left(f(\lambda\mathcal{H}) - 2\lambda\mathcal{H}f'(\lambda\mathcal{H})\right). \tag{157}$$

Substituting this into the flow equation $\partial_\lambda H(\lambda) = H(\lambda)T^{(\text{Hilb})}$, we then find

$$\lambda\mathcal{H}f'(\lambda\mathcal{H}) - f(\lambda\mathcal{H}) = f(\lambda\mathcal{H})\left(f(\lambda\mathcal{H}) - 2\lambda\mathcal{H}f'(\lambda\mathcal{H})\right), \tag{158}$$

which has the solution

$$f(\lambda\mathcal{H}) = \frac{1}{2}\left(\sqrt{1 + 4\lambda\mathcal{H}} - 1\right). \tag{159}$$

The deformed boundary action is therefore

$$I_{\text{bdry}}(\lambda) = -\frac{i}{2}\int d\tau\, \frac{1}{2\lambda}\left(\sqrt{1 + 4\lambda\,\text{Tr}(A_\tau^2)} - 1\right), \tag{160}$$

as expected.

We would also like an interpretation of this $HT$ deformation in terms of a rotation between sources and expectation values, as we had in the case of the $OT$ deformation of Section 4.1. However with this choice of boundary conditions such an interpretation is somewhat obscured because the two objects in the product $HT$ are not both expectation values dual to some

sources, as the objects $O$ and $T$ were in the case of JT-type boundary conditions. One partial interpretation is the following. The flow equation induced by $HT$ can be written as

$$\frac{\partial H}{\partial \lambda} = H^2 - 2H \frac{\partial H(g^{\tau\tau})}{\partial g^{\tau\tau}}\bigg|_{g^{\tau\tau}=1}. \tag{161}$$

This equation is functionally similar to the inviscid Burgers' equation determining the cylinder energy levels of a $T\overline{T}$-deformed QFT in two dimensions:

$$\frac{\partial E_n}{\partial \lambda} = \frac{1}{R} P_n^2 + E_n \frac{\partial E_n}{\partial R}. \tag{162}$$

Restricting to the sector $P_n = 0$, equation (162) has the implicit solution

$$E_n(R, \lambda) = E_n(R + \lambda E_n(R, \lambda), 0). \tag{163}$$

This has the interpretation that the energy eigenstates of the deformed theory see a cylinder with an effective energy-dependent radius.

The analogous manipulation of equation (161) is to ignore the $H^2$ term on the right side. This is harder to justify but we will return to this assumption in a moment. The resulting equation is

$$\frac{\partial H}{\partial \lambda} = -2H \frac{\partial H(g^{\tau\tau})}{\partial g^{\tau\tau}}\bigg|_{g^{\tau\tau}=1}, \tag{164}$$

which again has the implicit solution

$$H(g^{\tau\tau}, \lambda) = H(g^{\tau\tau} - 2\lambda H(g^{\tau\tau}, \lambda), 0). \tag{165}$$

The interpretation of (165) is very similar to that of the solution to the $T\overline{T}$ flow for $3d$ Chern-Simons theory reviewed in Section 2.2. In that case, the solution for the deformed boundary vielbein took the form

$$e_i^a(\lambda) = e_i^a(0) + \frac{\lambda}{8\pi G} f_i^a. \tag{166}$$

Here $e_i^a$ is the source which determines the metric and $f_i^a$ is the operator dual to $e_i^a$, which is related to the boundary stress tensor. Equation (166) therefore represents a rotation from the undeformed metric (controlled by $e_i^a(0)$) to a deformed metric (determined by $e_i^a(\lambda)$) which is a function of the stress tensor. But a stress-tensor-dependent (or Hamiltonian-dependent) metric is exactly what we see in the implicit solution (165).

Although this is an appealing interpretation, it is obstructed by the presence of the $H^2$ source term in (161). The full interpretation of this flow equation may involve simultaneously introducing a Hamiltonian-dependent metric and also allowing another source-like quantity to depend on its dual expectation value. The identity of this other source-like quantity is not obvious, since it seems that the metric and its dual stress tensor are the only pair of such fields which remain. One speculative possibility is suggested by the observation of [99] that, in a certain sense, the object dual to the Lagrangian density in $d$ dimensions is the quantity $\sqrt{T^{\mu\nu}T_{\mu\nu}}$ where $T_{\mu\nu}$ is the stress tensor. This proposal is motivated by certain considerations in non-linear electrodynamics, and indeed a similar operator involving a square root of stress tensor bilinears was shown to drive a flow which deforms the free Maxwell theory to the recently discovered ModMax theory [100–102] in four dimensions [103]; see [104] for a related analysis including supersymmetry.

In our case, the stress tensor has only a single component so this proposal reduces to the statement that the (Euclidean) Lagrangian $L_E = H$ is dual to $T$. If so, our deforming operator

$HT$ might admit an interpretation as a product of two expectation values dual to sources, with the first dual to the stress tensor and the second dual to the metric. In analogy with the $OT$ deformation of Section 4.1, one might expect this deformation to simultaneously introduce linear $\lambda$-dependence of the stress tensor on the Hamiltonian, and dependence of the metric on the stress tensor. It is conceivable that such a coupled system of mixing equations would give rise to the flow equation (161), but we will not pursue this interpretation further here.

We conclude this section by offering two interpretations of the deformation in the case of Schwarzian-type boundary conditions.

The first interpretation is motivated by the fact that a $T\overline{T}$ deformation of the boundary theory of $3d$ Chern-Simons can be viewed as replacing the field $e_i^a$ appearing in the expansion of $A_\mu$ with a $\lambda$-dependent version $e_i^a(\lambda)$. That is, the deformation corresponds to replacing the Chern-Simons gauge field $A_\mu$ with some $A_\mu(\lambda)$. This can be viewed as a field redefinition from an undeformed gauge field to a deformed gauge field.

Similarly, in the case of BF gauge theory with Schwarzian-type boundary conditions, we see that the boundary action for $A_\tau$ has been modified by making the replacement

$$\mathrm{Tr}\big(A_\tau^2\big) \longrightarrow \frac{1}{2\lambda}\left(\sqrt{1+4\lambda\,\mathrm{Tr}\big(A_\tau^2\big)}-1\right). \tag{167}$$

We may of course interpret this replacement by defining an effective $\lambda$-dependent gauge field $A_\tau(\lambda)$, which satisfies $\mathrm{Tr}\big(A_\tau(\lambda)^2\big) = \frac{1}{2\lambda}\big(\sqrt{1+4\lambda\,\mathrm{Tr}(A_\tau(0)^2)}-1\big)$. From this perspective, the functional form of the boundary action

$$I_{\mathrm{bdry}}(\lambda) = -\frac{i}{2}\int d\tau\,\mathrm{Tr}\big(A_\tau(\lambda)^2\big) \tag{168}$$

remains unchanged, but it appears different when we express the Lagrangian in terms of the undeformed gauge field $A_\tau(0)$, which yields

$$I_{\mathrm{bdry}}(\lambda) = -\frac{i}{2}\int d\tau\,\frac{1}{2\lambda}\left(\sqrt{1+4\lambda\,\mathrm{Tr}(A_\tau(0)^2)}-1\right), \tag{169}$$

as before.

In the above discussion, we assumed that the boundary condition (138) relating $A_\tau$ to $\phi$ on the boundary did not flow with $\lambda$. That is, the gauge field $A_\tau$ underwent a field redefinition but its relationship with $\phi$ was unmodified. However, a second interpretation is that the $T\overline{T}$ flow modifies the relationship between these fields.

To see this, we again consider the deformed Euclidean Lagrangian (169). Rather than thinking of this as an action defining a one-dimensional theory in isolation, suppose we consider the bulk theory with this choice of boundary term as

$$\begin{aligned}I &= I_{\mathrm{BF}} + I_{\mathrm{bdry}}\\ &= -i\int_{M_2}\mathrm{Tr}(\phi F) - \frac{i}{4\lambda}\int d\tau\left(\sqrt{1+4\lambda\,\mathrm{Tr}\big(A_\tau^2\big)}-1\right).\end{aligned} \tag{170}$$

The combined on-shell variation is then

$$\delta I\Big|_{\text{on-shell}} = i\int_{\partial M_2}\mathrm{Tr}\left(\phi\,\delta A_\tau - \frac{A_\tau\,\delta A_\tau}{\sqrt{1+4\lambda\,\mathrm{Tr}\big(A_\tau^2\big)}}\right). \tag{171}$$

This boundary term vanishes if we impose

$$\phi\Big|_{\mathrm{bdry}} = \frac{A_\tau}{\sqrt{1+4\lambda\,\mathrm{Tr}\big(A_\tau^2\big)}}\bigg|_{\mathrm{bdry}}, \tag{172}$$

which is a $\lambda$-dependent modification of (138). We therefore see that we can either interpret the boundary deformation as (1) redefining $A_\tau$ to $A_\tau(\lambda)$ while leaving the boundary condition unchanged, or (2) leaving the field $A_\tau$ unchanged but modifying the boundary condition which relates $A_\tau$ to $\phi$.

## 5  Gravitational AdS$_3$ Wilson Lines

In this section, we perturbatively compute the deformed classical and quantum gravitational Wilson line and its correlators in AdS$_3$. As a consistency check, our classical gravitational Wilson line correlator analysis is consistent with previous results on $T\overline{T}$-deformed scalar correlators [50, 51, 53] for constant stress tensor backgrounds.

Before beginning the calculations, we pause to clarify one point about the deformed Wilson line. In the undeformed theory, it is common to parameterize the Wilson line in terms of the boundary stress tensor as

$$
\begin{aligned}
W[z_2, z_1] &= P \exp\left( \int_{z_1}^{z_2} a_i \, dx^i \right) \\
&= P \exp\left( \int_{z_1}^{z_2} dy \left( L_1 + \frac{6}{c} T_{zz}(y) L_{-1} \right) \right).
\end{aligned}
\tag{173}
$$

In terms of the expansion coefficients $e_i^a, f_i^a$ appearing in the boundary connection $a_i$ which we introduced in Section 2.1, this corresponds to setting $e_i^+ = \frac{1}{2}$ and $e_i^- = -\frac{1}{2}$. Such a choice corresponds to a Bañados-type connection of the form in (14).

However, we have seen that this class of boundary conditions is not compatible with a boundary $T\overline{T}$ deformation. Although the undeformed theory can always be brought into Bañados form by a diffeomorphism, in the deformed theory the sources $e_i^a$ will become dependent upon the dual expectation values $f_i^a$. In this case, we cannot use equation (173) for the Wilson line and must instead consider an operator of the schematic form

$$
W[z_2, z_1]_\lambda = P \exp\left[ \int_{z_1}^{z_2} dy \left( e_i(\lambda) L_1 + \frac{6}{c} T_{zz}(y) L_{-1} \right) \right].
\tag{174}
$$

One can therefore imagine two possible prescriptions for finding the leading corrections to Wilson line correlators in the $T\overline{T}$-deformed theory. The first possibility is to use the expression (174) for the deformed Wilson line and expand the result to the first non-trivial order in $\lambda$. The second is to apply conformal perturbation theory to the expression (173) for the Wilson line in the undeformed theory.

We will primarily use the second approach, relying on conformal perturbation theory, in the analysis of this section. However, in some examples we will also examine the leading correction using the first approach and verify that it gives a contribution of the same form. We emphasize that, in applying conformal perturbation theory to find the leading correction, one does *not* need to include corrections from the deformed $e_i(\lambda)$. That is, to find the first non-trivial correction to correlation functions, conformal perturbation theory instructs us to use expressions in the undeformed theory, so we do not need to consider the mixing of sources and expectation values at this order. However we will comment in later calculations on the structures that one would expect to appear at higher orders in $\lambda$ due to this linear mixing.

## 5.1 Classical AdS$_3$ Wilson Line

The gravitational AdS$_3$ Wilson line anchored at the endpoints $z_1$ and $z_2$ is conjectured to be dual to a bi-local primary operator:

$$\langle W[z_2, z_1]\rangle_0 \longleftrightarrow \langle O(z_2)O(z_1)\rangle_0. \tag{175}$$

Given two arbitrary AdS$_3$ bulk points $Z_1 = (r_1, z_1)$ and $Z_2 = (r_2, z_2)$, the classical Wilson line is defined as the path-ordered integral

$$W[(r_2, z_2; r_1, z_1)]_0 = P \exp\left(\int_{(r_1,z_1)}^{(r_2,z_2)} A\right). \tag{176}$$

Under a gauge transformation, the Wilson line transforms as

$$W[(r_2, z_2; r_1, z_1)]_0 \to g(r_2, z_2)^{-1} W[(r_2, z_2; r_1, z_1)]_0 g(r_1, z_1), \tag{177}$$

with $g \in SL(2, \mathbb{R})$ and $A \to g^{-1}(d + A)g$. In particular, the radial dependence of the connection (13) arises through a gauge transformation:

$$W[(r_2, z_2; r_1, z_1)]_0 = b(r_2)^{-1} P \exp\left(\int_{z_1}^{z_2} a\right) b(r_1). \tag{178}$$

The matrix elements of $W[z_2, z_1]$ between the lowest and highest weight states are

$$\begin{aligned}
\langle W[z_2, z_1]\rangle_0 &= \langle j, -j \mid P \exp\left(\int_{z_1}^{z_2} a\right) \mid j, j\rangle_0 \\
&= \langle j, -j \mid P \exp\left[\int_{z_1}^{z_2} dz\left(L_1 + \frac{6}{c} T_{zz}(z) L_{-1}\right)\right] \mid j, j\rangle_0,
\end{aligned} \tag{179}$$

where we used (14) in the last line and $|j, m\rangle$ is the state of weight $m$ in the spin-$j$ representation of $SL(2, \mathbb{R})$. To see the bi-localness of the classical Wilson line, first consider the vacuum state of $3d$ gravity. In the vacuum state, the path-ordered integral reduces to an ordinary integral

$$\langle W[z_2, z_1]\rangle_0\big|_{T_{zz}=0} = \langle j, -j \mid \exp\left(\int_{z_1}^{z_2} dz\, L_1\right) \mid j, j\rangle_0 = z_{21}^{2j}, \tag{180}$$

where $z_{ij} = z_i - z_j$ and the bi-local primary field has dimension $h = -j$.

One can recover the case when $T_{zz} \neq 0$ through a local conformal transformation $z \to f(z)$ which is given by inverting the Schwarzian

$$T_{zz} = \frac{c}{12}\{f(z), z\}. \tag{181}$$

As a result, the classical Wilson line for a general background is

$$\langle W[z_2, z_1]\rangle_0\big|_{T_{zz}} = \frac{[f(z_2) - f(z_1)]^{2j}}{[f'(z_2)f'(z_1)]^j}, \tag{182}$$

and behaves as a bi-local primary operator at the endpoints. Intuitively, a way to argue for the bi-locality of the Wilson line is due to the fact that the Chern-Simons equations of motion for the connections are flat. Consequently, this makes the Wilson line path-independent between the two endpoints.

## 5.2 Classical Deformed AdS$_3$ Wilson Line

Observables in $T\overline{T}$-deformed theories can often be computed in terms of quantities in the undeformed QFT. Thus it is natural to wonder whether the operator $O_\lambda(z_2)O_\lambda(z_1)$, which descends from the operator $O(z_2)O(z_1)$ upon applying the $T\overline{T}$ deformation to a CFT, also admits an interpretation as a deformed Wilson line. More precisely, our aim is to compute the matrix element

$$\langle W[z_2, z_1]\rangle_\lambda \equiv \langle j, -j \,|\, P\exp\left(\int_{z_1}^{z_2} dz\, a(z)\right) \,|\, j, j\rangle_\lambda, \tag{183}$$

which corresponds to the expectation value of the deformed operator $O_\lambda(z_2)O_\lambda(z_1)$ in the $T\overline{T}$-deformed theory.

In this subsection, we classically compute the deformed Wilson line in terms of the undeformed Wilson line via a linear rotation of the coordinates $x^\pm$. From (166), we express the undeformed metric in terms of the deformed metric

$$\begin{aligned}
g_{ij}(0) &= \eta_{ab} e_i^a(0) e_j^b(0) \\
&= g_{ij}(\lambda) - \frac{\lambda}{8\pi G}\eta_{ab}\left(e_i^a(\lambda)f_j^b + e_j^b(\lambda)f_i^a\right) + \left(\frac{\lambda}{8\pi G}\right)^2 \eta_{ab}f_i^a f_j^b.
\end{aligned} \tag{184}$$

Using the conventions $\eta_{+-} = \eta_{-+} = -1$, the undeformed metric is

$$g_{ij}(0) = g_{ij}(\lambda) + \frac{\lambda}{8\pi G}\left(e_i^+ f_j^- + e_i^- f_j^+ + e_j^- f_i^+ + e_j^+ f_i^-\right) - \left(\frac{\lambda}{8\pi G}\right)^2\left(f_i^- f_j^+ + f_i^+ f_j^-\right), \tag{185}$$

where we have suppressed the $\lambda$ dependence in each $e_i^a(\lambda)$ on the right side. The explicit components are

$$g_{++}(0) = g_{++}(\lambda) + \frac{\lambda}{4\pi G}\left(e_+^+ f_+^- + e_+^- f_+^+\right) - 2\left(\frac{\lambda}{8\pi G}\right)^2 f_+^- f_+^+,$$

$$g_{+-}(0) = g_{+-}(\lambda) + \frac{\lambda}{8\pi G}\left(e_+^+ f_-^- + e_+^- f_-^+ + e_-^- f_+^+ + e_-^+ f_+^-\right) - \left(\frac{\lambda}{8\pi G}\right)^2\left(f_+^- f_-^+ + f_+^+ f_-^-\right), \tag{186}$$

$$g_{--}(0) = g_{--}(\lambda) + \frac{\lambda}{4\pi G}\left(e_-^+ f_-^- + e_-^- f_-^+\right) - 2\left(\frac{\lambda}{8\pi G}\right)^2 f_-^- f_-^+.$$

Furthermore, in the conventions of [77], the undeformed boundary information is

$$\begin{aligned}
e_+^+ &= \frac{1}{2}, & e_-^+ &= 0, & e_+^- &= 0, & e_-^- &= \frac{1}{2}, \\
f_+^+ &= 0, & f_-^+ &= \overline{\mathcal{L}}, & f_+^- &= \mathcal{L}, & f_-^- &= 0,
\end{aligned} \tag{187}$$

and provided the boundary metric is flat, $g_{ij}(0) = \eta_{ij}$, we can determine the diffeomorphism exactly to all orders by making the following change of coordinates:

$$y^+ = c_1 x^+ + c_2 x^-, \quad y^- = c_3 x^+ + c_4 x^-, \tag{188}$$

where the constants $c_i$ are given by

$$c_1 = \frac{1 + \left(\frac{\lambda}{8\pi G}\right)^2 \mathcal{L}\overline{\mathcal{L}} + \sqrt{1 + \left(\frac{\lambda}{8\pi G}\right)^2 \mathcal{L}\overline{\mathcal{L}} \left(1 + \left(\frac{\lambda}{8\pi G}\right)^2 \mathcal{L}\overline{\mathcal{L}}\right)}}{2\sqrt{1 + \left(\frac{\lambda}{8\pi G}\right)^2 \mathcal{L}\overline{\mathcal{L}} \left(1 + \left(\frac{\lambda}{8\pi G}\right)^2 \mathcal{L}\overline{\mathcal{L}}\right)}},$$

$$c_2 = \frac{1 + \left(\frac{\lambda}{8\pi G}\right)^2 \mathcal{L}\overline{\mathcal{L}} - \sqrt{1 + \left(\frac{\lambda}{8\pi G}\right)^2 \mathcal{L}\overline{\mathcal{L}} \left(1 + \left(\frac{\lambda}{8\pi G}\right)^2 \mathcal{L}\overline{\mathcal{L}}\right)}}{\frac{\lambda}{8\pi G}\mathcal{L}\sqrt{1 + \left(\frac{\lambda}{4\pi G}\right)^2 \mathcal{L}\overline{\mathcal{L}} \left(1 + \left(\frac{\lambda}{8\pi G}\right)^2 \mathcal{L}\overline{\mathcal{L}}\right)}},$$

$$c_3 = \frac{\frac{\lambda}{8\pi G}\mathcal{L}}{2\sqrt{1 + \left(\frac{\lambda}{8\pi G}\right)^2 \mathcal{L}\overline{\mathcal{L}} \left(1 + \left(\frac{\lambda}{8\pi G}\right)^2 \mathcal{L}\overline{\mathcal{L}}\right)}},$$

$$c_4 = \frac{1}{\sqrt{1 + \left(\frac{\lambda}{8\pi G}\right)^2 \mathcal{L}\overline{\mathcal{L}} \left(1 + \left(\frac{\lambda}{8\pi G}\right)^2 \mathcal{L}\overline{\mathcal{L}}\right)}}. \tag{189}$$

Therefore, the flat space metric transforms via

$$\eta_{ij} \to g_{ij} = \eta_{ij} + \begin{bmatrix} \frac{\lambda}{8\pi G}\mathcal{L} & -\left(\frac{\lambda}{8\pi G}\right)^2 \mathcal{L}\overline{\mathcal{L}} \\ -\left(\frac{\lambda}{8\pi G}\right)^2 \mathcal{L}\overline{\mathcal{L}} & \frac{\lambda}{8\pi G}\overline{\mathcal{L}} \end{bmatrix}. \tag{190}$$

In other words, (190) linearly mixes $z$ and $\bar{z}$ as

$$z' = c_1 z + c_2 \bar{z}, \quad \bar{z}' = c_3 z + c_4 \bar{z}. \tag{191}$$

Now, the deformed Wilson line can be evaluated *exactly* by evaluating the undeformed Wilson line at the new coordinates:

$$\langle W[z_2, z_1] \rangle_\lambda = \langle W[c_1 z_2 + c_2 \bar{z}_2, c_1 z_1 + c_2 \bar{z}_1] \rangle_0. \tag{192}$$

*Constant Stress Tensor Background*

For illustrative purposes, we will specialize to the elementary example of constant $T$ and $\overline{T}$ (i.e., the bulk contains a BTZ black hole of mass $M$ and angular momentum $J$). Thus, the Schwarzian condition (181) which we wish to invert to solve for $f$ trivializes to

$$T_{zz} = \frac{c}{12}\left(\frac{\partial_z^3 f(z)}{\partial_z f(z)} - \frac{3}{2}\left(\frac{\partial_z^2 f(z)}{\partial_z f(z)}\right)^2\right). \tag{193}$$

For $T_{zz} \neq 0$ and $T_{\bar{z}\bar{z}} \neq 0$, the solutions to (193) are

$$f(z) = b_2 \sqrt{\frac{c}{6T_{zz}}} \tan\left(\sqrt{\frac{6T_{zz}}{c}}(z + 2b_1 c)\right) + b_3, \tag{194}$$

where $b_i$ are integration constants. From knowing $f(z)$ in (194), we write the undeformed Wilson line as

$$\langle W[z_2, z_1] \rangle_0 = \frac{(f'(z_1) f'(z_2))^h}{(f(z_2) - f(z_1))^{2h}}$$

$$= \frac{\left( b_2^2 \frac{c}{6T_{zz}} \frac{6T_{zz}}{c} \sec^2\left( \sqrt{\frac{6T_{zz}}{c}} (z_1 + 2b_1 c) \right) \sec^2\left( \sqrt{\frac{6T_{zz}}{c}} (z_2 + 2b_1 c) \right) \right)^h}{\left( b_2 \sqrt{\frac{c}{6T_{zz}}} \tan\left( \sqrt{\frac{6T_{zz}}{c}} (z_2 + 2b_1 c) \right) - b_2 \sqrt{\frac{c}{6T_{zz}}} \tan\left( \sqrt{\frac{6T_{zz}}{c}} (z + 2b_1 c) \right) \right)^{2h}}$$

$$= \left( \sqrt{\frac{6T_{zz}}{c}} \frac{1}{\sin\left( \sqrt{\frac{6T_{zz}}{c}} (z_2 - z_1) \right)} \right)^{2h} .$$

$$(195)$$

Using the coordinate transformation (192), the deformed Wilson line is

$$\langle W[z_2, z_1] \rangle_\lambda = \left( \sqrt{\frac{6T_{zz}}{c}} \frac{1}{\sin\left( \sqrt{\frac{6T_{zz}}{c}} (c_1(z_2 - z_1) + c_2(\bar{z}_2 - \bar{z}_1)) \right)} \right)^{2h} . \qquad (196)$$

For the vacuum state of $3d$ gravity (i.e., $T_{zz} = T_{\bar{z}\bar{z}} = 0$), the solutions of (193) are fractional linear functions of $z$:

$$f(z)\big|_{T_{zz} = T_{\bar{z}\bar{z}} = 0} = \frac{a_1 z + a_2}{a_3 z + a_4}, \qquad (197)$$

where $a_1 a_4 - a_2 a_3 \neq 0$.

Therefore from (189) and (192), the deformed gravitational Wilson line in the vacuum state is *exactly* the same as the unperturbed CFT result

$$\langle W[z_2, z_1] \rangle_\lambda = z_{21}^{-2h} . \qquad (198)$$

This result is consistent with Cardy's $T\overline{T}$-deformed correlator analysis [51] which found that the vacuum $n$-point function of any string of chiral operators should be uncorrected to leading order in $\lambda$ under the $T\overline{T}$ deformation. In particular, the leading correction to any $n$-point function was shown to obey

$$\delta \left\langle \prod_P \Phi_p(z_p, \bar{z}_p) \right\rangle_\lambda = -4\lambda \sum_{m \neq n} \left( \log \left| \frac{z_m - z_n}{\varepsilon} \right| \right) \partial_{z_m} \partial_{\bar{z}_n} \left\langle \prod_P \Phi_p(z_p, \bar{z}_p) \right\rangle_0 . \qquad (199)$$

Before deforming, the gravitational Wilson line corresponds to a bi-local primary operator in the CFT. Its expectation value should therefore behave like a two-point correlation function evaluated in some state determined by the stress tensor, or equivalently a four-point function in the vacuum. However, Cardy's result (199) contains an anti-holomorphic derivative $\partial_{\bar{z}_n}$ which annihilates any string of purely chiral operators. Therefore, it seems that the leading correction to any Wilson line expectation value in a state prepared by purely chiral operators would be zero. In particular, the $O(\lambda)$ correction appears to vanish in the vacuum and in an extremal BTZ black hole background (which has $T_{zz} \neq 0$ but $\overline{\mathcal{L}}_0 = T_{\bar{z}\bar{z}} = 0$, corresponding to an excitation only on the chiral side of the CFT).

## 5.3 Quantum Deformed AdS$_3$ Wilson Line

We now address the quantum corrections to the deformed Wilson line, but we first review a few details which will be necessary for the later discussion. The quantum Wilson line is obtained by beginning with the definition (173) of the classical Wilson line, where the stress tensor $T_{zz}$ is thought of as a commuting number, and promoting the stress tensor to an operator of the CFT. The resulting object is conjectured to behave as a bi-local primary operator at its endpoints, $\langle W[z_2, z_1]\rangle_0 = z_{21}^{-2h(j,c)}$. Because the stress tensor is now an operator, short-distance singularities arise from the stress tensor OPE, so the scaling dimension $h(j,c)$ of the Wilson line experiences quantum corrections of the form[14]

$$h(j,c) = \sum_{n=0}^{\infty} \frac{h_n(j)}{c^n} \, . \tag{201}$$

Due to these short-distance singularities from the stress tensor OPE, we must regularize the gravitational Wilson line in order to verify that the quantum Wilson line

$$\langle W[z_2, z_1]\rangle_0 = \langle j, -j \mid P \exp\left(\int_{z_1}^{z_2} dz \left(L_1 + \frac{6}{c} T_{zz}(z) L_{-1}\right)\right) \mid j, j\rangle_0 = \tag{202}$$

$$\sum_{n=0}^{\infty} \int_{z_1}^{z_2} dy_n \int_{z_1}^{y_n} dy_{n-1} \cdots \int_{z_1}^{y_2} dy_1 \langle j, -j \mid \left(L_1 + \frac{6}{c} T_{zz}(y_n) L_{-1}\right) \times$$

$$\cdots \left(L_1 + \frac{6}{c} T_{zz}(y_1) L_{-1}\right) \mid j, j\rangle_0 \tag{203}$$

captures the correct scaling dimensions (201) as a bi-local primary operator. Further perturbative evidence of the Wilson line's bi-localness was provided in [9, 10, 17]. The authors of [17] successfully calculated the undeformed quantum Wilson line $\langle W[z; 0]\rangle_0$ up to $O(\frac{1}{c})$ and encountered some ambiguities in the coefficients at the two-loop order, $O(\frac{1}{c^2})$, due to the absence of a systematic renormalization scheme that preserves conformal invariance. The most promising scheme is the dimensional regularization approach used in [9], where an overall multiplicative renormalization $N(\varepsilon)$ and a renormalization of the vertex factor $\alpha(\varepsilon)$ were needed in $d = 2 - \varepsilon$ dimensions:

$$\lim_{\varepsilon \to 0} \langle W_\varepsilon[z_2, z_1]\rangle_0 = z_{21}^{2j} \lim_{\varepsilon \to 0} N(\varepsilon) \langle j, -j \mid P \exp\left(\frac{6\alpha(\varepsilon)}{c} \int_{z_1}^{z_2} dy \left(L_1 + \frac{6}{c} T_{zz}(y) L_{-1}\right)\right) \mid j, j\rangle_0 \, . \tag{204}$$

Here $N(\varepsilon)$ and $\alpha(\varepsilon)$ are chosen order-by-order in $\frac{1}{c}$ to cancel the poles in $\varepsilon$. The authors in [9] corrected the issue which arose at $O(\frac{1}{c^2})$ in [17]. They also carefully calculated and confirmed the $O(\frac{1}{c^3})$ corrections to the Wilson line.

Using the systematic renormalization approach in [9], the authors of [10] calculated Wilson line correlators with multiple stress tensors insertions $\left\langle \prod_{i=1}^{n} T_{zz}(w_i) W[z_2, z_1]\right\rangle$ and found results consistent with the expectation that the Wilson line yields the vacuum Virasoro OPE block (3)-(4). However, whether the quantum Wilson line behaves as a bi-local primary operator non-perturbatively in $\frac{1}{c}$ is still unknown as dimensional regularization may violate conformal invariance. This completes our review of the quantum Wilson line; we now set up the necessary formalism to compute the deformed quantum Wilson line.

---

[14]The specific values of $h_n(j)$ are easily calculable from [9,10,17]. For instance, a few values are:

$$h_0(j) = -j, \quad h_1(j) = -6j(j+1), \quad h_2(j) = -78j(j+1), \quad h_3(j) = -1230j(j+1), \quad h_4(j) = -21606j(j+1) \, . \tag{200}$$

We begin with the expression (173) for the Wilson line in terms of the boundary stress tensor, which is valid in the undeformed theory because the connections can be brought into Bañados form:

$$W[z_2, z_1] = P \exp\left( \int_{z_1}^{z_2} dy \left( L_1 + \frac{6}{c} T_{zz}(y) L_{-1} \right) \right). \tag{205}$$

Following [17], we write (205) in a more convenient form by defining

$$V[z_1, z_2] = \exp(-L_1 z_{21}) W[z_1, z_2], \tag{206}$$

so that

$$
\begin{aligned}
\frac{d}{dz_2} V[z_1, z_2] &= \exp(-L_1 z_{21}) \frac{6}{c} T_{zz}(z_2) L_{-1} \exp(L_1 z_{21}) V[z_1, z_2] \\
&= \frac{6}{c} \left( (1 - L_1 z_{21}) L_{-1} (1 + L_1 z_{21}) \right) T_{zz}(z_2) V[z_1, z_2] \\
&= \frac{6}{c} \left( L_{-1} + z_{21}[L_{-1}, L_1] - z_{21}^2 L_1 L_{-1} L_1 \right) T_{zz}(z_2) V[z_1, z_2] \\
&= \frac{6}{c} \left( L_{-1} - 2 z_{21} L_0 + z_{21}^2 L_1 \right) T_{zz}(z_2) V[z_1, z_2],
\end{aligned}
\tag{207}
$$

where we have used the facts $L_{\pm 1}^2 = 0$, $L_1 L_{-1} L_1 = -L_1$, and $[L_{-1}, L_1] = -2 L_0$.

Here (207) is solved by the usual path-ordered exponential and this allows us to write the Wilson line in a more convenient form to systematically implement a $\frac{1}{c}$ expansion:

$$
\langle W[z_2, z_1] \rangle =
$$
$$
\langle j, -j \mid \exp(z_{21} L_1) P \exp\left( \frac{6}{c} \int_{z_1}^{z_2} \left( L_{-1} - 2(y - z_1) L_0 + (y - z_1)^2 L_1 \right) T_{zz}(y) dy \right) \mid j, j \rangle.
\tag{208}
$$

The gravitational Wilson line in this form (208) can be understood as a perturbative expansion in $\frac{1}{c}$ of self-energy Feynman diagrams:

$$\langle W[z_2, z_1] \rangle_0 = \quad \longrightarrow \quad + \quad \text{(diagram)}$$

$$+ \quad \text{(diagrams)} \tag{209}$$

The first diagram in (209) contributes at $O(\frac{1}{c^0})$, the second diagram contributes at $O(\frac{1}{c})$, the final four diagrams contribute at $O(\frac{1}{c^2})$, and the ellipsis denotes higher order quantum corrections past $O(\frac{1}{c^2})$.

For every vertex, in the undeformed case, we have holomorphic stress tensor insertions. For $n$ vertices, we have an $n$-point correlator of holomorphic stress tensors to integrate over. Using

this setup for the quantum gravitational Wilson line, writing down formal expressions for the deformed corrections from the $n$-point deformed stress tensor correlators is straightforward.

It was checked in [60] that, up to eighth order in fields, the deformed planar boundary action of $3d$ gravity is the expected Nambu-Goto action written in Hamiltonian form

$$
\begin{aligned}
I &= \frac{1}{32\pi G} \int d^2 x \left[ if'\dot{f} - i\bar{f}'\dot{f} - 4 \frac{\sqrt{1 - \frac{r_c}{2}\left(f'^2 + \bar{f}'^2\right) + \frac{1}{16}r_c^2\left(f'^2 - \bar{f}'^2\right)^2} - 1}{r_c} \right] \\
&= \frac{1}{16\pi G} \int d^2 x \left( f'\partial_{\bar{z}} f + \bar{f}'\partial_z \bar{f} + \frac{1}{4} r_c f'^2 \bar{f}'^2 + O(r_c^2) \right),
\end{aligned}
\tag{210}
$$

where Lorentz symmetry is non-linearly realized upon the fundamental fields $(f', \bar{f}')$. By "up to eighth order in fields," we mean that all terms of the deformed action written in the schematic form $f^m \bar{f}^n$ with $m + n \le 8$ were computed and agree with (210).

Working perturbatively in $r_c$, the components of the stress tensor $T_{ij}(x)$ associated with (210) are a series of higher derivative corrections due to the non-localness of the interacting boundary graviton field theory. In complex coordinates, the stress tensor's components up to cubic order are

$$
\begin{aligned}
T_{zz} &= \frac{1}{8G} f'' - \frac{1}{16G} f'^2 + \frac{r_c}{16G} f''' \bar{f}' - \frac{r_c}{32G}\left(f'^2 - 2f'\bar{f}'\right)' \bar{f}' + \frac{r_c^2}{64G}\left(f'''' \bar{f}'^2 + \left(f'^2\right)'' \bar{f}''\right) \\
&\quad + \text{quartic},
\end{aligned}
$$

$$
\begin{aligned}
T_{\bar{z}\bar{z}} &= \frac{1}{8G} \bar{f}'' - \frac{1}{16G} \bar{f}'^2 + \frac{r_c}{16G} \bar{f}''' f' - \frac{r_c}{32G}\left(\bar{f}'^2 - 2\bar{f}'f'\right)' f' + \frac{r_c^2}{64G}\left(\bar{f}'''' f'^2 + \left(\bar{f}'^2\right)'' f''\right) \\
&\quad + \text{quartic},
\end{aligned}
$$

$$
\begin{aligned}
T_{z\bar{z}} &= -\frac{r_c}{16G} f'' \bar{f}'' + \frac{r_c}{32G}\left(f'' \bar{f}'^2 + f'^2 \bar{f}''\right) - \frac{r_c^2}{32G}\left(f''' \bar{f}' \bar{f}'' + f' f'' \bar{f}'''\right) \\
&\quad + \text{quartic},
\end{aligned}
\tag{211}
$$

and the position space tree-level propagators are

$$
\begin{aligned}
\left\langle f'(z_1) f'(z_2) \right\rangle_0 &= \quad\text{————————}\quad = -\frac{8G}{z_{12}^2}, \\
\left\langle \bar{f}'(\bar{z}_1) \bar{f}'(\bar{z}_2) \right\rangle_0 &= \quad\text{- - - - - - - - -}\quad = -\frac{8G}{\bar{z}_{12}^2}.
\end{aligned}
\tag{212}
$$

One can then use the Feynman rules of the fundamental fields derived in [60] to calculate the deformed stress tensor correlators. The Feynman rules for computing an $n$-point function of some combination of the fundamental fields $(f', \bar{f}')$ are as follows. Each propagator has a $G$ while each vertex has a $\frac{1}{G}$. For instance, a Feynman diagram with $n_p$ propagators, $n_v$ quartic vertices and $n_e$ external $(f', \bar{f}')$ lines has dependence $G^{n_p - n_v}$. Therefore, we have

$$
2n_p = n_e + 4n_v
\tag{213}
$$

and

$$
\langle f'(x_1) \cdots f'(x_{n_e}) \rangle \sim G^{\frac{n_e}{2} + n_v} \sim G^{n_p - n_v}.
\tag{214}
$$

Intuitively, the quantum corrections to the deformed gravitational Wilson line $\langle W[z_2, z_1] \rangle_\lambda$ involve non-vanishing self-energy interactions between both holomorphic and anti-holomorphic

exchanges of the fundamental fields denoted by solid and dashed propagators respectively.[15]

Determining the deformed quantum Wilson line at a given order in $\lambda$ and $\frac{1}{c}$ is computationally complicated for two reasons. The first reason is that the $n$-point stress tensor correlator is subject to both quantum corrections in $\frac{1}{c}$ and $\lambda$ corrections. To be more precise, we notice that the expectation value of the undeformed Wilson line $W_\varepsilon[z,0]$ has the expansion

$$\langle W_\varepsilon[z;0]\rangle_0 = z^{2j}N(\varepsilon)\sum_{n=0}^{\infty}\frac{(6\alpha(\varepsilon))^n}{c^n}\int_0^z dy_n\cdots\int_0^{y_2}dy_1 F_n(z;y_n,\ldots,y_1)\langle T_{zz}(y_n)\cdots T_{zz}(y_1)\rangle_0,$$
(215)

from (204). Here the $SL(2,\mathbb{R})$ group theory factor $F_n(z;y_n,\ldots,y_1)$ is defined by the following homogeneous polynomial in the variables $z,y_n,\cdots,y_1$ of degree $n$:

$$z^{2j}F_n(z;y_n,\ldots,y_1) = \langle j,-j \mid e^{zL_1}\left(L_{-1}-2y_nL_0+y_n^2L_1\right)\cdots\left(L_{-1}-2y_1L_0+y_1^2L_1\right)\mid j,j\rangle.$$
(216)

Computing $\langle W_\varepsilon[z;0]\rangle_\lambda$ via conformal perturbation theory in $\lambda$ involves an infinite $\lambda$ expansion at each order of the $O\left(\frac{1}{c}\right)$ expansion of the undeformed Wilson line $\langle W_\varepsilon[z;0]\rangle_0$. For instance, the $O\left(\frac{1}{c^2}\right)$ term in the $\frac{1}{c}$ expansion[16] of $\langle W_\varepsilon[z;0]\rangle_0$ is

$$z^{2j}N(\varepsilon)\frac{(6\alpha(\varepsilon))^2}{c^2}\int_0^z dy_1\int_0^{y_2}dy_1 \quad F_2(z;y_1,y_2)\langle T_{zz}(y_1)T_{zz}(y_2)\rangle_0$$

$$\to z^{2j}N(\varepsilon)\frac{(6\alpha(\varepsilon))^2}{c^2}\sum_{p=0}^{\infty}\int_0^z dy_1\int_0^{y_2} \quad F_2(z;y_1,y_2)\times$$
(217)

$$\left\langle T_{zz}(y_1)T_{zz}(y_2)\frac{\lambda^p}{p!}\left(\int d^2w\, T_{zz}(w)T_{\bar{z}\bar{z}}(\overline{w})\right)^p\right\rangle.$$

While computing these deformed correlators in general is computationally difficult, it is straightforward to work out the leading scaling in $c$ at a general order[17] $p$ of the $\lambda$-expansion using the results [9]. Doing this, one finds

$$\lambda^p\left\langle T_{zz}^{p+2}T_{\bar{z}\bar{z}}^p\right\rangle_0 \sim \lambda^p c^{\lfloor\frac{p+2}{2}\rfloor}c^{\lfloor\frac{p}{2}\rfloor}\left[\cdots+O\left(\frac{1}{c}\right)\right]$$

$$\lesssim \lambda^p c^{p+1}\left[\cdots+O\left(\frac{1}{c}\right)\right],$$
(218)

where the ellipsis denotes terms independent of $\lambda$ and $c$.

In order to have a controlled perturbative expansion, we must have $\lambda c < 1$ in addition to $c\to\infty$:

$$\lambda c = \frac{4Gr_c}{\pi}\frac{3}{2G} = \frac{6r_c}{\pi} < 1.$$
(219)

Again, we work in units where $\ell_{\mathrm{AdS}_3} = 1$ so that the combination $\lambda c$ is dimensionless. As one can check, this condition ensures that for each fixed $n$ in the original $c$ expansion, the expansion via conformal perturbation theory is well-defined if $\lambda c < 1$:

$$\lambda^p\langle T_{zz}^{p+n}T_{\bar{z}\bar{z}}^p\rangle_0 \sim \lambda^p c^{\lfloor\frac{p+n}{2}\rfloor}c^{\lfloor\frac{p}{2}\rfloor}\left[\cdots+O\left(\frac{1}{c}\right)\right],$$
(220)

---

[15]See (223) for an example of a typical Feynman diagram that we will use to compute the leading order correction to the deformed Wilson line. Holomorphic propagators represent $\langle f'(z_1)f'(z_2)\rangle$ and anti-holomorphic propagators represent $\langle \bar{f}'(\bar{z}_1)\bar{f}'(\bar{z}_2)\rangle$.

[16]We start with $O\left(\frac{1}{c^2}\right)$ because at $O\left(\frac{1}{c}\right)$, the one-point planar correlator $\langle T_{zz}(y_1)\rangle_\lambda$ vanishes identically by Lorentz and translational invariance. In the case of non-planar backgrounds, such as the cylinder or torus [105,106], the one-point function is non-zero.

[17]Notice that for small $p$ and $n$, there are special cases where the leading order term in $c$ vanishes.

where

$$\lambda^p c^{\lfloor \frac{p+n}{2} \rfloor} c^{\lfloor \frac{p}{2} \rfloor} = \begin{cases} (\lambda c)^p \cdot c^{\frac{n-1}{2}} & n \in 2\mathbb{Z}_{\geq 0} + 1 \\ \lambda^p c^{2\lfloor \frac{p}{2} \rfloor + \frac{n}{2}} \leq (\lambda c)^p c^{\frac{n+2}{2}} & n \in 2\mathbb{Z}_{>0} \end{cases}. \tag{221}$$

Similarly, there is another expansion from the linear mixing which introduces $\lambda \overline{T}$ to the coefficient of $L_1$. In the $O(c)$ analysis from $\overline{T}$, we can roughly assign $\sqrt{c}$ to $\overline{T}$ since $\langle \overline{T}^n \rangle \sim c^{\lfloor \frac{n}{2} \rfloor}$. Then we see the condition above where $\lambda \lesssim \frac{1}{c}$ automatically guarantees this expansion is well-defined in the large-$c$ limit.

The second reason is that the divergences from the integrals are handled by the dimensional regularization scheme, which we mentioned in the discussion of the renormalized vertex factor and multiplicative renormalization around equation (204). For the sake of exposition, we content ourselves with determining the leading order corrections to the quantum Wilson line (208). Expanding the exponential (208), we find

$$\langle W[z_2, z_1] \rangle_\lambda$$
$$= z^{2j} \Bigg[ 1 + \left( \frac{6}{c} \right)^2 \int_{z_1}^{z_2} dy_1 \int_{z_1}^{y_2} dy_2 \langle j, -j | \exp(L_1 z_{21}) \left( L_{-1} - 2(y_1 - z_1)L_0 + (y_1 - z_1)^2 L_1 \right)$$
$$\times \left( L_{-1} - 2(y_2 - z_1)L_0 + (y_2 - z_1)^2 L_1 \right) | j, j \rangle \langle T_{zz}(y_1) T_{zz}(y_2) \rangle_\lambda \Bigg]. \tag{222}$$

The tree-level deformed planar stress tensor two-point function at $O(\lambda^2 c^2)$ was determined first by [50] via translational/rotational invariance and stress tensor conservation. Alternatively, using the approach of [60], this is easily understood from the propagators of the fundamental fields (211) as[18]

$$\langle T_{zz}(y_1) T_{zz}(y_2) \rangle_\lambda = \frac{1}{(8G)^2} \partial_{y_1} \partial_{y_2} \langle f'(y_1) f'(y_2) \rangle_0 +$$
$$\frac{r_c^2}{(16G)^2} \left( \partial_{y_1}^2 \partial_{y_2}^2 \langle f'(y_1) f'(y_2) \rangle_0 \right) \langle \overline{f}'(\overline{y}_1) \overline{f}'(\overline{y}_2) \rangle_0$$
$$= \frac{1}{(8G)^2} \partial_{y_1} \partial_{y_2} \underline{\qquad} + \frac{r_c^2}{(16G)^2} \cdot \left( \partial_{y_1}^2 \partial_{y_2}^2 \underline{\qquad} \right) \cdot (\,\text{-----}\,)$$
$$= \frac{c}{2(y_1 - y_2)^4} + \frac{5\pi^2 \lambda^2 c^2}{6(y_1 - y_2)^6 (\overline{y}_1 - \overline{y}_2)^2}. \tag{223}$$

To further simplify (222), we use commutation relations to arrange the generators in the normal order $(L_1)^{n_1}(L_0)^{n_0}(L_{-1})^{n_{-1}}$ and use the fact that $L_{-1} | j, j \rangle = 0$ and $L_0 | j, j \rangle = 0$. After doing this, the only non-zero matrix elements arise from terms proportional to $(L_1)^{2j}$:

$$\langle j, -j | \exp(L_1 z) \left( L_{-1} - 2y_1 L_0 + y_1^2 L_1 \right) \left( L_{-1} - 2y_2 L_0 + y_2^2 L_1 \right) | j, j \rangle$$
$$= z^{2j-2} \left[ 2j y_2 (z - y_1)(2j y_1 (z - y_2) - y_2 (z - y_1)) \right]. \tag{224}$$

The integral (222) reduces to

$$\langle W[z, 0] \rangle_\lambda = z^{2j} \Bigg[ 1 + \frac{36j}{c z^2} \int_0^z dy_1 \int_0^{y_1} dy_2 \frac{y_2 (z - y_1)(2j y_1 (z - y_2) - y_2 (z - y_1))}{(y_1 - y_2)^4}$$
$$+ \frac{60\pi^2 \lambda^2 j}{z^2} \int_0^z dy_1 \int_0^{y_1} dy_2 \frac{y_2 (z - y_1)(2j y_1 (z - y_2) - y_2 (z - y_1))}{(y_1 - y_2)^6 (\overline{y}_1 - \overline{y}_2)^2} \Bigg], \tag{225}$$

---

[18] We use $r_c = \frac{\pi \lambda}{4G} = \frac{\pi \lambda c}{6}$ and $c = \frac{3}{2G}$ for the tree-level deformed stress tensor correlators.

which clearly diverges when $y_2 \to y_1$ or $\bar{y}_2 \to \bar{y}_1$.

We dimensionally regularize the stress tensor correlators to evaluate these divergent integrals (225). The $O(\frac{\lambda^0}{c})$ integral has already been evaluated in [9] via dimensional regularization, which gives

$$\frac{36j}{cz^2} \int_0^z dy_1 \int_0^{y_1} dy_2 \frac{y_2(z-y_1)(2jy_1(z-y_2)-y_2(z-y_1))}{(y_1-y_2)^4} = \frac{12j(j+1)}{c} \ln z. \tag{226}$$

To deal with the $O(\lambda^2 c^0)$ integral in (225), we first specify an integration contour in the complex plane that is a straight line along the direction towards $z$:

$$\begin{aligned} y_2 &= y_1 t, \quad 0 \le t \le 1, \\ y_1 &= zT, \quad 0 \le T \le 1, \end{aligned} \tag{227}$$

and we find

$$\begin{aligned} &\frac{60\pi^2\lambda^2 j}{z^2} \int_0^z dy_1 \int_0^{y_1} dy_2 \frac{y_2(z-y_1)(2jy_1(z-y_2)-y_2(z-y_1))}{(y_1-y_2)^6(\bar{y}_1-\bar{y}_2)^2} \\ &= \frac{60\pi^2\lambda^2 j}{|z|^4} \int_0^1 dT \frac{1-T}{T^5} \int_0^1 dt \frac{2j(t-t^2T)-t^2(1-T)}{(1-t)^8}. \end{aligned} \tag{228}$$

The above integral is evaluated via dimensional regularization:

$$\begin{aligned} &\frac{60\pi^2\lambda^2 j}{|z|^4} \int_0^1 dT \frac{1-T}{T^{5-2\varepsilon}} \int_0^1 dt \frac{2j(t-t^2T)-t^2(1-T)}{(1-t)^{8-2\varepsilon}} \\ &= \frac{\pi^2 j(9j-1)\lambda^2}{21|z|^4}. \end{aligned} \tag{229}$$

In summary, the leading order correction to the Wilson line is

$$\langle W[z,0] \rangle_\lambda = z^{2j} \left( 1 + \frac{12j(j+1)}{c} \ln z + \frac{\pi^2 j(9j-1)\lambda^2}{21|z|^4} \right). \tag{230}$$

An alternative renormalization approach, which yields the same numerical coefficient as (229), is to introduce cutoffs $\varepsilon_1$ and $\varepsilon_2$ as

$$\frac{60\pi^2\lambda^2 j}{|z|^4} \int_{\varepsilon_2}^1 dT \frac{1-T}{T^5} \int_0^{1-\varepsilon_1} dt \frac{2j(t-t^2T)-t^2(1-T)}{(1-t)^8}, \tag{231}$$

and perform "minimal subtraction" to remove the divergent terms and then set $\varepsilon_1 = \varepsilon_2 = 0$. Evaluating (231) gives

$$\frac{\pi^2 j(9j-1)\lambda^2}{21|z|^4}. \tag{232}$$

## 5.4  AdS$_3$ Wilson Line Correlators

The correlator involving the product of a holomorphic and anti-holomorphic Wilson line can be thought of as a scalar correlator. A useful sanity check is to compute this Wilson line product correlator and see if it is consistent with $T\bar{T}$-deformed scalar correlators [50, 51, 53]. We use

conformal perturbation theory at $O(\lambda)$ to find

$$
\begin{aligned}
\langle W[z,0]\overline{W}[\bar{z},0]\rangle_\lambda &= \left\langle W[z,0]\overline{W}[\bar{z},0]\exp\left(\lambda\int d^2w\, T_{zz}(w)T_{\bar{z}\bar{z}}(\bar{w})\right)\right\rangle \\
&= \langle W[z,0]\overline{W}[\bar{z},0]\rangle_0 + \lambda\int d^2w\,\langle T_{zz}(w)T_{\bar{z}\bar{z}}(\bar{w})W[z,0]\overline{W}[\bar{z},0]\rangle_0 \\
&= |z|^{-4h(j)} + \lambda\int d^2w\,\langle T_{zz}(w)W[z,0]\rangle_0\,\langle T_{\bar{z}\bar{z}}(\bar{w})\overline{W}[\bar{z},0]\rangle_0 \\
&= |z|^{-4h(j)} + \lambda h^2|z|^4\int\frac{d^2w}{|w|^4|w-z|^4}\,\langle W[z,0]\rangle_0\langle\overline{W}[\bar{z},0]\rangle_0 \\
&= |z|^{-4h} + \lambda h^2|z|^4\int\frac{d^2w}{|w|^4|w-z|^4}|z|^{-4h} \\
&= |z|^{-4h(j)}\left(1 + \lambda h^2|z|^4\mathcal{I}_{2222}(0,z,0,\bar{z})\right).
\end{aligned}
\tag{233}
$$

Here we have used the Ward identity in [10], namely

$$
\begin{aligned}
\langle T_{zz}(w)W[z,0]\rangle_0 &= \frac{h(j)z^2}{(z-w)^2\,w^2}\,\langle W[z,0]\rangle_0\,, \\
\langle T_{\bar{z}\bar{z}}(\bar{w})\overline{W}[\bar{z},0]\rangle_0 &= \frac{h(j)\bar{z}^2}{(\bar{z}-\bar{w})^2\,\bar{w}^2}\,\langle\overline{W}[\bar{z},0]\rangle_0\,,
\end{aligned}
\tag{234}
$$

which displays the bi-local structure of the gravitational Wilson line.

From Appendix A in [53], the integral (233) is of the form

$$
\mathcal{I}_{a_1,\cdots,a_m,b_1,\cdots,b_n}\left(z_{i_1},\cdots,z_{i_m},\bar{z}_{j_1},\cdots,\bar{z}_{j_n}\right) = \int\frac{d^2z}{\prod_{k=1}^m\left(z-z_{i_k}\right)^{a_k}\prod_{p=1}^n\left(\bar{z}-\bar{z}_{j_p}\right)^{b_p}}\,,
\tag{235}
$$

and is evaluated via dimensional regularization. In particular,

$$
\begin{aligned}
\mathcal{I}_{2222}(0,z,0,\bar{z}) &= \int\frac{d^2w}{|w|^4|w-z|^4} \\
&= \frac{4\pi}{|z|^6}\left(\frac{4}{\varepsilon} + 2\ln|z|^2 + 2\ln\pi + 2\gamma - 5\right) \\
&= \frac{1}{|z|^6}\left(C_1 + C_2\ln|z|^2\right),
\end{aligned}
\tag{236}
$$

where the constants are

$$
C_1 = 8\pi\ln\pi - 20\pi + 8\pi\gamma\,, \quad C_2 = 8\pi\,.
\tag{237}
$$

We arrive at

$$
\langle W[z,0]\overline{W}[\bar{z},0]\rangle_\lambda = |z|^{-4h(j)}\left(1 + \frac{\lambda h(j)^2\left(C_1 + C_2\ln|z|^2\right)}{|z|^2}\right),
\tag{238}
$$

which exactly matches what we expect at $O(\lambda c^0)$ from previous analyses of $T\overline{T}$-deformed scalar correlators [50, 51, 53]. This confirms the claim that the correlator of two Wilson lines behaves as a scalar correlator, at least at this order.

Additionally, at leading order in $\lambda$ and in the large-$c$ limit, (233) agrees with the structure one would expect from the linear mixing of sources and expectation values discussed above. Schematically, at the leading order,

$$\left\langle P \exp\left[\int_0^z dy \left(e_i(\lambda)L_1 + \frac{6}{c}T_{zz}(y)L_{-1}\right)\right] P \exp\left[\int_0^{\bar{z}} d\bar{y}\left(\bar{e}_i(\lambda)\bar{L}_1 + \frac{6}{c}T_{\bar{z}\bar{z}}(\bar{y})\bar{L}_{-1}\right)\right]\right\rangle_\lambda$$

$$= \left\langle P \exp\left[\int_0^z dy \left((1 + \lambda T_{\bar{z}\bar{z}}(y))L_1 + \frac{6}{c}T_{zz}(y)L_{-1}\right)\right]\right.$$

$$\left.\cdot P \exp\left[\int_0^{\bar{z}} d\bar{y}\left((1 + \lambda T_{zz}(y))\bar{L}_1 + \frac{6}{c}T_{\bar{z}\bar{z}}(\bar{y})\bar{L}_{-1}\right)\right]\right\rangle_\lambda$$

$$= \langle W[z,0]\rangle_0 \langle \overline{W}[\bar{z},0]\rangle_0 + \lambda\langle\exp(zL_1)\,L_1\rangle\int_0^z dy\left\langle T_{\bar{z}\bar{z}}(\bar{y})\overline{W}[\bar{z},0]\right\rangle_0 + O(\lambda^2)$$

$$= |z|^{-4h(j)} + \lambda\partial_z\langle\exp(zL_1)\rangle\int_0^z dy\left\langle T_{\bar{z}\bar{z}}(\bar{y})\overline{W}[\bar{z},0]\right\rangle_0 + O(\lambda^2)$$

$$= |z|^{-4h(j)} - 2\lambda h(j)z^{-2h(j)-1}\int_0^z dy\,\frac{h(j)\bar{z}^2}{(\bar{z}-\bar{y})^2\bar{y}^2}\langle\overline{W}[\bar{z},0]\rangle_0 + O(\lambda^2)$$

$$= |z|^{-4h(j)} - 2\lambda h(j)^2 z^{-2h(j)-1}\bar{z}^{-2h(j)+2}\int_0^z \frac{dy}{(\bar{z}-\bar{y})^2\bar{y}^2} + O(\lambda^2)$$

$$= |z|^{-4h(j)}\left(1 + \lambda h(j)^2 z^{-1}\bar{z}^2\left(\frac{c_1 + c_2\ln|z|^2}{\bar{z}^3}\right) + O(\lambda^2)\right)$$

$$= |z|^{-4h(j)}\left(1 + \lambda h(j)^2\left(\frac{c_1 + c_2\ln|z|^2}{|z|^2}\right) + O(\lambda^2)\right), \tag{239}$$

where in the large-$c$ limit, the quantum corrections to the Wilson line's scaling dimension $h(j) = -j$ are suppressed and $\langle\exp(zL_1)\rangle = z^{-2h} = z^{2j}$. The integral in (239) may be evaluated via the integration cutoff introduced in (231) or by dimensional regularization. Using either method, one finds that the result has a similar structure as (238), where $C_1$ and $C_2$ are constant coefficients.

We emphasize that if one had not used the linear mixing or conformal perturbation theory, but rather expanded each path-ordered exponential in $\langle W[z_2, z_1]\overline{W}[\bar{z}_2, \bar{z}_1]\rangle_\lambda$, then the leading contribution in $\lambda$ would be at $O(\lambda^2 c^0)$, which arises from integrating the tree-level deformed stress tensor two-point function. To see this, let us compute the correction

$$\delta\langle W[z,0]\overline{W}[\bar{z},0]\rangle_\lambda = \langle W[z,0]\overline{W}[\bar{z},0]\rangle_\lambda - \langle W[z,0]\overline{W}[\bar{z},0]\rangle_0 \tag{240}$$

to the correlator using this prescription. We expand the path-ordered exponential[19] for $W[z,0]$ and $\overline{W}[\bar{z},0]$ up to $O(1/c)$ in (208), which gives

$$\left(\frac{6}{c}\right)^2\int_0^z dy_1\int_0^{\bar{z}} d\bar{y}_2\langle j, -j\,|\,e^{L_1 z}(L_{-1} - 2y_1L_0 + y_1^2)e^{\bar{L}_1\bar{z}}(\bar{L}_{-1} - 2\bar{y}_2\bar{L}_0 + \bar{y}_2^2)\,|\,j,j\rangle\times$$
$$\langle T_{zz}(y_1)T_{\bar{z}\bar{z}}(y_2)\rangle_\lambda. \tag{241}$$

---

[19]For the single Wilson line (225), we expanded up to $O(1/c^2)$ in the path-ordered exponential since the planar one-point function vanishes. At $O(1/c)$, the path-ordered exponential reduces to a regular integral.

Using (211) and the Feynman rules for the relevant tree diagrams,

$$
\begin{aligned}
\langle T_{zz}(y_1)T_{\bar{z}\bar{z}}(y_2)\rangle_\lambda &= \frac{r_c^2}{(16G)^2}\left(\partial_{y_1}^2\langle f'(y_1)f'(y_2)\rangle_0\right)\left(\partial_{\bar{y}_2}^2\langle \bar{f}'(\bar{y}_1)\bar{f}'(\bar{y}_2)\rangle_0\right)+O(r_c^3) \\
&= \frac{r_c^2}{(16G)^2}\cdot\left(\partial_{y_1}^2 \,\underline{\qquad}\,\right)\cdot\left(\partial_{\bar{y}_2}^2 \,\text{-------}\,\right)+O(r_c^3) \\
&= \frac{\pi^2\lambda^2 c^2}{4}\frac{1}{(y_1-y_2)^4(\bar{y}_1-\bar{y}_2)^4}+O(\lambda^3).
\end{aligned}
\tag{242}
$$

Thus the above prescription involving correlators of Wilson lines (241) is incorrect because the leading correction enters at $O(\lambda^2 c^0)$, rather than the expected order of $O(\lambda c^0)$ for scalar two-point correlators (cf. (199), (238) and (239)).

Furthermore, one may also consider a string of holomorphic and anti-holomorphic stress tensor insertions in correlators involving a Wilson line. For instance, we can calculate these kind of correlators via conformal perturbation theory at $O(\lambda)$:

$$
\langle T_{zz}(w_1)T_{\bar{z}\bar{z}}(\bar{w}_2)W[z,0]\rangle_\lambda = \lambda\int d^2 y\,\langle T_{zz}(y)T_{zz}(w_1)W[0,z]\rangle_0\,\langle T_{\bar{z}\bar{z}}(\bar{y})T_{\bar{z}\bar{z}}(\bar{w}_2)\rangle_0\,.
\tag{243}
$$

In [10], the following tree-level correlator to $O(1/c^0)$ was derived:

$$
\langle T_{zz}(w_1)T_{zz}(w_2)W[z,0]\rangle_0 = \frac{j^2 z^{2j+4}}{w_1^2(z-w_1)^2 w_2^2(z-w_2)^2}+ \\
\frac{jz^{2j+2}}{w_1(z-w_1)w_2(z-w_2)(w_1-w_2)^2}\,,
\tag{244}
$$

in agreement with the predictions from the conformal Ward identities. Therefore, using the fact that

$$
\langle T_{\bar{z}\bar{z}}(\bar{y})T_{\bar{z}\bar{z}}(\bar{w}_2)\rangle_0 = \frac{c}{2(\bar{y}-\bar{w}_2)^4}\,,
\tag{245}
$$

the integral (243) is reduced to

$$
\langle T_{zz}(w_1)T_{\bar{z}\bar{z}}(\bar{w}_2)W[z,0]\rangle_\lambda = \frac{cj\lambda z^{2j}}{2}\int d^2 y\left[\frac{jz^4}{y^2(y-z)^2 w_1^2(z-w_1)^2(\bar{y}-\bar{w}_2)^4}\right. \\
\left.-\frac{z^2}{y(y-z)w_1(z-w_1)(y-w_1)^2(\bar{y}-\bar{w}_2)^4}\right],
\tag{246}
$$

and is evaluated in terms of the integrals defined in (235):

$$
\langle T_{zz}(w_1)T_{\bar{z}\bar{z}}(\bar{w}_2)W[z,0]\rangle_\lambda = \frac{j\lambda c z^{2j+2}}{2w_1(z-w_1)}\left[\frac{jz^2}{w_1(z-w_1)}\mathcal{I}_{224}(0,z,\bar{w}_2)-\mathcal{I}_{1124}(0,z,w_1,\bar{w}_2)\right].
\tag{247}
$$

Another example is a correlator involving two insertions of anti-holomorphic stress tensors, a holomorphic stress tensor, and a holomorphic Wilson line. The desired correlator

$$
\left\langle T_{zz}(w_1)T_{\bar{z}\bar{z}}(\bar{w}_2)T_{\bar{z}\bar{z}}(\bar{w}_3)W[0,z]\exp\left(\lambda\int d^2 y\,T_{zz}(y)T_{\bar{z}\bar{z}}(\bar{y})\right)\right\rangle
\tag{248}
$$

is easily computable at $O(\lambda)$ via conformal perturbation theory.

Noting that the undeformed tree-level planar three-point stress tensor correlator is

$$
\langle T_{\bar{z}\bar{z}}(\bar{y})T_{\bar{z}\bar{z}}(\bar{w}_2)T_{\bar{z}\bar{z}}(\bar{w}_3)\rangle_0 = \frac{c}{(\bar{y}-\bar{w}_2)^2(\bar{w}_2-\bar{w}_3)^2(\bar{w}_3-\bar{y})^2}\,,
\tag{249}
$$

the leading order correction to the integral (248) at $O(\lambda c)$ is

$$
\begin{aligned}
&\langle T_{zz}(w_1) T_{\bar z \bar z}(\bar w_2) T_{\bar z \bar z}(\bar w_3) W[z,0]\rangle_\lambda \\
&= \langle T_{zz}(w_1) W[z,0]\rangle_0 \langle T_{\bar z \bar z}(\bar w_2) T_{\bar z \bar z}(\bar w_3)\rangle_0 \\
&\quad + \lambda \int d^2 y \; \langle T_{zz}(y) T_{zz}(w_1) W[0,z]\rangle_0 \langle T_{\bar z \bar z}(\bar y) T_{\bar z \bar z}(\bar w_2) T_{\bar z \bar z}(\bar w_3)\rangle_0 \\
&= \frac{h(j)z^2}{(z-w_1)^2 w_1^2} \langle W[z,0]\rangle_0 \frac{c}{2(\bar w_2 - \bar w_3)^4} \\
&\quad + c j \lambda z^{2j} \int d^2 y \left[ \frac{j z^4}{y^2(y-z)^2 w_1^2 (z-w_1)^2 (\bar y - \bar w_2)^2 (\bar w_2 - \bar w_3)^2 (\bar w_3 - \bar y)^2} \right. \\
&\qquad\qquad \left. - \frac{z^2}{y(y-z)w_1 (z-w_1)(y-w_1)^2 (\bar y - \bar w_2)^2 (\bar w_2 - \bar w_3)^2 (\bar w_3 - \bar y)^2} \right].
\end{aligned}
$$
(250)

Evaluating (250) in terms of the integrals defined in (235), we find

$$
\begin{aligned}
&\langle T_{zz}(w_1) T_{\bar z \bar z}(\bar w_2) T_{\bar z \bar z}(\bar w_3) W[z,0]\rangle_\lambda \\
&= \frac{h(j)c z^{2-2h(j)}}{2(z-w_1)^2 w_1 (\bar w_2 - \bar w_3)^4} \\
&\quad + \frac{j \lambda c z^{2j+2}}{w_1(z-w_1)(\bar w_2 - \bar w_3)^2} \left[ \frac{j z^2}{w_1(z-w_1)} \mathcal{I}_{2222}(0,z,\bar w_2,\bar w_3) - \mathcal{I}_{11222}(0,z,w_1,\bar w_2,\bar w_3) \right].
\end{aligned}
$$
(251)
(252)

The integrals presented here, which are of the general form given in (235) but with higher-valued indices, can be expressed in terms of derivatives and linear combinations of known integrals with lower-valued indices. See Appendix A in [53] for several detailed examples.

One can automate the above perturbative analysis in $\lambda$ to produce more complicated expressions for correlators involving products of $m$-insertions of holomorphic stress tensors, $n$-insertions of anti-holomorphic stress tensors, and a network of Wilson lines (e.g., $p$-insertions of the holomorphic Wilson line and $q$-insertions of the anti-holomorphic Wilson line) following [10]. The leading correction for such a general correlator takes the form

$$
\begin{aligned}
&\left\langle \prod_{i=1}^{m} T_{zz}(x_i) \prod_{j=1}^{n} T_{\bar z \bar z}(\bar w_j) \prod_{k=1}^{p} W[z_{k+1},z_k] \prod_{l=1}^{q} \overline{W}[\bar r_{l+1}, \bar r_l] \exp\left( \lambda \int d^2 y \; T_{zz}(y) T_{\bar z \bar z}(\bar y) \right) \right\rangle \\
&= \left\langle \prod_{i=1}^{m} T_{zz}(x_i) \prod_{k=1}^{p} W[z_{k+1},z_k] \right\rangle_0 \left\langle \prod_{j=1}^{n} T_{\bar z \bar z}(\bar w_j) \prod_{l=1}^{q} \overline{W}[\bar r_{l+1}, \bar r_l] \right\rangle_0 \\
&\quad + \lambda \int d^2 y \left\langle T_{zz}(y) \prod_{i=1}^{m} T_{zz}(x_i) \prod_{k=1}^{p} W[z_{k+1},z_k] \right\rangle_0 \left\langle T_{\bar z \bar z}(\bar y) \prod_{j=1}^{n} T_{\bar z \bar z}(\bar w_j) \prod_{l=1}^{q} \overline{W}[\bar r_{l+1}, \bar r_l] \right\rangle_0 \\
&\quad + O(\lambda^2).
\end{aligned}
$$
(253)

# 6 Gravitational BF Wilson Lines

To complete our study of Wilson lines in low dimensional gravity, we conclude by investigating Wilson lines and their correlators in BF theory under the $T\overline{T}$ deformation.

We first study the boundary spectrum of the BF theory under the $T\overline{T}$ deformation. In the undeformed theory, between the two classes of irreducible representations of the gauge group $SL(2,\mathbb{R})$ with normalizable characters [18], the principal series dominates over the discrete series. This domination leads to the spectrum of the Schwarzian theory and matches with the result from the metric formalism of JT gravity.

Following the same treatment in the deformed BF theory, we find the principal series remains dominant compared to the discrete series *below* the Hagedorn transition temperature, defined after (273).[20] The deformed theory's dynamics are captured by the $T\overline{T}$-deformed Schwarzian theory. We find that above the Hagedorn transition temperature the discrete series dominates over the principal series, implying the boundary theory's dynamics are no longer captured by the $T\overline{T}$-deformed Schwarzian theory. The correct description above the transition temperature should be some effective field theory which captures the discrete series contribution. This is consistent with the fact that the deformed partition function of the Schwarzian theory diverges at the Hagedorn temperature indicating the breakdown of the deformed Schwarzian description. Our analysis provides a glimpse into what happens across the Hagedorn transition, and understanding the entire phase diagram is an interesting and important future direction.

We then move on to study Wilson lines in BF theory. Due to the subtleties mentioned above, we will only study the correlators of Wilson lines *below* the Hagedorn transition temperature where the boundary theory is captured by the deformed Schwarzian theory.

Just as the Wilson line in $3d$ Chern-Simons is conjectured to transform as a bi-local primary operator at its endpoints, a Wilson line in $2d$ BF theory in representation $\eta$ is also believed to transform as a bi-local primary. We indicate this schematically by writing

$$\langle W_\eta[\mathcal{C}_{\tau_1,\tau_2}]\rangle \longleftrightarrow \langle O_\eta(\tau_2)O_\eta(\tau_1)\rangle,\tag{254}$$

for a boundary-anchored path $\mathcal{C}_{\tau_1,\tau_2}$ on the disk $D$ which intersects the string defect $L$ mentioned in Section 3.1 at points $\tau_1$ and $\tau_2$.

In the context of $3d$ gravitational Chern-Simons theory, we saw that Wilson line operators admitted interpretations from both the bulk perspective and the boundary perspective. Similar interpretations exist in the case of BF theory. In the $1d$ boundary theory, we can view the bi-local operator (254) as a two-point function for an operator $O_\eta$ in some matter CFT which has been coupled to the Schwarzian theory. From the viewpoint of the $2d$ bulk, the Wilson line computes a certain path integral

$$W_\eta[\mathcal{C}_{\tau_1,\tau_2}] \cong \int_{\text{paths}\sim\mathcal{C}_{\tau_1,\tau_2}} \mathcal{D}x \, \exp\left(-m\int_{\mathcal{C}_{\tau_1,\tau_2}} ds\sqrt{g_{\mu\nu}\frac{dx^\mu}{ds}\frac{dx^\nu}{ds}}\right),\tag{255}$$

involving the action for a probe particle coupled to gravity with mass $m^2 = \eta(\eta-1) = -C_2(\eta)$. The right-hand-side of (255) is a functional integral weighted by the point particle action over all paths $x(s)$ diffeomorphic to $\mathcal{C}_{\tau_1,\tau_2}$.

## 6.1 Single BF Wilson Line

Motivated by (255), one notices that the basic building block for constructing the gravitational Wilson line in BF theory is the disk partition function [18, 98]. We therefore would like to develop a formulation of the $T\overline{T}$ deformation which is convenient for computing these disk partition functions. We first illustrate this formalism for compact groups and then generalize to the non-compact group $SL(2,\mathbb{R})$.

---

[20]See [107, 108] for detailed discussions on the thermodynamics of $T\overline{T}$-deformed $1d$ quantum mechanical systems.

To produce the deformed theory whose seed is given by (66), we consider deforming the boundary term $\oint_L du\, V(\phi(u))$, where $V(\phi(u)) = v\,\mathrm{Tr}\,\phi^2$ with a constant $v$,

$$\oint_L du\, V(\phi(u)) \longrightarrow \oint_L du\, V_\lambda(\phi(u)), \quad V_\lambda(\phi(u)) = \frac{1-\sqrt{1-8\lambda V(\phi)}}{4\lambda}. \tag{256}$$

To compute the disk partition function with a fixed holonomy $g = P\exp\left(\oint A\right)$, we fix the boundary value of $A_\tau$ accordingly such that (65) vanishes to guarantee a well-defined variational principle. It is important to note that the string defect $L$ supporting the potential $\oint du\, V(\phi)$ is arbitrarily close to the boundary, but not actually *on* the boundary, so that no new boundary terms appear and spoil the variational principle emphasized in [18].

For a compact group with a generic potential $V(\phi)$, the disk partition function has been computed in [18]. Specializing to the potential $V_\lambda(\phi)$ in (256), the partition function is

$$Z_\lambda(g, v, \beta) = \sum_R (\dim R)\, \chi_R(g) \exp\left(-\beta f_\lambda^-\left(\frac{v C_2(R)}{N}\right)\right), \tag{257}$$

where $f_\lambda^-\left(\frac{v C_2(R)}{N}\right)$ is the negative branch of the deformed JT gravity's spectrum [82,83]

$$f_\lambda^-\left(\frac{v C_2(R)}{N}\right) = \frac{1-\sqrt{1-8\lambda v C_2(R)/N}}{4\lambda}, \tag{258}$$

$C_2(R)$ is the quadratic Casimir in the representation $R$ and $\chi_R(g)$ is a character serving as a wavefunction in canonical quantization. Compared to the formula for a $2d$ YM partition function [32,34], the absence of surface area in the exponent in (257) shows that the BF theory is truly topological.

Taking the boundary holonomy $g \to \mathbb{I}$, we recover the partition function of the $T\bar{T}$-deformed quantum mechanics describing a particle-on-a-group[21]

$$Z_\lambda(\mathbb{I}, v, \beta) = \sum_R (\dim R)^2 \exp\left(-\beta f_\lambda^-\left(\frac{v C_2(R)}{N}\right)\right). \tag{259}$$

We next work out the generalization of the above to non-compact groups following [18]. We first choose the gauge group to be

$$\mathcal{G}_B = \frac{\widetilde{SL}(2,\mathbb{R}) \times \mathbb{R}}{\mathbb{Z}}, \tag{260}$$

as in [18]. The identification associated to the quotient $\mathbb{Z}$ is given by

$$(\tilde{g}, \theta) \sim (h_n \tilde{g}, \theta + Bn), \tag{261}$$

where $\tilde{g} \in \widetilde{SL}(2,\mathbb{R})$, the universal cover[22] of $SL(2,\mathbb{R})$, $\theta \in \mathbb{R}$, $h_n$ is the $n$-th element of $\mathbb{Z} \subset \widetilde{SL}(2,\mathbb{R})$, and $B \in \mathbb{R}$ defines the extension.

The irreducible representations of $\mathcal{G}_B$ are given by the irreducible representations of $\widetilde{SL}(2,\mathbb{R}) \times \mathbb{R}$ which are invariant under the action of elements $(h_n, Bn)$, for $n \in \mathbb{Z}$, in the $\mathbb{Z}$ subgroup of $\widetilde{SL}(2,\mathbb{R}) \times \mathbb{R}$. The irreducible representations of $\widetilde{SL}(2,\mathbb{R})$ are labeled by two quantum numbers $\eta$ and $\mu$, and the irreps of $\mathbb{R}$ are labelled by $k \in \mathbb{R}$. The irreducible representations of $\mathcal{G}_B$ are given by the irreps of $\widetilde{SL}(2,\mathbb{R}) \times \mathbb{R}$ satisfying

$$\mu = -\frac{Bk}{2\pi} + p, \quad p \in \mathbb{Z}. \tag{262}$$

---

[21]See [109] for more detailed discussions of Wilson lines in theories with compact gauge groups.

[22]For a comprehensive exposition on its representations, see [110].

The boundary BF action is modified accordingly to

$$I = -i \int_{M_2} \text{Tr}(\phi F) - \oint_0^\beta du V_\lambda(\phi), \tag{263}$$

where

$$A = e^1 P_1 + e^2 P_2 + \omega P_0 + \frac{B^2}{\pi^2} A^{\mathbb{R}} \mathbb{I}, \quad \phi = \phi^1 P_1 + \phi^2 P_2 + \phi^0 P_0 + \phi^{\mathbb{R}} \mathbb{I}. \tag{264}$$

Motivated by the results of Section 4.2, we choose the deformed potential $V_\lambda(\phi)$ to be

$$V_\lambda(\phi) = \frac{1 - \sqrt{1 - 8\lambda \widetilde{V}(\phi^0, \phi^\pm, \phi^{\mathbb{R}})}}{4\lambda} = \frac{1 - \sqrt{1 - 8\nu\lambda \left(\frac{1}{2} + \frac{1}{4} \text{Tr}_{(2, -\frac{\pi}{B})} \phi^2\right)}}{4\lambda} + O\left(\frac{1}{B}\right), \tag{265}$$

where we used that, in the large-$B$ limit, the potential $\widetilde{V}(\phi)$ in (265) is

$$\widetilde{V}(\phi^0, \phi^\pm, \phi^{\mathbb{R}}) = \frac{\nu}{2} + \frac{\nu}{4} \text{Tr}_{(2, -\frac{\pi}{B})} \phi^2 + O\left(\frac{1}{B}\right). \tag{266}$$

By $\text{Tr}_{(2, -\frac{\pi}{B})}$, we mean the trace is taken over the 2-dimensional representation with $k = -\frac{\pi}{B}$. In the large-$B$ limit, the trace is only over $\mathfrak{sl}(2, \mathbb{R}) \subset \mathfrak{sl}(2, \mathbb{R}) \oplus \mathbb{R}$. For the boundary conditions, we add a boundary term

$$i \oint_{\partial \Sigma} \phi^{\mathbb{R}} A^{\mathbb{R}}, \tag{267}$$

and fix the boundary value of $\phi^{\mathbb{R}}$ to be $k_0$. The partition function $Z_{k_0}(\tilde{g}, \nu, \beta)$ that we find is related to the partition function $Z((\tilde{g}, \theta), \nu\beta)$ with a fixed holonomy $\tilde{g} \in \widetilde{SL}(2, \mathbb{R})$ and $\theta \in \mathbb{R}$ via the Fourier transform

$$Z_{k_0}(\tilde{g}, \nu, \beta) = \int_{-\infty}^{\infty} d\theta Z((\tilde{g}, \theta), \nu, \beta) \exp(-ik_0 \theta). \tag{268}$$

We are ready to compute the disk partition function $Z_{k_0}(\tilde{g}, \nu, \beta)$ with $k_0 = -i$ and $B \to \infty$ in the deformed theory.

The non-trivial irreducible unitary representations of $\widetilde{SL}(2, \mathbb{R})$ consist of three types, and among the three, only the principal unitary series $C^\mu_{\eta = \frac{1}{2} + is}$ with $\mu \in [-1/2, 1/2]$ and the positive/negative discrete series $D^\pm_\eta$ with $\eta = \pm\mu > 0$ admit a well-defined Hermitian inner product allowing one to define a density of states given by the Plancherel measure. Taking the $\mathbb{Z}$ quotient fixes $\exp(2\pi i\mu) = \exp(-iBk)$ (see (3.19) in [18]) and the disk partition function $Z(g, \nu\beta)$ receive contributions only from the above two types of representations:

$$Z(g, \nu, \beta) \propto \int_{-\infty}^{\infty} dk \int_0^{\infty} ds \frac{s \sinh(2\pi s)}{\cosh(2\pi s) + \cos(Bk)} \chi_{(s, \mu = -\frac{Bk}{2\pi}, k)}(g) \exp\left(-\beta f_\lambda^- \left(\frac{\nu s^2}{2}\right)\right)$$

$$+ \sum_{n=1}^{n_{\max}} \frac{1}{2\pi^2} \left(-\frac{Bk}{2\pi} + n - \frac{1}{2}\right) \chi(g)$$

$$\times \exp\left[-\beta f_\lambda^- \left(\frac{\nu}{2} \left(\left(-\frac{Bk}{2\pi} + n\right)\left(1 + \frac{Bk}{2\pi} - n\right) - \frac{1}{4}\right)\right)\right], \tag{269}$$

where the first term is the contribution from the principal series representations, the second term is the contribution from the discrete series representations, and $n_{\max}$ is a cut-off on the discrete series representations.

We consider the boundary condition $k_0 = -i$ and the limit $B \to \infty$ to compute $Z_{k_0}(\tilde{g}, \nu, \beta)$. We arrive at important subtleties. In the undeformed theory, the leading order contribution in this limit comes from the principal series and scales as

$$\frac{1}{\cosh(2\pi s) + \cos(Bk_0)} \sim \exp(-B).$$ (270)

The contribution from the discrete series scales as

$$\exp\left[-\frac{\nu\beta}{2}\left(\left(-\frac{Bk_0}{2\pi}\right)\left(1 + \frac{Bk_0}{2\pi} - n\right) - \frac{1}{4}\right)\right] \sim \exp\left(-\frac{\nu\beta}{8\pi^2}B^2\right),$$ (271)

which is subleading and can be dropped. In the deformed theory, the scaling of the contribution from the principal series remains the same while the scaling of the contribution from the discrete series can change depending on the sign of the deformation. For the good sign of $\lambda$, we have

$$\exp\left[-\beta f_\lambda^-\left(\frac{\nu}{2}\left(\left(-\frac{Bk}{2\pi} + n\right)\left(1 + \frac{Bk}{2\pi} - n\right) - \frac{1}{4}\right)\right)\right] \sim \exp\left(-\frac{\beta B}{4\pi}\sqrt{\frac{\nu}{-\lambda}}\right).$$ (272)

Comparing with the suppression $\exp(-B)$ from the principal series, we find the principal series remains dominant as long as

$$\frac{\beta}{4\pi}\sqrt{\frac{\nu}{-\lambda}} > 1.$$ (273)

Consequently, we identify the critical temperature $T_c = \frac{1}{4\pi}\sqrt{\frac{\nu}{-\lambda}}$ as the temperature for the Hagedorn transition.

For the bad sign of $\lambda$, the function

$$f_\lambda^-\left(\frac{\nu}{2}\left(\left(-\frac{Bk_0}{2\pi}\right)\left(1 + \frac{Bk_0}{2\pi} - n\right) - \frac{1}{4}\right)\right)$$ (274)

becomes complex when $B \to \infty$ so we will not find the desired suppression. We suspect that the resolution to this issue for the bad sign is to the follow an analysis similar to that of [84], where including the non-perturbative contribution $f_\lambda^+(E)$ makes the partition function real.

For a non-compact group, which is relevant for JT gravity, the corresponding expression for the partition function is

$$Z_\lambda(g, \nu, \beta) = \int dR\,\rho(R)\,\chi_R(g)\exp\left(-\beta f_\lambda^-\left(\frac{\nu C_2(R)}{N}\right)\right).$$ (275)

Here $g$ is the holonomy, $R$ is the representation, $\chi_R$ is the character, and $\rho$ is the density of states. We note that only the energy flows via $f_\lambda^-(E)$ but the other factors in the integrand are $\lambda$-independent. This result is reminiscent of the expression for the deformed partition function in terms of an integral transformation involving the undeformed partition function and kernel discussed in [82, 83] (see also [111] for analogous integral kernel expressions in $2d$ theories). We write the principal series portion of the deformed Wilson line in terms of the un-normalized Wilson line anchored at $\tau_1$ and $\tau_2$ on the boundary as (schematically $E = \frac{\nu s^2}{2}$)

$$\langle W(\tau_1, \tau_2)\rangle_\lambda(g) = \int dh\, Z_\lambda(h, \nu, \tau_{21})\chi(h) Z_\lambda\left(gh^{-1}, \nu, \tau_{12}\right) =$$

$$\int_0^\infty ds_1^2 ds_2^2 \sinh(2\pi s_1)\sinh(2\pi s_2) N_{s_1, \eta^\pm}^{s_2} \exp\left(-\left[\tau_{21} f_\lambda^-\left(\frac{\nu s_1^2}{2}\right) + \tau_{12} f_\lambda^-\left(\frac{\nu s_2^2}{2}\right)\right]\right) =$$

$$\prod_{i=1}^2 \mathcal{D}_{y_i; \lambda}\big|_{y_i = \tau_{i+1,i}} \int ds_1^2 ds_2^2 \sinh(2\pi s_1)\sinh(2\pi s_2) N_{s_1, \Lambda^\pm}^{s_2} \exp\left(-\frac{\nu}{2}\left[y_1 s_1^2 + y_2 s_2^2\right]\right),$$ (276)

where[23] the differential operator $\mathcal{D}_{y;\lambda}$, also defined in [58, 83], is given by the infinite series of $y$-derivatives as follows[24]:

$$
\begin{aligned}
\exp\left(-\tau_{i+1,i} f_\lambda^-\left(\frac{\nu s_i^2}{2}\right)\right) &= \exp\left(-\tau_{i+1,i}\sum_{m=1}^\infty c_m \lambda^m (\nu s_i^2)^{m+1}\right)\exp\left(-\frac{\tau_{i+1,i}\,\nu s_i^2}{2}\right) \\
&= \exp\left(-\tau_{i+1,i}\sum_{m=1}^\infty c_m \lambda^m (-2\partial_y)^{m+1}\right)\bigg|_{y=\tau_{i+1,i}}\exp\left(-\frac{\nu s_i^2\, y}{2}\right) \quad (277) \\
&= \mathcal{D}_{y;\lambda}|_{y=\tau_{i+1,i}}\exp\left(-\frac{\nu y s_i^2}{2}\right).
\end{aligned}
$$

Here $\tau_{21} \equiv \tau_2 - \tau_1$, $\tau_{12} \equiv \beta - \tau_{21}$ and $N^{s_2}_{s_1,\eta^\pm}$ are fusion coefficients between two continuous series representations and a discrete series representation provided in Appendix D of [18]:

$$
N^{s_2}_{s_1,\eta^\pm} = \frac{\Gamma(\eta \pm is_1 \pm is_2)}{\Gamma(2\eta)}. \tag{278}
$$

## 6.2 Non-intersecting BF Wilson Lines and Local Operators

Additionally, one may also consider other examples. Again with the boundary holonomy $g \to \mathbb{I}$ and $k_0 = -i$ for $n$ non-intersecting Wilson lines, we write the unrenormalized expression

$$
\begin{aligned}
\left\langle \prod_{i=1}^n W(\tau_{2i-1}, \tau_{2i})\right\rangle &= \int\left(\prod_{i=1}^n dh_i\right)\left(\prod_{i=1}^n Z_\lambda(h_i, \nu, \tau_{2i,2i-1})\bar\chi(h_i)\right)Z_\lambda\left(g(h_1\ldots h_n)^{-1}, \nu\tau_{1,2n}\right) \\
&= \int ds_0\rho(s_0)\left(\prod_{i=1}^n ds_i\rho(s_i)N^{s_0}_{s_i,\eta^\pm}\right) \\
&\quad \times \exp\left(-\left[\left(\sum_{i=1}^n f_\lambda^-\left(\frac{\nu s_i^2}{2}\right)\tau_{2i,2i-1}\right) + f_\lambda^-\left(\frac{\nu s_0^2}{2}\right)\left(\beta - \sum_{i=1}^n \tau_{2i,2i-1}\right)\right]\right),
\end{aligned}
$$
$$(279)$$

with $\tau_{2i,2i-1} = \tau_{2i} - \tau_{2i-1}$, $i = 1,\ldots,n$ as defined below (277), and $\tau_{2n,1} \equiv \beta - \tau_{21} - \ldots - \tau_{2n,2n-1}$ is the total boundary length not enclosed by $n$ Wilson lines.[25] Equivalent to the single Wilson line case (276), one may also express (279) in terms of a product of the derivative operator defined in (277).

Moreover, it is interesting to consider correlators involving the topological term[26] $\mathrm{Tr}\,\phi^2(x)$, because they are the zero-length limit of various loop or line operators. This correlator is equivalent to insertions of the Hamiltonian operator $H(x)$ at different points in the path integral:

$$
\begin{aligned}
\left\langle \mathrm{Tr}\,\phi^2(x_1)\cdots\mathrm{Tr}\,\phi^2(x_n)\right\rangle_{k_0} &= \left(\frac{\nu}{4}\right)^{-n}\langle H(x_1)\cdots H(x_n)\rangle_{k_0} \\
&= \Xi\int_0^\infty ds\rho(s)s^{2n}\exp\left(-\frac{\nu\beta s^2}{2}\right) \tag{280} \\
&= (-2)^n \partial_{\nu\beta}^n Z_{k_0}(\nu\beta),
\end{aligned}
$$

---

[23]$\tau_{i+1,i} \equiv \tau_{i+1} - \tau_i$. Here, for 2-point function, $\tau_{32} = \beta - \tau_1 + \tau_2$. In general, for $n$-point function, we would have $\tau_{n+1,n} = \beta - \sum_{i=1}^{n-1}\tau_{i+1,i}$.

[24]We slightly abuse the notation here. Strictly speaking, we should add an another subscript $\tau_{i+1,i}$ such that $\mathcal{D}_{y;\tau_{i+1,i};\lambda} = \exp\left(-\tau_{i+1,i}\sum_{m=1}^\infty c_m\lambda^m(-\partial_y)^{m+1}\right)$, but since we will then later fix $y = \tau_{i+1,i}$, we will drop the additional subscript $\tau_{i+1,i}$.

[25]Note that the notation $\tau_{2n,1}$ is unambiguous – it cannot be interpreted as $\beta - \tau_{1,2n}$ because $\tau_{1,2n}$ is never $\tau_{2i,2i-1}$.

[26]The reason why it is topological can be seen from the Schwinger-Dyson equation [18].

where the partition function is

$$Z_{k_0}(\nu\beta) = \Xi \int_0^\infty ds \rho(s) \exp\left(-\frac{\nu\beta s^2}{2}\right). \tag{281}$$

The divergent factor $\Xi = \lim\limits_{x \to 1^+, n=0} \chi_{s,\mu}(g)$ is a limit of the character $\chi_{s,\mu}(\widetilde{g})$, related to $\widetilde{\mathrm{SL}}(2,\mathbb{R})$ principal series representations, which arises from summing over all states in each continuous series irrep $\eta = \frac{1}{2} + is$.[27] The independence of $x_1, \ldots, x_n$ in the last line of (281) simply reflects the topological nature of the BF theory.

In the $B \to \infty$ limit and with $k_0 = -i$, the integral (280) is easily evaluated as

$$\left\langle \mathrm{Tr}\,\phi^2(x_1) \cdots \mathrm{Tr}\,\phi^2(x_n) \right\rangle_{k_0} \propto \frac{2^{n+\frac{3}{2}}\pi\Xi\Gamma\left(n+\frac{3}{2}\right)}{(\nu\beta)^{n+\frac{3}{2}}} \, {}_1F_1\left(n+\frac{3}{2}; \frac{3}{2}; \frac{2\pi^2}{\nu\beta}\right), \tag{282}$$

where ${}_1F_1(a; b; z)$ is the Kummer confluent hypergeometric function defined for $n > -1$ and $\nu\beta > 0$. The disk's density of states and partition function in JT gravity are the usual

$$\rho(s) = s\sinh(2\pi s), \quad Z_{k_0}(\nu\beta) = \Xi\left(\frac{2\pi}{\nu\beta}\right)^{\frac{3}{2}} \exp\left(\frac{2\pi^2}{\nu\beta}\right). \tag{283}$$

In the deformed setting, the integral of concern is

$$\Xi \int_0^\infty ds \rho(s) s^{2n} \exp\left(-\beta f_\lambda^-\left(\frac{\nu s^2}{2}\right)\right). \tag{284}$$

In fact, a similar integral was also evaluated in [82] but now the deformed correlator $\left\langle \mathrm{Tr}\,\phi^2(x_1) \cdots \mathrm{Tr}\,\phi^2(x_n) \right\rangle_{k_0,\lambda}$ involves $\nu\beta'$-derivatives of their deformed partition function. We first express the deformed Boltzmann weight using a kernel $K(\beta, \beta')$ as

$$\exp\left(-\beta f_\lambda^-\left(\frac{\nu s^2}{2}\right)\right) = \int_0^\infty d\beta' K(\beta, \beta') \exp\left(-\frac{\nu\beta' s^2}{2}\right), \tag{285}$$

so that we can re-express the deformed partition function as [82, 83]

$$Z_{k_0}(\beta)_\lambda = \int_0^\infty d\beta' K(\beta, \beta') Z_{k_0}(\beta'). \tag{286}$$

The kernel is the inverse Laplace transform of the Boltzmann weight of the deformed theory:

$$\begin{aligned}
K(\beta, \beta') &= \frac{1}{2\pi i} \int_{-i\infty}^{i\infty} dE \exp\left(-\beta f_\lambda^-(E) + \beta' E\right) \\
&= \frac{\beta}{\sqrt{-8\pi\lambda}(\beta')^{\frac{3}{2}}} \exp\left(\frac{(\beta-\beta')^2}{8\beta'\lambda}\right).
\end{aligned} \tag{287}$$

Then, for our integral (284), we have

$$\begin{aligned}
\Xi \int_0^\infty ds \rho(s) s^{2n} \exp\left(-\beta f_\lambda^-\left(\frac{\nu s^2}{2}\right)\right) &= \Xi \int_0^\infty d\beta' K(\beta, \beta') \int_0^\infty ds \rho(s) s^{2n} \exp\left(-\frac{\nu\beta'}{2}s^2\right) \\
&= (-2)^n \int_0^\infty d\beta' K(\beta, \beta') \partial_{\nu\beta'}^n Z_{k_0}(\nu\beta').
\end{aligned} \tag{288}$$

---

[27]See [18] for more comments on the divergent factor $\Xi$.

In other words, one can perform an integral transformation for the undeformed correlators $\left\langle \mathrm{Tr}\,\phi^2(x_1)\cdots\mathrm{Tr}\,\phi^2(x_n)\right\rangle_{k_0}$ against a kernel (287) to obtain the deformed correlators for any $n$ in principle. Equivalent to the above method (288), we also derive a recursion relation. We denote

$$\langle X\rangle \equiv \Xi \int_0^\infty ds\rho(s)X\exp\left(-\beta f_\lambda^-\left(\frac{vs^2}{2}\right)\right) \tag{289}$$

and define $F_n = \langle s^{2n}\rangle$. Then by the linearity of $\langle X\rangle$, we arrive at

$$\partial_\beta F_n = -\left\langle s^{2n}\frac{1-\sqrt{1-4v\lambda s^2}}{4\lambda}\right\rangle = -\frac{F_n}{4\lambda}+\left\langle\frac{\sqrt{1-4v\lambda s^2}}{4\lambda}\right\rangle, \tag{290}$$

$$\partial_\beta^2 F_n = \left\langle s^{2n}\left(\frac{1-\sqrt{1-4v\lambda s^2}}{4\lambda}\right)^2\right\rangle = \frac{1}{8\lambda^2}F_n - \frac{v}{4\lambda}F_{n+1} - \frac{1}{2\lambda}\left\langle\frac{\sqrt{1-4v\lambda s^2}}{4\lambda}\right\rangle. \tag{291}$$

We find

$$F_{n+1} = \frac{-4\lambda\partial_\beta^2 F_n - 2\partial_\beta F_n}{v}. \tag{292}$$

From this recursion relation (292), one obtains

$$F_n = v^{-n}(-4\lambda\partial_\beta^2 - 2\partial_\beta)^n F_0, \tag{293}$$

where the deformed disk partition function from [82, 83] is

$$F_0 = \Xi\frac{2\pi\beta}{\sqrt{-v\lambda}}\frac{\exp\left(-\frac{\beta}{4\lambda}\right)}{v\beta^2 + 16\pi^2\lambda}K_2\left(-\frac{\sqrt{\beta^2 + 16v^{-1}\pi^2\lambda}}{4\lambda}\right). \tag{294}$$

Here $K_2(x)$ is the modified Bessel function of the second kind and is defined up to the inverse Hagedorn temperature

$$\beta_H = 4\pi\sqrt{\frac{-\lambda}{v}}, \tag{295}$$

agreeing with (273). For general $n$, the explicit formulas for the deformed correlators are not illuminating to write down but are easily computed with Mathematica.

Our analysis leads to a new understanding of the Hagedorn transition in the Schwarzian quantum mechanics. In the $B \to \infty$ limit, when turning on the $T\overline{T}$ deformation, the contribution from the principal series competes with the discrete series. Below the critical temperature $T_c = \frac{1}{4\pi}\sqrt{\frac{v}{-\lambda}}$, the principal series remains dominant over the discrete series, just as in the undeformed theory. Therefore, the effective boundary theory is described by $T\overline{T}$-deformed Schwarzian quantum mechanics. This critical temperature $T_c$ coincides with the critical temperature $T_H = 1/\beta_H$ of the Hagedorn transition of the Schwarzian quantum mechanics. In other words, the BF description of JT gravity provides a UV completion which allows us to understand what happens when crossing the transition temperature $T_H = T_c$: the discrete series becomes dominant over the principal series, and therefore the boundary theory is no longer described by the $T\overline{T}$ deformation of the Schwarzian theory, but rather some other $T\overline{T}$-deformed theory associated with the discrete series.

# 7 Summary and Discussion

In this paper, we have interpreted the dimensionally reduced $T\overline{T}$ deformation in a $1d$ theory from the perspective of its $2d$ holographic dual, which can be presented as either a JT gravity theory or a BF gauge theory.

In BF variables, we saw that the effect of this deformation depends on the boundary term (and thus the variational principle) which defines the undeformed seed theory. For one choice of boundary term, we find that a $T\overline{T}$-like deformation in the $1d$ dual causes a rotation of the sources and expectation values of the $2d$ BF theory. This matches the expectation from the analogous deformation of the $3d$ gravitational Chern-Simons theory which is dual to the ordinary $2d$ $T\overline{T}$ deformation of a CFT. For the choice of boundary term which yields the Schwarzian theory as the holographic dual, we find that the $T\overline{T}$-like deformation of the boundary can be presented in the so-called $HT$ form, where the flow is driven by a product of the Hamiltonian (or Euclidean Lagrangian) and the corresponding Hilbert stress tensor. In the bulk, such a deformation can be interpreted either as an asymptotic field redefinition of the gauge field $A_\tau$, or as a modification of the boundary conditions relating $A_\tau$ to the BF scalar field $\phi$.

As we have stressed, Wilson lines and loops are natural observables in gauge theories, including the $3d$ Chern-Simons theory which is classically equivalent to AdS$_3$ gravity and the analogous $2d$ BF gauge theory which repackages the fields of JT gravity. Motivated by this, we compute corrections to the Wilson line and related correlators induced by a $T\overline{T}$ deformation on the boundary. In the case of $3d$ Wilson lines in the gravitational Chern-Simons theory, the classical deformed Wilson line can be evaluated exactly for constant stress tensor backgrounds, and corrections to the quantum Wilson line can be calculated perturbatively when both $\lambda c$ is small and $c$ is large. In the context of $2d$ BF theory, the deformed Wilson lines can be expressed in terms of deformed disk partition functions, and an analysis of the contributions from the principal and discrete series allows us to identify a critical temperature which is interpreted as the point of the Hagedorn transition.

We now describe a few interesting directions for future research.

*Higher spin gravity*

One substantial advantage of the Chern-Simons formulation of $3d$ gravity is that it can be straightforwardly generalized to a theory of gravity coupled to finitely many massless higher-spin fields. To do this, one simply replaces the $SL(2,\mathbb{R})$ gauge group with $SL(N,\mathbb{R})$ while keeping the same form of the Chern-Simons action.[28] This prescription for defining a $3d$ higher spin gravity theory is very convenient compared to the analogous problem in $d > 3$ dimensions, where one must generically include an infinite tower of higher spin fields and where writing down an action is more difficult.

It is natural to wonder about the bulk interpretation of $T\overline{T}$-deforming a boundary CFT which is dual to such a higher-spin Chern-Simons theory. In the case of $SL(2,\mathbb{R})$, as studied in [77] and reviewed in Section 2.2, one begins by writing down a more general class of boundary conditions for the Chern-Simons gauge field $A_\mu$ than the usual Bañados-type boundary conditions. These generalized boundary conditions involve sources $e_i^a$ and expectation values $f_i^a$; the effect of a boundary $T\overline{T}$ deformation is then to modify the sources to $e_i^a(\lambda)$ which depend on the dual expectation values.

The most naïve way to proceed in the higher spin case would be to imitate this procedure and turn on *all* possible sources and expectation values in the $SL(N,\mathbb{R})$ expansion of the higher-spin gauge field $A_\mu$. One way of parameterizing these higher-spin contributions, discussed for instance in [119], is to let

$$a(x^+) = \sum_{i=-1}^{1} f^i(x^+) L_i + \sum_{i=2}^{r} \sum_{m=-\ell_i}^{\ell_i} w^{\ell_i,m} W_{\ell_i,m}\,, \tag{296}$$

where the $L_i$ generate the usual $SL(2,\mathbb{R})$ subgroup of $SL(N,\mathbb{R})$, and the $W_{\ell_i,m}$ are additional generators associated with spin $(\ell_i + 1)$ degrees of freedom.

---

[28]See [13, 19, 112–122] and references therein for discussions of higher spin $3d$ Chern-Simons theory.

A general expansion (296) with all coefficient functions $f^i, w^{\ell_i, m}$ turned on will not yield a bulk solution which is asymptotically AdS. In fact, for some choices of coefficient functions the metric can even diverge as one approaches the boundary. It is therefore important to impose restrictions on an expansion like (296) which guarantee that the field configuration for $A_\mu$ yields a sensible gravity solution.

One common way of imposing such restrictions is the Drinfeld-Sokolov reduction, which from the bulk perspective implies that we have $AdS_3$ boundary conditions at infinity [119,120]. This reduces the current algebra in the dual field theory from $\mathfrak{sl}(N,\mathbb{R}) \times \mathfrak{sl}(N,\mathbb{R})$, which we would expect for a general gauge field, to a $W$-algebra.

However, this Drinfeld-Sokolov procedure restricts us to a class of boundary conditions which is too constraining to allow a $T\overline{T}$-type deformation. A simple way to see this is to consider the simple $SL(2,\mathbb{R})$ case, where the Drinfeld-Sokolov reduction imposes Bañados-type boundary conditions on the metric. We have already seen from [77] that these boundary conditions are too restrictive and that one must consider generalized boundary conditions in order to accommodate a $T\overline{T}$ deformation.

It would be very interesting to find the appropriate modified boundary conditions for $SL(N,\mathbb{R})$ Chern-Simons theory which allow a $T\overline{T}$-type deformation. One might expect that, within this generalized class, the deformation would again correspond to a linear mixing of certain sources and expectation values, but even this is not clear. As a first step towards solving this problem, one might consider the analogous question in $2d$ BF theory with gauge group $G = SL(N,\mathbb{R})$. The dual $1d$ theory, as we have seen, is simply that of a free particle moving on the $SL(N,\mathbb{R})$ group manifold, and it is trivial to deform this theory using the $(0 + 1)$-dimensional version of $T\overline{T}$. One might hope that studying this deformation might suggest an appropriate modification of the boundary conditions for the $SL(N,\mathbb{R})$ gauge field in the BF theory.[29] Given such a modification, we could then ask whether a dimensional lift of this deformation provides an answer to the original question of how to generalize the Chern-Simons boundary conditions in three dimensions.

*Higher order corrections in $\lambda$ and $c$ to the quantum $AdS_3$ Wilson line*

As alluded to in Section 5.3, the deformed $AdS_3$ quantum Wilson line is computationally difficult due to the double expansion in $\lambda$ and $\frac{1}{c}$ and the regularization of the path-ordered exponential integrals. While we only have considered the leading correction to the quantum $AdS_3$ Wilson line, at $O(\lambda^2 c^0)$, it is desirable to systematically study higher order contributions at different orders in $\lambda$ and $\frac{1}{c}$ as automated for $\lambda = 0$ in Section 5 of [9].

For instance, when we expand the path-ordered exponential (204) to $O(\frac{1}{c^2})$ in dimension $2-\varepsilon$, one uses the deformed two-loop two-point planar stress tensor correlator recently computed by [60]

$$\langle T_{zz}(z_1) T_{zz}(z_2) \rangle = \frac{1}{z_{12}^4} \left[ \frac{\frac{3}{2G}+1}{2} + \frac{10(3+4G)r_c^2}{|z_{12}|^4} + \frac{96Gr_c^3 \left(8 + 60\ln\left(\mu^2|z_{12}|^2\right)\right)}{|z_{12}|^6} + \frac{2520Gr_c^4}{|z_{12}|^8} \right] \tag{297}$$

to calculate the Wilson line's loop contributions.[30]

In general, expanding the path-ordered exponential (204) in powers of $\frac{\alpha(\varepsilon)}{c}$,

$$\langle W_\varepsilon[0,z] \rangle_\lambda =$$
$$z^{2j} N(\varepsilon) \sum_{n=0}^{\infty} \frac{(6\alpha(\varepsilon))^n}{c^n} \int_0^z dy_n \cdots \int_0^{y_2} dy_1 F_n(z; y_n, \ldots, y_1) \langle T_{zz}(y_n) \cdots T_{zz}(y_1) \rangle_\lambda, \tag{298}$$

---

[29]See also [123] for recent related work on higher spin BF theory and its generalization of [28].

[30]Here $\frac{3}{2G} + 1$ is the one-loop corrected Brown-Henneaux central charge of the $r_c = 0$ theory following the renormalization conventions in [60] and $\mu$ is an unspecified renormalization parameter.

one can systematically calculate the quantum gravitational Wilson line to any order in $\lambda$ or $\frac{1}{c}$ by using (loop-corrected) deformed $n$-point planar stress tensor correlators. The tree-level higher point planar stress tensor correlators were found perturbatively in $\lambda$ by [50, 56].

*Charting the phase diagram of deformed Schwarzian theory*

As we found in Section 6, below the Hagedorn transition temperature the principal series dominates over the discrete series in the deformed BF theory which captures the $T\overline{T}$-deformed Schwarzian theory. However, above the transition temperature the discrete series dominates the principal series, which is consistent with the breakdown of the deformed Schwarzian theory description at and across the Hagedorn transition. One would naturally expect that, above the Hagedorn transition temperature, the correct effective theory should correspond to the spectrum of the discrete series from the BF theory. At the critical temperature, the boundary theory should capture the contributions from both the principal and discrete series, since the contributions from the two are comparable at the Hagedorn temperature. Furthermore, correlation functions in these theories should have the bulk interpretation as correlation functions of Wilson lines. One could then ponder how to find the correct quantum mechanics that describes these boundary theories at and above the Hagedorn temperature.

*Connecting the 2d Wilson lines with the 3d Wilson lines in the deformed theory*

Given the intimate and yet subtle relationship between $2d$ JT gravity and $3d$ gravity [21, 23, 124, 125], we expect it is possible to compute the correlation functions involving the Wilson line in $2d$ BF theory from the correlators of Wilson lines in the Chern-Simons description of $3d$ gravity under the $T\overline{T}$ deformation. This has been explored in the undeformed theory by [22, 126, 127]. To study this relation in the deformed theory, one possible direction is to use the result that the Wilson line in $3d$ gravity corresponds to a bi-local operator in the boundary CFT. Then one can turn on the $T\overline{T}$ deformation in the boundary CFT to study correlation functions of these bi-local operators on the torus in the same limit studied in [127], which leads to a Schwarzian sector for any CFT with large central charge $c$. The $T\overline{T}$-deformed CFT correlation functions on the torus were computed via conformal perturbation theory in [55]. Determining the deformed correlation functions of $2d$ Wilson lines from the correlation functions of bi-local operators [55, 127] is desirable.

*The (graded) Poisson sigma model and generalized dilaton (super)gravity*

Our work focused on a special case of the most general $2d$ dilaton gravity theory, namely JT gravity and its BF theory description. However, one could consider other kind of models such as those listed in the bestiary in Appendix A of [128]. One could then study $T\overline{T}$-deformations of these more general models, as [129] did for a broad class of Maxwell-dilaton-gravity theories and showed that these theories exhibit the typical square-root behavior for the deformed energy spectrum. Limiting ourselves to an action functional containing at most two derivatives, the most general bulk (Euclidean) action supplemented with the Gibbons-Hawking-York boundary term is [128, 130]

$$I[g_{\mu\nu}, \Phi] = -\frac{1}{16\pi G} \int d^2 x \sqrt{g} \left[ \Phi R - U(\Phi) g^{\mu\nu} \nabla_\mu \Phi \nabla_\nu \Phi - 2V(\Phi) \right] - \frac{1}{8\pi G} \int d\tau \sqrt{\gamma} \, \Phi K \,, \tag{299}$$

where different $2d$ dilaton gravity models are distinguished by kinetic and potential functions $U(\Phi)$ and $V(\Phi)$ respectively.

Analogously to the Chern-Simons description of $3d$ gravity, one has a gauge-theoretic formulation of (299) as the topological Poisson sigma model [131, 132] with $3d$ target space.

The gravitational Poisson sigma model action is[31]

$$I_{\text{PSM}}[A_i, X_i] = \frac{1}{8\pi G} \int \left( A_i \wedge dX^i + \frac{1}{2} P^{ij}(X) A_i \wedge A_j \right).$$ (300)

Here $X_i$ are the set of target space coordinates spanning a Poisson manifold with Poisson tensor $P^{ij}(X) = -P^{ji}(X)$ and $A_i$ are the one-form gauge fields which, in general, transform non-linearly under gauge transformations to preserve the action (300). Generalizing our $T\overline{T}$-deformed analysis of the BF theory and our previous studies on supersymmetric $\mathcal{N} = 1, 2$ quantum mechanics [72] under the framework of general dilaton supergravity theories described by a graded Poisson sigma model (300) is an interesting direction.

*Irrelevant Deformations as Re-Coupling Throat Regions*

The single-trace $T\overline{T}$ deformation has a well-known bulk gravity interpretation in the context of type IIB supergravity on $\text{AdS}_3 \times S^3 \times T^4$ [39,134]. More specifically, consider the IIB solution for a bound state of fundamental strings and NS5-branes. The F-strings wrap a circular direction $x_5$ of the $\text{AdS}_3$, whereas the NS5-branes wrap both $x_5$ and all cycles of the $T^4$.

This gravity solution is characterized by two length scales, $r_1$ and $r_5$, associated with the horizons of the F-strings and NS5-branes respectively. If we restrict to the deep bulk, where the radial $\text{AdS}_3$ coordinate $r$ is small compared to both $r_1$ and $r_5$, then we are in the near-horizon region of both the strings and the five-branes. This region looks like a conventional $\text{AdS}_3$ spacetime which is dual to an ordinary CFT. The other supergravity fields are essentially spectators, since the dilaton is constant in the deep interior while the $H_3$ flux has two terms which thread the $S^3$ and $\text{AdS}_3$ but is otherwise non-dynamical. Thus this limit is effectively a solution of a three-dimensional pure gravity theory.

On the other hand, suppose that we assume $r \ll r_5$ but not necessarily $r \ll r_1$. In this limit we are in the near-horizon region of the five-branes but not of the strings. The resulting gravity solution interpolates between a pure $\text{AdS}_3$ solution at small $r$ to a linear dilaton spacetime at large $r$, with an additional parameter $\lambda$ in the solution which characterizes the slope of the linear dilaton. The holographic interpretation of such an interpolating solution is that we have deformed the dual CFT by the single-trace $T\overline{T}$ operator and flowed by the deformation parameter $\lambda$. In other words, the single-trace $T\overline{T}$ deformation has re-coupled the linear dilaton throat region of the bulk spacetime.

It would be interesting to explore whether some version of the $T\overline{T}$ deformation has a similar interpretation as re-coupling an intermediate region in other gravitational settings. One such setting is the near-horizon region of a near-extremal black hole in four dimensions. It was pointed out in [25] that this region is described by JT gravity on $\text{AdS}_2$, which is of course dual to a one-dimensional Schwarzian or particle-on-a-group theory. Is there an irrelevant deformation of this one-dimensional theory which has the interpretation of re-coupling more of the throat, between the near-horizon and asymptotically flat regions, of the 4$d$ black hole? That is, does some irrelevant operator in the 1$d$ theory capture the leading corrections as we move away from the limit $\frac{r}{r_H} = 1$ in the gravity solution, where $r_H$ is the horizon radius? If so, this would suggest that irrelevant current-type deformations have a more general holographic interpretation as capturing corrections to near-horizon limits.

---

[31]For recent works on general dilaton (super)gravity and its relation to the (graded) Poisson sigma model, see [94,133].

# Acknowledgements

We would like to thank Eric D'Hoker, Per Kraus, Ruben Monten, Mukund Rangamani and Savdeep Sethi for helpful discussions. S.E. is supported from the Bhaumik Institute. C. F. is supported by U.S. Department of Energy grant DE-SC0009999 and by funds from the University of California. H.-Y.S. is supported from the Simons Collaborations on Ultra-Quantum Matter, which is a grant from the Simons Foundation (651440, AK). Z.S. is supported from the US Department of Energy (DOE) under cooperative research agreement DE-SC0009919 and Simons Foundation award No. 568420.

# A Conventions

Here we collect this paper's conventions in $3d$ and JT gravity.

## A.1 3d Gravity

In the main text, we denote the generators of the Lie algebra $\mathfrak{sl}(2,\mathbb{R})$ by $L_{0,\pm1}$ obeying the standard commutation relation

$$[L_m, L_n] = (m-n)L_{m+n}. \tag{A.1}$$

The fundamental representation of $\mathfrak{sl}(2,\mathbb{R})$ and the generators have an explicit matrix representation

$$L_0 = \begin{pmatrix} \frac{1}{2} & 0 \\ 0 & -\frac{1}{2} \end{pmatrix}, \quad L_1 = \begin{pmatrix} 0 & 0 \\ -1 & 0 \end{pmatrix}, \quad L_{-1} = \begin{pmatrix} 0 & 1 \\ 0 & 0 \end{pmatrix}. \tag{A.2}$$

In the body of the paper, we use the notation $L_+$ for $L_1$ and $L_-$ for $L_{-1}$ for convenience, since the $\pm$ notation more closely resembles the indices appearing in light-cone coordinates.

In terms of the Pauli matrices

$$\sigma_1 = \begin{pmatrix} 0 & 1 \\ 1 & 0 \end{pmatrix}, \quad \sigma_2 = \begin{pmatrix} 0 & -i \\ i & 0 \end{pmatrix}, \quad \sigma_3 = \begin{pmatrix} 1 & 0 \\ 0 & -1 \end{pmatrix}, \tag{A.3}$$

the $\mathfrak{sl}(2,\mathbb{R})$ generators can be expressed as

$$L_0 = \frac{1}{2}\sigma_3, \quad L_1 = -\frac{1}{2}(\sigma_1 - i\sigma_2), \quad L_{-1} = \frac{1}{2}(\sigma_1 + i\sigma_2), \tag{A.4}$$

which obey

$$\mathrm{Tr}(L_1 L_{-1}) = -1, \quad \mathrm{Tr}(L_0^2) = \frac{1}{2}, \quad \mathrm{Tr}(L_{\pm1}^2) = \mathrm{Tr}(L_{\pm1}L_0) = 0. \tag{A.5}$$

Our expression for the $T\bar{T}$ operator in equation (27) involves two variants of the Levi-Civita symbol which are distinguished by using early Latin versus middle Latin letters for the indices. The version with flattened indices is written as $\varepsilon^{ab}$ and has components

$$\varepsilon_{+-} = -\varepsilon_{-+} = 1. \tag{A.6}$$

The Levi-Civita symbol with curved indices, $\varepsilon^{ij}$, has components

$$\varepsilon^{x^+ x^-} = -\varepsilon^{x^- x^+} = \frac{1}{\sqrt{-g}} = \frac{1}{2\det e}. \tag{A.7}$$

Using both of these versions of the Levi-Civita symbol, we can express the determinant of the stress tensor $T_i^a$ as

$$\det T_i^a = (\det e)\varepsilon_{ab}\varepsilon^{ij} T_i^a T_j^b. \tag{A.8}$$

## A.2 JT Gravity

We adopt the generators $P_0, P_1, P_2$ used in [18] and Section 3. The generators are defined as

$$P_0 = \begin{pmatrix} 0 & \frac{1}{2} \\ -\frac{1}{2} & 0 \end{pmatrix}, \quad P_1 = \begin{pmatrix} 0 & \frac{1}{2} \\ \frac{1}{2} & 0 \end{pmatrix}, \quad P_2 = \begin{pmatrix} \frac{1}{2} & 0 \\ 0 & -\frac{1}{2} \end{pmatrix}, \tag{A.9}$$

which can be written in the basis defined in (A.2) as

$$P_0 = \frac{1}{2}(L_{-1} + L_1), \quad P_1 = \frac{1}{2}(L_{-1} - L_1), \quad P_2 = L_0. \tag{A.10}$$

These generators obey the commutation relations

$$[P_0, P_1] = P_2, \quad [P_0, P_2] = -P_1, \quad [P_1, P_2] = -P_0, \tag{A.11}$$

and trace conditions

$$\mathrm{Tr}\left(P_0^2\right) = -\mathrm{Tr}\left(P_1^2\right) = -\mathrm{Tr}\left(P_2^2\right) = -\frac{1}{2}, \tag{A.12}$$
$$\mathrm{Tr}\left(P_0 P_1\right) = \mathrm{Tr}\left(P_1 P_2\right) = \mathrm{Tr}\left(P_0 P_2\right) = 0.$$

These generators are convenient because the fields (51) admit real expansions of the form

$$A(x) = \frac{1}{2}\begin{pmatrix} e^2 & \omega + e^1 \\ e^1 - \omega & -e^2 \end{pmatrix}, \quad \phi(x) = \frac{1}{2}\begin{pmatrix} \phi^2 & \phi^0 + \phi^1 \\ \phi^1 - \phi^0 & -\phi^2 \end{pmatrix}. \tag{A.13}$$

For the string defect, we used another set of generators

$$\ell_0 = \begin{pmatrix} 0 & \frac{i}{2} \\ -\frac{i}{2} & 0 \end{pmatrix}, \quad \ell_+ = \begin{pmatrix} -\frac{1}{2} & -\frac{i}{2} \\ -\frac{i}{2} & \frac{1}{2} \end{pmatrix}, \quad \ell_- = \begin{pmatrix} \frac{1}{2} & -\frac{i}{2} \\ -\frac{i}{2} & -\frac{1}{2} \end{pmatrix}, \tag{A.14}$$

which obey the commutation relations

$$[\ell_\pm, \ell_0] = \pm \ell_\pm, \quad [\ell_+, \ell_-] = 2\ell_0, \tag{A.15}$$

and trace conditions

$$\mathrm{Tr}\left(\ell_0^2\right) = \frac{1}{2}, \quad \mathrm{Tr}\left(\ell_\pm^2\right) = \mathrm{Tr}(\ell_\pm \ell_0) = 0, \quad \mathrm{Tr}(\ell_- \ell_+) = -1. \tag{A.16}$$

The fields (67) are

$$A_\tau = \frac{1}{2}\begin{pmatrix} e_- - e_+ & i(-e_- - e_+ + \omega) \\ i(-e_- - e_+ - \omega) & e_+ - e_- \end{pmatrix},$$
$$\phi = \frac{1}{2}\begin{pmatrix} \phi_- - \phi_+ & i(-\phi_- - \phi_+ + \phi_0) \\ i(-\phi_- - \phi_+ - \phi_0) & \phi_+ - \phi_- \end{pmatrix}. \tag{A.17}$$

# B  Family of $T\overline{T}$-Deformed Schwarzian Actions

In this appendix, we consider different boundary geometries in JT gravity and recover the deformed 1$d$ Schwarzian at $O(\lambda)$. We then make a conjecture for the deformed non-perturbative action using our expectation from flowing the 1$d$ Schwarzian theory, as was done in [82], where the deformed bulk JT gravity and Schwarzian actions were checked up to $O(\lambda)$. We use the bulk JT gravity techniques developed in [135] for assistance.

## B.1  Euclidean AdS Disk

For pedagogical purposes and as a warm-up exercise, we confirm that the deformed boundary action for the Euclidean AdS$_2$ disk given in [82] is correct by performing a bulk analysis.

The line element for a $2d$ black hole in JT gravity with unit horizon radius is

$$ds^2 = \left(r^2 - 1\right) d\tau^2 + \frac{dr^2}{r^2 - 1}. \tag{B.1}$$

We select a curve $(r(u), \tau(u))$ subject to the boundary conditions for the metric and dilaton,

$$ds^2\big|_{\partial M_2} = \frac{du^2}{\varepsilon^2}, \tag{B.2}$$
$$\Phi\big|_{\partial M_2} = \frac{\Phi_r}{\varepsilon},$$

where $\varepsilon$ is a small parameter which is related to $\lambda$ via

$$\lambda = \frac{2\pi G \varepsilon^2}{\Phi_r}. \tag{B.3}$$

Solving for the curve $(r(u), \tau(u))$ subject to the above boundary conditions (B.2) at $O(1/\varepsilon)$, we find

$$\lim_{r \to \infty} ds^2 = \frac{du^2}{u^2} \implies r^2 d\tau^2 = \frac{du^2}{\varepsilon^2}, \tag{B.4}$$

giving us

$$r = \frac{1}{\varepsilon \tau'}. \tag{B.5}$$

Going to $O(\varepsilon)$, we find

$$r = \frac{1}{\varepsilon \tau'} + \varepsilon \left( \frac{\tau'}{2} - \frac{\tau''^2}{2\tau'^3} \right). \tag{B.6}$$

To determine the boundary action (1), we compute the trace of the extrinsic curvature

$$K = \nabla_\mu n^\mu = \frac{\partial_u n^r}{r'}, \tag{B.7}$$

where the normal vector component in the radial direction is

$$n^r = \frac{\left(r^2 - 1\right)^{\frac{3}{2}} \tau'}{\sqrt{\tau'^2 \left(r^2 - 1\right)^2 + r'^2}}. \tag{B.8}$$

The trace of the extrinsic curvature at the radial cutoff is

$$K = \sqrt{r^2 - 1} \left[ \frac{r'\tau'' \left(r^2 - 1\right) + \tau' r'' + r\tau' \left(3r'^2 + \tau'^2(r^2-1)^2 - r^2 r''\right)}{(r'^2 + \tau'^2(1-r^2)^2)^{\frac{3}{2}}} \right]. \tag{B.9}$$

Substituting the curve $(r(u), \tau(u))$ from (B.6) into the trace of extrinsic curvature (B.9) and expanding in $\varepsilon$, we find up to $O(\varepsilon^4)$

$$
\begin{aligned}
K = & 1 + \varepsilon^2 \left( \frac{\tau'^4 - 3\tau''^2 + 2\tau'\tau'''}{2\tau'^2} \right) \\
& + \varepsilon^4 \left( -\frac{\tau'^7 + \tau'''\tau''^2 + 6\tau'''\tau'^4}{8\tau'^3} + \left( \frac{\tau''\left(-9\tau''^2 + 2\tau'^4 + 8\tau'''\tau'\right)}{8\tau'^3} \right)' \right).
\end{aligned}
\tag{B.10}
$$

Therefore, the boundary action is

$$I_{\text{bdry}} = -\frac{1}{8\pi G} \int_{\partial M_2} \frac{du}{\varepsilon} \frac{\Phi_r}{\varepsilon} (K-1)$$

$$= \int_{\partial M_2} du \left( L_0 + 2\lambda L_0^2 + O(\lambda^2) \right), \tag{B.11}$$

where $L_0$ is the finite temperature Schwarzian Lagrangian

$$L_0 = -\frac{\Phi_r}{8\pi G} \left\{ \tan\left(\frac{\tau}{2}\right), u \right\}. \tag{B.12}$$

Here (B.11) confirms the results in [82] and the expected deformed non-perturbative Lagrangian of the Schwarzian theory is

$$L_\lambda = \frac{1}{4\lambda} \left( 1 - \sqrt{\left(1 - \frac{\lambda \Phi_r}{2\pi G} \varphi'^2\right)\left(1 + \frac{\lambda \Phi_r}{2\pi G} \exp(2\varphi)\right)} \right), \tag{B.13}$$

where $\exp(\varphi) = \tau'$ and the seed theory is the Schwarzian Lagrangian (B.12). The next two cases in the following subsections have not been considered under the $T\bar{T}$ deformation.

## B.2 Euclidean AdS Double Trumpet

Here we consider a geometry with two asymptotic boundaries, namely the Euclidean $\text{AdS}_2$ double trumpet. The two asymptotic boundaries are located at $r \to \pm\infty$ in the $2d$ geometry whose line element is

$$ds^2 = \left(r^2 + 1\right) d\tau^2 + \frac{dr^2}{r^2 + 1}, \quad \tau \sim \tau + b, \tag{B.14}$$

where $b$ is a modulus which is usually integrated over in the path integral. We go through the same exercise as in the disk case to compute the trace of the extrinsic curvature. The relevant unit vector component is

$$n^r = \pm \frac{(1+r^2)\tau'}{\sqrt{\tau'^2(1+r^2)^2 + r'^2}}, \tag{B.15}$$

where the upper sign is the right boundary while the lower sign is the left boundary. From (B.15), the trace of the extrinsic curvature is

$$K = \frac{\sqrt{r^2+1}\left((r^2+1)r'\tau'' + \tau'\left(r\left(3r'^2 + (r^2+1)^2\tau'^2\right) - (r^2+1)r''\right)\right)}{\left(r'^2 + (r^2+1)^2\tau'^2\right)^{3/2}}. \tag{B.16}$$

Now, as before, we wish to solve for the curves $(r, \tau) = (r(u), \tau(u))$ on each of the two boundaries at their own respective finite radial cutoff.[32] At $O(\varepsilon_\pm)$, we have

$$r = \pm\left(\frac{1}{\varepsilon_\pm \tau'} - \varepsilon_\pm \left(\frac{\tau'}{2} + \frac{\tau''^2}{2\tau'^3}\right)\right). \tag{B.17}$$

Plugging in (B.17) into (B.16), we find the same trace of the extrinsic curvature for both signs as (B.10) but with different cutoffs $\varepsilon_\pm$. Therefore, the non-perturbative action is (B.13) at each boundary:

$$I_{\text{bdry}}(\lambda_+, \lambda_-) = I(\lambda_+) + I(\lambda_-), \tag{B.18}$$

---

[32]The two cutoff surfaces do not necessarily have to be at the same value of $\varepsilon$: we do not require $\varepsilon_- = \varepsilon_+$.

where

$$I(\lambda_{\pm}) = \frac{1}{4\lambda_{\pm}} \int_{\partial M_2^{\pm}} du \left[ 1 - \sqrt{\left(1 - \frac{\lambda_{\pm}\Phi_r}{2\pi G}\varphi'^2\right)\left(1 + \frac{\lambda_{\pm}\Phi_r}{2\pi G}\exp(2\varphi)\right)} \right]. \tag{B.19}$$

With this deformed double trumpet action (B.18), one can then compute the deformed partition function

$$Z_{\text{DT}}(\lambda_+, \lambda_-) = \int \frac{\mathcal{D}\Phi\,\mathcal{D}g_{\mu\nu}}{\text{Vol}(\text{Diff})}\exp(-I_{\text{JT}}(\lambda_-) - I_{\text{JT}}(\lambda_+)), \tag{B.20}$$

from integrating over all metric and dilaton configurations modulo all diffeomorphisms that leave the geometry invariant which is denoted by $\text{Vol}(\text{Diff})$. Evaluating (B.20) serves as an independent check to [58, 136], where an integral transformation was performed for $n$-boundaries of the $\lambda_- = \lambda_+ = 0$ partition function

$$Z_{\text{DT}}(0) = \frac{1}{\pi}\frac{\sqrt{\beta_1\beta_2}}{\beta_1 + \beta_2}, \tag{B.21}$$

to find the deformed partition function when $\lambda_- = \lambda_+ = \lambda$.

This is an example of the $T\overline{T}$ deformation with multiple deformation parameters $\lambda_{\pm}$ for $N = 2$ boundaries which can be thought of as the diagonal elements of the $N \times N$ deformation matrix $\lambda_{ij}$. It would be interesting to explore the consequences of the $T\overline{T}$ deformation when the scalar $T\overline{T}$ deformation coupling $\lambda$ is promoted to a non-trivial $N \times N$ matrix-valued quantity $\lambda_{ij}$.

## B.3 Lorentzian dS Disk

The final case we consider is the $\text{dS}_2$ disk. The Lorentzian $\text{dS}_2$ line element in global coordinates is

$$ds^2 = -dt^2 + \cosh^2 t\,d\tau, \tag{B.22}$$

where the radial coordinate $r$ in $\text{AdS}_2$ is replaced by a time coordinate $t$. The boundary curve is parametrized by $(t(u), \tau(u))$ with $u$ being the proper boundary time. Using the same boundary conditions (B.2) with the $\text{dS}_2$ metric (B.22), we find the curve at $O(\varepsilon^2)$:

$$t = -\ln\left(\frac{\varepsilon\tau'}{2}\right) + \frac{\varepsilon^2}{2}\left(\frac{\tau''^2}{\tau'^2} - \frac{\tau'^2}{2}\right). \tag{B.23}$$

The relevant component of the unit normal vector pointing in the time direction is

$$n^t = -\frac{\tau'\cosh t}{\sqrt{\tau'^2\cosh^2 t - t'^2}}, \tag{B.24}$$

and the trace of the extrinsic curvature is

$$K = \frac{\tau'\left(t''\cosh t - 2t'^2\sinh t + \tau'^2\cosh^2 t\sinh t\right) - \tau''t'\cosh t}{\left(\tau'^2\cosh^2 t - t'^2\right)^{\frac{3}{2}}}. \tag{B.25}$$

Substituting (B.23) into (B.25),

$$K = 1 - \varepsilon^2\left\{\tan\left(\frac{\tau}{2}\right), u\right\} + O(\varepsilon^4), \tag{B.26}$$

and we find the deformed boundary action is the deformed Schwarzian similar to the Euclidean AdS disk case (B.13).

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
