# Peer review of "$T\bar{T}$ in JT Gravity and BF Gauge Theory"

_SciPost Physics, doi:SciPost Phys. 13, 096 (2022)_

## Round 2 · Referee Report · Anonymous (Referee 1) · 2022-7-17

Strengths

The article is well written and contains interesting novel results.

Report

The authors study the TTbar deformation in JT gravity and BF theory. They show that deformation can be interpreted as a modification of the BF theory boundary. They also calculate the Wilson line correlators in the deformed theory.

The article meets the SciPost publication criteria, and I do recommend the publication on SciPost Physics after some clarification (see the Requested Changes section).

Requested changes

  1. The $e$ in the left-hand side of equation (3.28) should be a typo.

  2. The derivation of equation (4.11) is a bit confusing because the inverse relation of partial derivatives does not hold in general. For example, if one computes $\delta \gamma^{\tau\tau} / \delta \gamma^{\tau\tau}$ using the chain rule and (4.11) , the result will be 2 instead of 1.

  3. On page 36, the authors write, "Thus both $H_\lambda$ and $L_\lambda$ are (up to an overall sign) determined by the same square-root-type function of their undeformed values". In fact, the two functions are related by $\lambda \rightarrow -\lambda$ and therefore their flow equations differ by a minus sign. The minus sign in equation (4.51) indicates that they are discussing the Euclidean Lagrangian rather than the Hamiltonian.

  • validity: high
  • significance: high
  • originality: good
  • clarity: good
  • formatting: excellent
  • grammar: perfect

Author:  Stephen Ebert  on 2022-08-22  [id 2744]

(in reply to Report 1 on 2022-07-17)

We thank the referee for their thorough review and comments. We reply here to each question in turn:

1.) The fate of local operators in $T \overline{T}$-deformed theories is a very intriguing question. We agree with the referee that, at finite $\lambda$, local operators like the stress tensor are very likely undefined in such deformed theories.

Our equations for the modified boundary conditions in $3d$ Chern-Simons and in $2d$ BF theory should be interpreted in the same way as the trace flow equation $T^\mu_{\; \, \mu} ( \lambda ) = - 2 \lambda T \overline{T} ( \lambda )$ for a $T \overline{T}$ deformed CFT. Formally, this equation purports to relate the expectation values of local operators $T^\mu_{\; \, \mu} ( x )$ and $T \overline{T} ( x )$ at any finite value of the deformation parameter $\lambda$. However, although this relation is "exact in $\lambda$" insofar as it seems to hold to any order in conformal perturbation theory around the seed theory, it cannot be taken as a literal equality since local operators likely fail to exist in the full $T \overline{T}$-deformed theory.

We should add that this subtlety is not limited to our analysis, but is also present in other holographic investigations of $T \overline{T}$-deformed theories. For instance, in 1906.11251 the authors write down deformed boundary conditions for the $\mathrm{AdS}_3$ metric which correspond to a boundary $T \overline{T}$ deformation, and these mixed boundary conditions are also "exact in $\lambda$" in the same sense as ours. In fact, the authors of that work explicitly comment that "We are assuming that the stress tensor in the $T \overline{T}$-deformed CFT can be obtained as the response of the action to an arbitrary change in the background metric...it is not clear this must be the case in a non-local QFT such as $T \overline{T}$." The same issue applies to our analysis.

A deeper understanding of the relationship between such modified boundary conditions and the non-existence of local operators in $T \overline{T}$-deformed theories, although very interesting, is likely to be a difficult question which is beyond the scope of this work.

2.) Yes, our comment about dimensional reduction of boundary conditions is a reference to the fact that dimensional reduction of the usual $2d$ $T \overline{T}$ operator of Zamolodchikov and Smirnov 1608.05499 reproduces the $1d$ $T \overline{T}$-like operator of Gross et al. [1907.04873, 1912.06132], at least for deformations of a seed CFT. In fact, this is how the operator of Gross et al. is derived in Section 2.1 of 1907.04873. By beginning with the Zamolodchikov-Smirnov operator and using the trace flow equation (which holds for deformations of a CFT), those authors arrive at the form of the $1d$ deformation which we use in our work, e.g. in our equation (4.39). Because the holographic boundary conditions we consider correspond to deformations of conformal seed theories -- as they are related by dimensional reduction and change of variables to the conventional boundary conditions in $\mathrm{AdS} / \mathrm{CFT}$ -- this procedure also applies to our analysis.

3.) The thermodynamic stability of these models is also an interesting question. Because our deformed boundary conditions are engineered to reproduce the usual $1d$ $T \overline{T}$-like deformation (proposed by Gross et al. in [1907.04873, 1912.06132]) of the boundary theory, we believe that our models will exhibit the same stability properties as those considered in the works 2004.10138 and 2008.01333 that the referee mentions. For instance, it is possible for such theories to exhibit a negative specific heat capacity, and when $\lambda$ is sufficiently large that one has passed the Hagedorn transition, it is possible for such theories to be thermodynamically unstable. We added a footnote at the beginning of Section 6 to cite these two papers.

However, it is not clear to us whether this is a genuine property of these deformed theories, or an artifact of using an effective description which has broken down. We see a similar phenomenon in Section 6 of our paper, where the discrete series contribution begins to dominate over the principal series contribution above a certain critical temperature which corresponds to the Hagedorn transition. In that case, we argue that a different effective description should be used above this critical temperature, which may have different thermodynamic properties than the effective description used at low temperatures. Similar comments may apply to the observations regarding the thermodynamic properties of general $T \overline{T}$-deformed models at large $\lambda$.

4.) We defined the generators $P_0, P_1, P_2$ below equation (3.5) and added a reference where the definitions are located in Appendix A.2 of our paper.

Author:  Stephen Ebert  on 2022-07-22  [id 2678]

(in reply to Report 1 on 2022-07-17)

We thank the referee for his or her thoughts and comments on our paper. We address the requests:

1.) The $e$ in the left-hand side of equation (3.28) is indeed a typo and should actually be a $\nu$ instead as seen from the equations following after (3.28). We corrected this typo.

2.) We have clarified the derivation of (4.11) by emphasizing which quantities are held fixed in each of the derivatives. As the referee points out, the required inverse relation of partial derivatives holds only when the appropriate quantities are held fixed -- for instance, one has

$$\frac{\delta e^+\tau}{\delta \gamma^{\tau \tau}} \Big\vert{\delta e^+} = \left( \frac{\delta \gamma^{\tau \tau} \tau} \right)^{-1} \Big\vert$$
as in the Maxwell relations of thermodynamics. Applying this to the referee's example, one would compute, for instance,
$$\frac{\delta \gamma^{\tau \tau}}{\delta \gamma^{\tau \tau}}\Big\vert_{e_\tau^-} = \frac{\delta \gamma^{\tau \tau} }{\delta e^+\tau} \frac{\delta e^+\tau}{\delta \gamma^{\tau \tau}} \Big\vert_{e_\tau^-},$$
which gives $1$ rather than $2$. The difference between this $\frac{\delta \gamma^{\tau \tau}}{\delta \gamma^{\tau \tau}}\Big\vert_{e_\tau^-}$ example and the derivative appearing in the first line of equation (4.8) is that $I$ is not fixed in terms of $e_\tau^{\pm}$ as $\gamma^{\tau \tau}$ is. In particular, the manipulations used in going from the first line to the second line of (4.8) would not hold if $I$ were replaced with $\gamma^{\tau \tau}$. Here we calculate a total derivative of $I$ as both $e_\tau^+$ and $e_\tau^-$ vary, since both variations $\delta e_\tau^{\pm}$ appear in (4.7).

3.) We added a footnote and slight modifications to the wording on the distinction between the Euclidean Lagrangian and Hamiltonian on page 36. The standard convention is $H(X^\mu, p^\mu) = - L_E (X^\mu, \dot{X}^\mu)$, but the conventions in our paper (and Gross et al.'s $T\overline{T}$-deformed JT gravity paper 1907.04873) are different as the Hamiltonian is equivalent to the Euclidean Lagrangian: $I = \int d \tau H$.

---

## Round 2 · Referee Report · Anonymous (Referee 2) · 2022-8-17

Strengths

(1) The article is well written and self-sufficient.

(2) It reviews whatever is necessary for the reader to understand the main results even if he/she is not an expert on the subject.

Report

The paper studies TTbar-like deformation of the boundary theory dual to JT gravity and BF theory. Double trace deformations in holography in the gravity approximation can be thought of as imposing mixed boundary conditions on the various fields in the bulk. In this paper, the authors have derived the boundary condition of the 2d bulk metric analogous to the method adopted in the 3d Cherns Simons gravity case with the boundary CFT_2 deformed by TTbar. Finally, the authors compute perturbatively the correlation function of the Wilson loop operators of the deformed theory both in the 3d CS theory and its 2d reduced BF theory.

As I said above the paper is well written and has enough new results that deserve publication in SciPost. But at the same time, I would like to request the authors to clarify certain points that I itemize below.

Requested changes

I would request the readers to clarify the following points.

(1) TTbar deformed CFT_2 (or its 1d cousin) gives rise to a theory of quantum gravity in the UV. For example, in 2d it gives rise to JT gravity in flat space nominally coupled to conformal matter (works of Dubovsky et al), and 1d it gives rise to 1d quantum gravity coupled to quantum matter (works of Gross et al). It's quite well known that a theory of quantum gravity doesn't have a stress tensor. It's possible to make sense of stress tensor perturbatively in the TTbar coupling where one can still treat the theory as a local quantum theory. In the original paper of Zamolodchkov and Smirnov, the flow equations are constructed at the CFT/QFT limit and then lifted to high energies. But non-perturbatively there is no well-defined notion of the stress tensor or any other local operators. What makes sense non-perturbatively are the deformed energies of the eigenstate Given this, how would you interpret your equations (2.27)-(2.38) for the case of 3d or the analogous equation in section (3) and (4) in the case of 2d BF theory? It seems to me that all these equations computing the stress tensor (computed from the modified boundary condition) are exact in \lambda. It would be nice to clarify this point.

(2) In the last paragraph of page (28) it's suggested that one way to compute the mixed boundary condition in the case of BF theory is to dimensionally reduce the 2d TTbar operator. Are you trying to say that the dimensional reduction of TTbar of Zamolodchikov-Smirnov will give rise to TTbar-like operator of Gross et al in 1d? If so it would be nice to add a brief discussion regarding this.

(3) A TTbar deformed 1d quantum mechanical systems are not always thermodynamically stable. It has been shown in 2004.10138 and 2008.01333 that undeformed theories with linear specific heat are thermodynamically stable. Could you comment on the thermodynamic stability of the theory obtained by TTbar-like deformation of the boundary theory dual to BF theory in the bulk?

(4) I would request the authors to define P_{0,1,2} where they first appear around eq(3.5).

---

## Round 3 · Referee Report · Anonymous (Referee 2) · 2022-8-26

Report

I think I'm happy with the authors' reply to my previous questions and I strongly recommend the paper for publication.

I would however like to share the following comment regarding the response to my first question.

I think the equation $T_\mu^\mu(\lambda)=-2\lambda T\bar{T}(\lambda)$ should strictly be understood perturbatively to arbitrary order $\lambda$. I guess the authors also agree with this statement.

What is on a better footing at "arbitrary $\lambda$" is the flow equation when expressed in terms of the partition function (Aharony et al or Cardy) or the energy eigenfunction (Zamolodchikov-Smirnov). This can be written unambiguously at finite $\lambda$ because they don't involve the stress tensor (or any other local operators). Another quantity that is also well defined at arbitrary $\lambda$ is the kernel formula of Dubovsky et al for a generic local QFT (seed) or the Hashimoto-Kutasov kernel for a seed CFT. This is like an integral form of the above-mentioned flow equation that relates the theory at finite $\lambda$ with the seed. It would be interesting to see how one can construct the bulk-boundary map using the kernel formula as the definition of the theory at finite $\lambda$, something like rewriting the kernel formula in terms of bulk variables and reading off the dictionary (but not in terms of expectation of the stress tensor operator).

Similar kernel formula exists in $d=1$ as well. Once understood in $d=2$, one should be able to understand in $d=1$.

Requested changes

No changes are necessary.

---

## Round 3 · Referee Report · Anonymous (Referee 1) · 2022-8-29

Report

I thank the authors for the clarifications to address my comments and questions. I am now happy to recommend publication.

---

## Round 3 · List of Changes

1.) Added comments and clarifications to address the referees' remarks.
2.) Added references.

---

## Editorial Decision

published